# Distillation Scaling Laws

Dan Busbridge [1]   Amitis Shidani [2]   Floris Weers [1]   Jason Ramapuram [1]   Etai Littwin [1]   Russ Webb [1]

## Abstract

We propose a distillation scaling law that esti-mates distilled model performance based on a compute budget and its allocation between the student and teacher. Our findings mitigate the risks associated with large-scale distillation by enabling compute-optimal allocation for both the teacher and student to maximize student perfor-mance. We provide compute-optimal distillation recipes for two key scenarios: when a teacher already exists, and when a teacher needs train-ing. In settings involving many students or an existing teacher, distillation outperforms super-vised learning up to a compute level that scales predictably with student size. Conversely, if only one student is to be distilled and a teacher also re-quires training, supervised learning is generally preferable. Additionally, our large-scale study of distillation increases our understanding of the process and helps inform experimental design.

## 1. Introduction

The study of scaling laws (Hestness et al., 2017; Rosen-feld et al., 2020; Kaplan et al., 2020; Hoffmann et al., 2022) revealed that previously trained Language Models (LMs) could have been more capable if they had followed a *compute optimal* training paradigm, which determines the model size and the number of training tokens that give the best performing model under a given compute budget. Many subsequent works have followed compute optimal training (Dey et al., 2023; Muennighoff et al., 2023b).

The size of compute optimal models grows with compute (Hoffmann et al., 2022), which makes them challenging to use due to the *growth in inference costs*. In practice,

[1]Apple [2]University of Oxford, UK. Work done during an internship at Apple. For a full breakdown of contributions see Appendix J. Correspondence to: Dan Busbridge <dbus-bridge@apple.com>.

*Proceedings of the 42$^{nd}$ International Conference on Machine Learning*, Vancouver, Canada. PMLR 267, 2025. Copyright 2025 by the author(s).

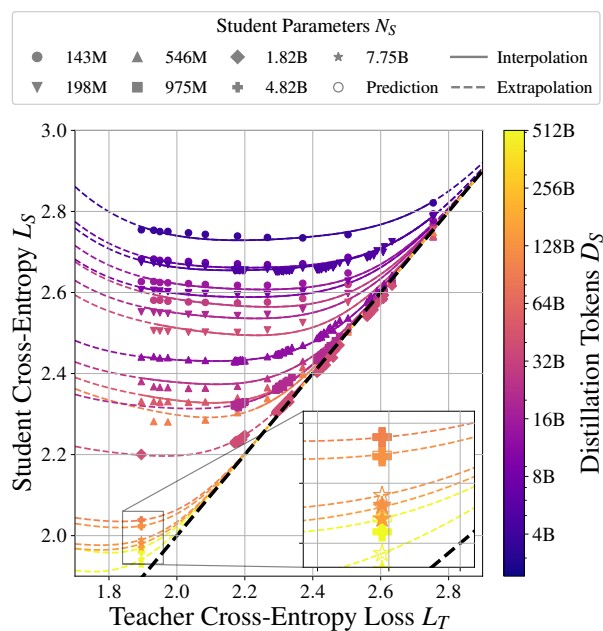

*Figure 1.* **Extrapolations of the Distillation Scaling Law.** The distillation scaling law (Equation 8) is fitted to students with high cross-entropy ($L_S > 2.3$) for a range of teachers with cross-entropies $L_T$. Solid lines represent predicted model behavior for unseen teachers for a given student configuration (interpolation), and dashed lines represent predicted model behavior beyond seen teachers and for low cross-entropy students ($L_S \leq 2.3$). The diagonal block dashed line indicates where student and teacher cross-entropies are equal. Teachers with lower cross-entropy gen-erally produce students with lower cross-entropy, until the *capac-ity gap* (see Figure 4 and Appendix B.3). As shown, a student can also outperform its teacher (see Figures 2, 3, and 41).

this means compute optimal models are slow, expensive to serve, consume more battery life, provide high barriers to entry for academic study, and have a significant carbon footprint. With an inference volume of billions of tokens per day (OpenAI & Pilipiszyn, 2021), the inference cost of an LM is typically significantly larger than its pretrain-ing cost (Chien et al., 2023; Wu et al., 2024a) and is going to further increase in an era of test-time compute scaling (Snell et al., 2024; Brown et al., 2024; Wu et al., 2024b).

Unsustainable inference costs have led to an alternative training paradigm, *overtraining* (Gadre et al., 2024), where

the amount of training data used is much greater than in the compute optimal case, enabling *small, capable models*. Overtrained models better satisfy compute optimality when compute is measured over a model's lifetime, rather than just the pretraining cost (Sardana et al., 2024). As supervised scaling laws follow power laws in model size and training data, diminishing returns in performance occur much sooner than in the compute-optimal case. To achieve reasonable capabilities, these models need to be trained on many trillions of tokens, (Snell et al., 2024; Brown et al., 2024; Wu et al., 2024b), which is expensive and time-consuming.

We seek models that match the performance of small overtrained models but at lower training cost. A popular candidate is *distillation* (Hinton et al., 2015), where a capable *teacher* LM produces targets for a smaller *student* LM. When distillation is used for LM pretraining, we will call this *distillation pretraining*. There are many explanations for *why* distillation works, from *dark knowledge transfer*, where information is contained in the ratio of probabilities of incorrect classes (Hinton et al., 2015), to being a form of regularization (Mobahi et al., 2020), or reducing noise in the learning process (Menon et al., 2020), among many other explanations. Despite a lack of consensus for why distillation works, distillation pretraining has produced more capable models than supervised pretraining in the Gemma and Gemini (Rivière et al., 2024), Minitron (Muralidharan et al., 2024; Sreenivas et al., 2024), and AFM (Gunter et al., 2024) families of LMs in of both pretraining loss and downstream evaluations. Yet, at the same time, Liu et al. (2024) reported that distillation produces *less* capable models than supervised pretraining does.

With such significant compute resources being devoted to distillation pretraining of LMs, it is *essential* to understand how to correctly allocate these resources, to produce the most capable models possible, and to understand if gains are even possible compared to supervised pretraining when both methods have access to the same resources (Dehghani et al., 2021).

To close this knowledge gap, we conduct an comprehensive, controlled study of distillation, with *transformer* students and teachers ranging from 143M to 12.6B parameters, trained on data of a few billion to 512B tokens. These experiments yield our *distillation scaling law*, which estimates student performance as a function of resources (the teacher, the student size, and the amount of distillation data). This resolves when distillation *is* and *is not* effective for producing models of a desired capability under practical resource constraints of interest. We find the following:

1. The cross-entropy of a student of size $N_S$ distilled on $D_S$ tokens from a teacher of size $N_T$ trained on $D_T$ tokens can be predicted using our distillation scaling

law (Equation 8).

2. The teacher size $N_T$ and number of teacher training tokens $D_T$ determine the student cross-entropy *only* through the resulting teacher cross-entropy $L_T = L_T(N_T, D_T)$ (Figure 3b).

3. The influence of the teacher cross-entropy upon the student loss follows a power law which transitions between two behaviors depending on the relative learning capacities of student and the teacher, reflecting a phenomenon in distillation called the *capacity gap*, where a stronger teacher produces a *worse* student. Our parameterization resolves outstanding questions about the capacity gap, showing that it is a gap in learning capacity (both hypothesis space and ability to optimize) between the teacher and student, and not only about their relative sizes, which is a special case.

Our results show that *distillation can not produce lower model cross-entropies than supervised learning* when both learning processes are given enough data or compute. However, distillation *is more efficient* than supervised learning *if both of the following are true*:

1. The total compute or tokens used for the student is not larger than student size-dependent threshold given by our scaling law (Section 5.1).

2. A teacher already exists, or the teacher to be trained has uses beyond a single distillation (Section 5.3).

We hope the laws and analyses we provide will guide the community to produce even more capable models with lower inference cost and lower lifetime compute costs.

## 2. Background

Predicting model performance is essential when scaling, as it lets us understand i) the value of increasing the available compute ($C$), and ii) how that compute should be distributed, typically between model parameters ($N$) and data ($D$), in order to achieve a model with desired properties. These properties may be predicting the data distribution sufficiently well, measured in cross-entropy ($L$), or achieving a level of performance on downstream tasks of interest.

Fortunately, cross-entropy *is predictable*, with substantial empirical and theoretical evidence that $L$ follows a power-law in parameters $N$ and data $D$ (measured in tokens)

$$\underbrace{L(N, D)}_{\text{Model Cross-Entropy}} = \underbrace{E}_{\text{Irreducible Error}} + \underbrace{\left(\frac{A}{N^\alpha} + \frac{B}{D^\beta}\right)^\gamma}_{\text{Model ability to mimic data}} , \quad (1)$$

where $\{E, A, B, \alpha, \beta, \gamma\}$ are task-specific positive coefficients[1] estimated from $n$ training runs $\{(N_i, D_i, L_i)\}_{i=1}^n$.

The choice of runs is critical; not all experiments enable identifying the coefficients of Equation 1. One could use *compute optimal* models whose size parameters $N^*$ and number of training tokens $D^*$ give the lowest cross-entropy subject to a compute constraint $C$

$$N^*, D^* = \underset{N,D}{\arg\min}\, L(N, D) \text{ s.t. FLOPs}(N, D) = C. \quad (2)$$

This is tempting, as compute-optimal models offer the largest loss variation for a total experiment budget. Unfortunately, compute optimal models have a *constant token to parameter ratio* $M \equiv D/N = \text{const.}$ (Hoffmann et al., 2022), removing a degree of freedom.

To achieve reliable identification of scaling coefficients, Hoffmann et al. (2022) uses two training strategies:

1. *(Fixed model, varied data)* The number of training tokens is varied for a fixed family of models.

2. *(IsoFLOP profiles)* Model size and training tokens are both varied subject to a total compute constraint.

Data from both strategies is then combined for the fit. See Appendix B for an extended background.

The goal of this paper is to predict the cross-entropy $L_S$ of a student produced by distillation. This will reveal the value of increasing compute for distillation, crucially, which distillation produces the student of a given size that achieves the lowest cross-entropy for a given compute budget.

## 3. Preliminaries

**Notation**   For a sequence $\boldsymbol{x}$, $\boldsymbol{x}^{(i:j)} = (x^{(i)}, x^{(i+1)}, \ldots, x^{(j)})$ is a slice of the sequence, and $\boldsymbol{x}^{(<i)} = \boldsymbol{x}^{(1:i-1)} = (x^{(1)}, \ldots, x^{(i-1)})$ is the *context* of $x^{(i)}$. We use the shorthand $\mathcal{X}^* = \cup_{n \in \mathbb{N}} \mathcal{X}^n$ to denote the set of sequences with arbitrary length $n \in \mathbb{N} = \{1, 2, \ldots\}$.

**Language modeling**   We focus on the LM setting where the training objective is to model the probability of sequences $\boldsymbol{x}$ of tokens $x_i$ drawn from a vocabulary $\mathcal{V} = \{1, 2, \ldots, V\}$. Let $f : \mathcal{V}^* \times \Theta \to \mathbb{R}^V$ be a next-token classifier parameterized by $\boldsymbol{\theta} \in \Theta$ whose outputs define a predictive categorical distribution over $\mathcal{V}$ given a context $\boldsymbol{x}^{(<i)}$

$$\hat{p}(x^{(i)} = a | \boldsymbol{x}^{(<i)}; \boldsymbol{\theta}) = \sigma_a(f(\boldsymbol{x}^{(<i)}; \boldsymbol{\theta})) = \sigma_a(\boldsymbol{z}^{(i)}), \quad (3)$$

where $\sigma_a(\boldsymbol{z}) = \exp(z_a)/\sum_b \exp(z_b)$ is the softmax function. The next-token classifier outputs $\boldsymbol{z}^{(i)} = f(\boldsymbol{x}^{(<i)}; \boldsymbol{\theta})$ are the *logits*.[2] Autoregressive LMs produce sequence likelihoods through $\hat{p}(\boldsymbol{x}; \boldsymbol{\theta}) = \prod_{i=1}^L \hat{p}(x^{(i)} | \boldsymbol{x}^{(<i)}; \boldsymbol{\theta})$ and are trained to maximize this likelihood on observed data through the Next Token Prediction (NTP) loss

$$\mathcal{L}_{\text{NTP}}(x^{(i)}, \boldsymbol{z}^{(i)}) = -\sum_{a=1}^V \boldsymbol{e}(x^{(i)})_a \log \sigma_a(\boldsymbol{z}^{(i)}), \quad (4)$$

where $\boldsymbol{e}(i)$ is the $i$-th basis vector. It is common to also use the following token-level $Z$-loss to improve training stability (Chowdhery et al., 2023; Wortsman et al., 2023)

$$\mathcal{L}_Z(\boldsymbol{z}^{(i)}) = ||\log Z(\boldsymbol{z}^{(i)})||_2^2 = \left\|\log \sum_{a=1}^V \exp(z_a^{(i)})\right\|_2^2. \quad (5)$$

**Distillation**   In distillation, a *teacher* with predicted next-token distribution $\hat{p}_T(x^{(i)} | \boldsymbol{x}^{(<i)}; \boldsymbol{\theta}_T)$ and corresponding logits $\boldsymbol{z}_T^{(i)}$ replaces the one-hot basis vector in Equation 4 and is used as the target for a student predicted next-token distribution $\hat{q}_S(x^{(i)} | \boldsymbol{x}^{(<i)}; \boldsymbol{\theta}_S)$ and corresponding logits $\boldsymbol{z}_S^{(i)}$. The resulting *knowledge distillation loss* is used to optimize the student parameters

$$\mathcal{L}_{\text{KD}}(\boldsymbol{z}_T^{(i)}, \boldsymbol{z}_S^{(i)}) = -\tau^2 \sum_{a=1}^V \sigma_a\left(\frac{\boldsymbol{z}_T^{(i)}}{\tau}\right) \log \sigma_a\left(\frac{\boldsymbol{z}_S^{(i)}}{\tau}\right), \quad (6)$$

and is equivalent to optimizing the Kullback-Leibler Divergence (KLD) between the teacher and student predictions. $\tau > 0$ is the distillation *temperature*. Combining the losses together results in a total token-level loss for the student:

$$\mathcal{L}_S(x^{(i)}, \boldsymbol{z}_T^{(i)}, \boldsymbol{z}_S^{(i)}) = (1 - \lambda)\, \mathcal{L}_{\text{NTP}}(x^{(i)}, \boldsymbol{z}_S^{(i)})$$
$$+ \lambda\, \mathcal{L}_{\text{KD}}(\boldsymbol{z}_T^{(i)}, \boldsymbol{z}_S^{(i)}) + \lambda_Z\, \mathcal{L}_Z(\boldsymbol{z}_S^{(i)}). \quad (7)$$

## 4. Distillation Scaling Laws

Here we outline the steps taken to arrive at our distillation scaling law. First we describe the experimental setting (Section 4.1) and the experiments needed to determine the scaling coefficients (Section 4.2). Given the empirical observations, we discuss the form our distillation scaling law takes (Section 4.3), find the coefficients, and verify the law under extrapolation (Section 4.4).

### 4.1. Experimental Setup

All models are based on Gunter et al. (2024) and use decoupled weight decay Loshchilov & Hutter (2019) for regularization, as well as a simplified version of $\mu$P (Yang &

---

[1]Hoffmann et al. (2022) use $\gamma = 1$, whereas Kaplan et al. (2020) use $\beta = 1$. We observe a significantly better fit and extrapolation without coefficient tying, which may be due to our use of Maximal Update Parameterization ($\mu$P) (see Section 4.1).

[2]We *do not* write this as $\boldsymbol{z}^{(<i)}$ to avoid confusion with the sequence $\boldsymbol{z}^{(<i)} = (\boldsymbol{z}^{(1)}, \ldots, \boldsymbol{z}^{(i-1)})$.

*Table 1.* Expressions related to scaling laws used in this work. In each case, $S$ always refers to *student* and *not* supervised.

| Expression | Meaning |
|---|---|
| $N\,/\,N_S\,/\,N_T$ | The number of model/student/teacher *non-embedding* parameters. Whenever we mention parameters in text, we always mean *non-embedding* parameters unless explicitly stated otherwise. See Appendix H.2 for more details. |
| $D\,/\,D_T$ | The number of tokens the model/teacher is pretrained on. |
| $D_S$ | The number of tokens the student is distilled on. |
| $M \equiv D/N$ | The tokens per parameter ratio, or $M$-ratio. In Hoffmann et al. (2022), $M$ takes a compute optimal value $M^* \approx 20$ which is the *Chinchilla rule of thumb*. |
| $L \approx L(N, D)$ | The *model cross-entropy*, which is the model validation cross entropy *under data*, estimated by the supervised scaling law for a model with $N$ parameters trained on $D$ tokens. (Equation 1). |
| $L_T \approx L(N_T, D_T)$ | The *teacher cross-entropy*, which is the teacher validation cross entropy *under data*, estimated by the supervised scaling law for a teacher with $N_T$ parameters trained on $D_T$ tokens. |
| $L_S \approx L_S(N_S, D_S, L_T)$ | The *student cross-entropy*, which is the student validation cross entropy *under data*, estimated by our distillation scaling law for a student with $N_S$ parameters distilled on $D_S$ tokens using a teacher with pretraining loss $L_T$ (Equation 8). |
| $\widetilde{L}_S \approx L(N_S, D_S)$ | The *student supervised cross-entropy*, which is the student validation cross entropy *under data if the student had been trained in a supervised way*, estimated by the supervised scaling law for a student with $N_S$ parameters trained on $D_S$ tokens. |

Hu, 2021; Yang & Littwin, 2023; Yang et al., 2022; Wortsman et al., 2023; Yang et al., 2023), following $\mu$P (simple) in (Wortsman et al., 2024). $\mu$P simplifies the scaling law experimental setup as it enables *hyperparameter transfer* of the learning rate across model sizes. We validate that $\mu$P functions as expected for distillation in Appendix G.3. Models have sizes which range from 143M to 12.6B parameters, and we allow the teacher to be smaller or larger than the student. Multi-headed attention (MHA) is used, with Pre-Normalization (Nguyen & Salazar, 2019) using RMSNorm (Zhang & Sennrich, 2019). We train all models with a sequence length of 4096, with Rotary Position Embedding (RoPE) (Su et al., 2024). We use the English-only subset of the C4 dataset (Raffel et al., 2020) for all experiments. For all distillation trainings, the teacher is trained on a different split from the student. Except for the largest models, all Chinchilla-optimal models do not repeat data. Full hyperparameters and details can be found in Appendix I. As our goal is to understand the role of the teacher in the distillation process we distill in the *pure distillation* case ($\lambda = 1$, Equation 7) to avoid confounding coming from the data, as was done in Stanton et al. (2021). We verify the choice $\lambda = 1$ produces results statistically similar to the optimal $\lambda^*$ (see Appendix G.1). Similarly, all experiments use distillation temperature ($\tau = 1$), as we found this produces the best performing students (see Appendix G.2).

### 4.2. Distillation Scaling Law Experiments

Here we discuss the experiments that produce the data for fitting our distillation scaling law. The distillation scaling law will estimate *student cross-entropy* $L_S$[3], which

---

[3]By *cross-entropy*, we always mean with respect to *data*, *not* the teacher. We summarize our scaling law notation in Table 1.

in general depends on the student parameters $N_S$, number of distillation tokens $D_S$, the teacher parameters $N_T$ and the number of teacher training tokens $D_T$: $L_S \approx L_S(N_S, D_S, N_T, D_T)$. As discussed in Section 2, only certain combinations of data support reliable identification of scaling law coefficients. We combine three experimental protocols to produce data for our distillation scaling law fit.

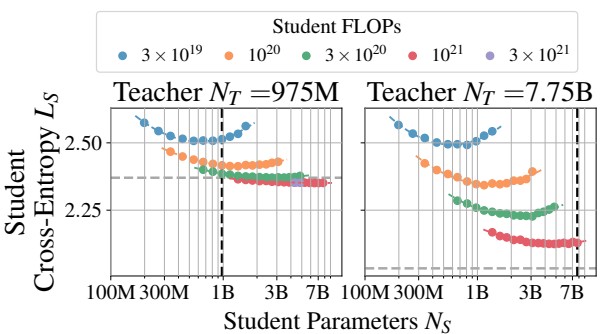

*Figure 2.* **Fixed $M$ Teacher/Student IsoFLOP profiles.** Two of six teachers with a token-to-parameter ratio $M_T = D_T/N_T \approx 20$ are distilled into students across four IsoFLOP profiles defined by compute budgets $C_S \in \{3 \times 10^{19}, 10^{20}, 3 \times 10^{20}, 10^{21}\}$ FLOPs. A small number of additional distillations were also performed using $C_S = 3 \times 10^{21}$ FLOPs. Here, $C_S$ *only* includes the standard training cost of a model of size $N_S$ trained on $D_S$ tokens, i.e. the cost of teacher training and teacher inference is not included. Horizontal and vertical dashed lines indicate teacher cross entropy $L_T$ and size $N_T$ respectively. See Appendix E.4, Figure 38a for all six teacher profiles corresponding to $N_T \in \{546M, 975M, 1.82B, 2.72B, 4.82B, 7.75B\}$.

**Fixed $M$ Teachers/Student IsoFLOPs** To simplify the experimental protocol we make the following assumption: *Training a student $(N_S, D_S)$ on the signal provided by a teacher $(N_T, D_T)$ is qualitatively similar to training that student on a fixed dataset.* As power law behavior has been observed in a wide variety of datasets and domains (Henighan et al., 2020), it is expected that there should be a power law behavior in $(N_S, D_S)$ given a fixed teacher.

To identify these coefficients correctly, a similar protocol to the Chinchilla protocol described in Section 2 should be performed. However, we cannot do this for *only* one teacher, as the way student size and tokens affects downstream performance may be different for different teachers, just as the scaling laws are different for different domains and dataset. For distillation we anticipate this is the case so that different teachers produce different students. To produce the widest range of teachers for a compute budget, we train six Chinchilla-optimal ($M_T = D_T/N_T \approx 20$) teachers ranging from 198M to 7.75B parameters. [4] For

---

[4]We generally refer to these as *fixed-m* models rather than *Chinchilla-optimal* models as we do not yet know whether $M \approx 20$ is a good choice in this specific setting.

each of those teachers, we distill into students with four IsoFLOP profiles, taking only the standard training cost into account. The resulting student cross-entropies are in Figure 2. We note that in some cases, the student is able to outperform the teacher, i.e. exhibits *weak-to-strong-generalization* (Burns et al., 2024; Ildiz et al., 2024) and investigate this further in Appendix E.7.

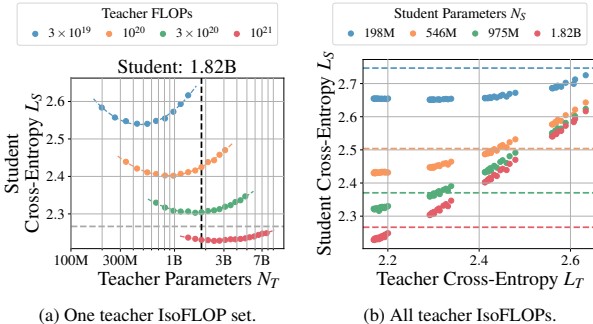

*Figure 3.* **IsoFLOP Teacher/Fixed $M$ Students. (a)** One of four students with a token-to-parameter ratio $M_S = D_S/N_S \approx 20$ is distilled from teachers with four IsoFLOP profiles defined by compute budgets $C_T \in \{3 \times 10^{19}, 10^{20}, 3 \times 10^{20}, 10^{21}\}$ FLOPs. For all four student sizes $N_S \in \{546M, 975M, 1.82B, 7.75B\}$, see Appendix E.4, Figure 38b. **(b)** All profiles are plotted against teacher cross-entropy $L_T$. Horizontal (vertical) dashed lines show student supervised cross-entropy $\widetilde{L}_S$ (student size $N_S$).

**IsoFLOP Teachers/Fixed $M$ Students** The fixed-$M$ teacher IsoFLOP student protocol is insufficient to identify how $N_T$ and $D_T$ *independently* influence student cross-entropy. To ensure our experiment can detect this influence, we perform experiments where the student $(N_S, D_S)$ is fixed, and vary $N_T$ and $D_T$ subject to a compute constraint, i.e., a teacher IsoFLOP. We perform distillations into four Chinchilla-optimal ($M_S = D_S/N_S \approx 20$) students ranging from 198M to 1.82B parameters from teachers trained according to four IsoFLOP profiles. The resulting student cross-entropies are in Figure 3.

**Fixed $M$ Teachers/Fixed $M$ Students** Finally, although not necessary for fitting our distillation scaling law, it is instructive to see how student cross-entropies vary over as large a range as possible. To achieve this, we train fixed-$M$ teacher fixed-$M$ student combinations, with ten teachers with $M_T \approx 20$, and students of five sizes, with at least four choices of $M_S$ per student. The resulting student cross-entropies for two of the students are in Figure 4.

**Capacity gap** In Figure 4, we observe the *capacity gap*, where *improving teacher performance does not always improve student performance, and even reduces student performance eventually*. The capacity gap has been observed often in distillation (see Appendix B.3). The KLD between

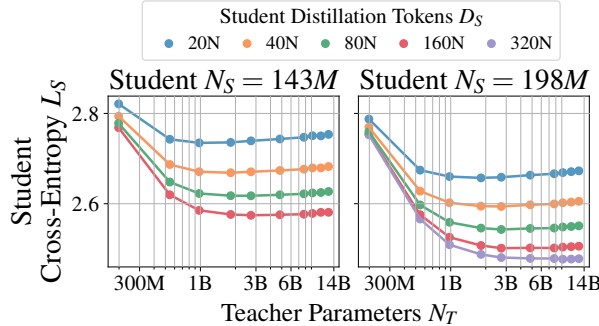

*Figure 4.* **Fixed $M$ Teacher/Fixed $M$ Student.** Students of two sizes trained with different token-to-parameter ratios $M_S = D_S/N_S \in \{20, 40, 80, 160, 320\}$ are distilled from teachers of various sizes with a token-to-parameter ratio $M_T = D_T/N_T \approx 20$. The capacity gap is visible: student cross-entropy decreases to an optimum and then increases with increasing teacher size $N_T$.

teacher and student is an increasing function of teacher capability in all cases (see Appendix E.3), which means as the teacher improves its own performance, the student finds the teacher more challenging to model, eventually preventing the student from taking advantage of teacher gains. We use calibration metrics to investigate aspects that the student finds challenging to model in Appendix E.8. In Appendices C.1 and C.2 we offer a simple explanation in a kernel regression and synthetic Multi-Layer Perceptron (MLP) setting and, to the best of our knowledge, are the first controlled demonstrations of the capacity gap.

### 4.3. Distillation Scaling Law Functional Form

We need to determine the functional form of the distillation scaling law. First, we observe that *contributions from teacher size $N_T$ and pretraining tokens $D_T$ are summarized by the teacher cross-entropy $L_T$.* This can be seen from Figures 1 and 3b which contains the IsoFLOP Teacher/Fixed $M$ Students of Figure 3, yet smooth dependence as a function of $L_T$ is observed. Next, the distillation scaling law should reflect the following properties:

1. An *infinitely capable student* should be able to model *any* teacher: $\lim_{N_S, D_S \to \infty} L_S(N_S, D_S, L_T) \to L_T$.

2. A *random* teacher produces *random* students *independent* of how capable those students are: $\lim_{L_T \to \infty} L_S(N_S, D_S, L_T) \to L_T$.

3. There is a *capacity gap*: making a teacher too capable eventually reduces the student performance.

A transition between two power law regions: i) where the student is a stronger learner than the teacher, and ii) where the student is a weaker learner than the teacher is described by a broken power law (Caballero et al., 2023). Together,

we propose that student cross-entropy follows a broken power law in $L_T$ and a power law in $N_S$ and $D_S$:

$$\underbrace{L_S(N_S, D_S, L_T)}_{\text{Student cross-entropy}} = \underbrace{L_T}_{\text{Teacher cross-entropy}}$$
$$+ \underbrace{\frac{1}{L_T^{c_0}} \left(1 + \left(\frac{L_T}{\widetilde{L}_S d_1}\right)^{1/f_1}\right)^{-c_1 f_1} \left(\frac{A}{N_S^{\alpha'}} + \frac{B}{D_S^{\beta'}}\right)^{\gamma'}}_{\text{Student ability to mimic teacher}} \quad (8)$$

where $\{c_0, c_1, d_1, f_1, \alpha', \beta', \gamma'\}$ are positive coefficients to be fitted following the procedure outlined in Appendix F.2 on the data produced in Section 4.2. The first two properties of our distillation scaling law can be readily checked. For the third, recall, $\widetilde{L}_S = L(N_S, D_S)$ is the cross-entropy a student would have achieved if it had been trained in a supervised way (Table 1), and is determinable from the supervised scaling law (Equation 1). The capacity gap behavior follows from a transition based on the ratio of the *algorithmic learning capacities* of the student and teacher, when $L_T / \widetilde{L}_S \equiv L(N_T, D_T)/L(N_S, D_S) = d_1$, which can be interpreted as a measure of the *relative learning abilities* of the teacher and the student on a reference task.

## 4.4. Distillation Scaling Law Parametric Fit

We use the teachers $(N_T, D_T)$ for fitting our supervised scaling law (Appendix E.2), and all the data for fitting our distillation scaling law (Equation 8). Our fitting procedure is described in detail in Appendix F and the resulting scaling coefficients are presented in Appendix F.3. Our supervised and distillation scaling laws fit the observations at the level of $\lesssim 1\%$ relative prediction error, including when extrapolated from weaker to stronger models (see Figure 5b).

As a further verification, we confirm that for a fixed model size, distillation in the infinite data regime is consistent with supervised learning on infinite data (Appendix E.6).

## 5. Distillation Scaling Law Applications

Here, we apply our distillation scaling law (Equation 8) and investigate scenarios of interest. Typically, the resources in distillation pretraining include a compute budget, or a dataset containing a number of tokens. For a distillation process, the compute cost can be approximated by

$$\text{FLOPs} \approx \underbrace{3F(N_S)D_S}_{\substack{\text{Student} \\ \text{Training}}} + F(N_T)(\underbrace{\delta_T^{\text{Lgt}} D_S}_{\substack{\text{Teacher} \\ \text{Logits}}} + \underbrace{\delta_T^{\text{Pre}} 3 D_T}_{\substack{\text{Teacher} \\ \text{Training}}}) \quad (9)$$

where $\delta_T^{\text{Lgt}}, \delta_T^{\text{Pre}} \in [0, 1]$ indicate whether we account for the cost of teacher logit inference for the student targets[5],

---

[5] Appendix G.4 evaluates distribution truncation via Top-$p$ and Top-$k$ to mitigate the overhead of computing these logits online.

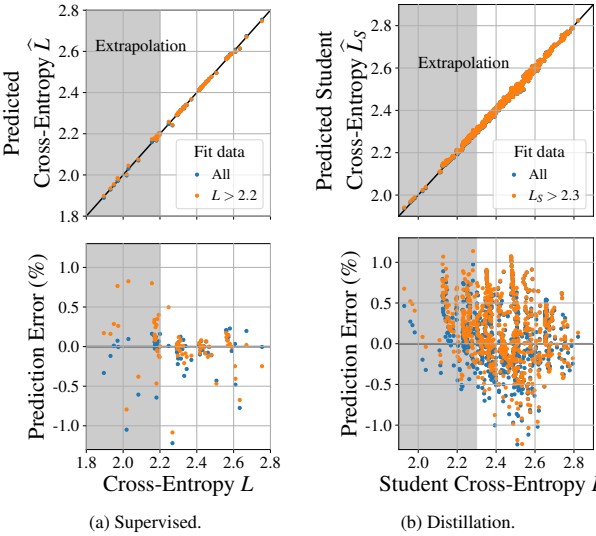

*Figure 5.* **Scaling law fits. (a)** The supervised scaling law (Equation 1) applied to the data in Figure 36a. **(b)** Our distillation scaling law (Equation 8) applied to the data in Figures 2 to 4. Orange points show predictions from a scaling law fitted on high cross-entropy models, for which the grey region is extrapolation. Blue points show predictions from a scaling law fitted on all data.

*Table 2.* The four practical distillation settings we study, and how their compute accounting is implemented through Equation 9.

| Compute Scenario | $\delta_T^{\text{Lgt}}$ | $\delta_T^{\text{Pre}}$ | Description |
|---|---|---|---|
| Best case (fully amortized teacher) | 0 | 0 | The teacher incurs no additional FLOPs and so we are free to choose the teacher $L_T^*$ that minimizes the student cross-entropy. |
| Teacher inference | 1 | 0 | We don't account for the teacher cost because the teacher already exists, or we intend to use the teacher as e.g., a server model. We still need to pay to use it for distilling a student. |
| Teacher pretraining | 0 | 1 | The teacher needs training, but we store the logits for reuse, either during training, or after training for distilling into sufficiently many students. |
| Teacher pretraining + inference | 1 | 1 | The teacher needs training and we pay for distilling into one student, the worst case scenario. |

and teacher pretraining cost in the total compute budget (see Table 2). $F(N)$ is the number of Floating Operations (FLOPs) a model with $N$ parameters performs per token during a forward pass. $F(N) \approx 2N$ is often used, giving supervised FLOPs $\approx 6ND$. We cannot use the $2N$ approximation, as (i) using *non-embedding* parameters $N$ induces systematic errors (Porian et al., 2024), and (ii) we are interested in *small models with large context sizes* where the FLOP contribution from attention is significant. To resolve these issues, we derive a simple expression $F(N) \approx 2N(1 + c_1 N^{-1/3} + c_2 N^{-2/3})$ for *fixed-aspect ratio* models in Appendix H.1, and recommend the scaling community consider adopting this hyperparameter setting.

## 5.1. Fixed Tokens or Compute (Best Case)

To build intuition for when distillation may (and may *not*) be beneficial, we ask *how well can distillation do in the best case scenario, compared with supervised learning?* We superimpose the data of Figures 2 and 3 onto contours of distilled cross-entropy $L_S$ compared to a supervised model with the same resources $\widetilde{L}_S$ (Figure 6).

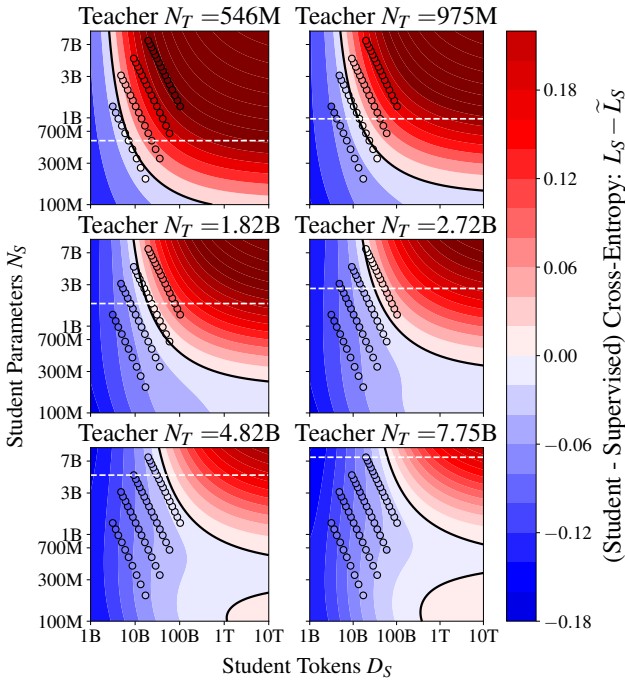

*Figure 6.* **Fixed-$M$ Teacher/IsoFLOP students (data).** The cross-entropy difference between best case distillation and supervised learning, as determined by our supervised and distillation scaling laws (Figure 5) for six student sizes $N_S \in \{546M, \ldots, 7.75B\}$ and a range of token budgets $D_S \in [1B, 10T]$. The scatter points correspond to cross-entropies achieved by the runs in Figures 2 and 38a. Blue indicates distillation outperforms supervised learning ($L_S < \widetilde{L}_S$), while red indicates supervised learning outperforms distillation ($L_S > \widetilde{L}_S$). The white horizontal dashed line indicates the teacher size.

**Supervised learning always outperforms distillation given enough student compute or tokens.** For a modest token budget, distillation is favorable; however, when a large number of tokens are available, supervised learning outperforms distillation. This is expected; in the large data regime, supervised learning can find the best solution limited by model size $N$ (Equation 1), whereas distillation *only* finds this solution for the optimal teacher $L_T^*$ (see Appendix E.6), and is otherwise limited by the distillation process. Although this finding appears to contradict the *patient teacher* finding of Beyer et al. (2022), it *does not*, pri-

marily due to the differences in supervised baselines (see Appendix D.1). A compute-constrained student version of Figure 6 and IsoFLOP Teacher/Fixed $M$ student contours are provided in Appendix D.2.

## 5.2. Fixed Tokens or Compute (Teacher Inference)

Next, we focus on the common scenario of planning to distill and trying to decide among an existing set of teachers $\{(L_T^{(i)}, N_T^{(i)})\}_{i=1}^n$. A larger teacher may provide a better learning signal (lower cross-entropy) but will also be more expensive to use because of the teacher logits cost (Equation 9, $\delta_T^{\text{Lgt}} = 1$), inducing a trade-off. Given a target student size $N_S$ and budget $D_S$ or $C_{\text{Total}}$, the only degree of freedom is the choice of teacher.

**For a fixed data budget, as the student size increases, teacher cross-entropy should be decreased as a power law.** Here, the compute cost from $N_T$ is not relevant as we are considering a token budget. Student cross-entropy at different distillation token budgets is shown in Figure 7. An equivalent plot for different student sizes while varying

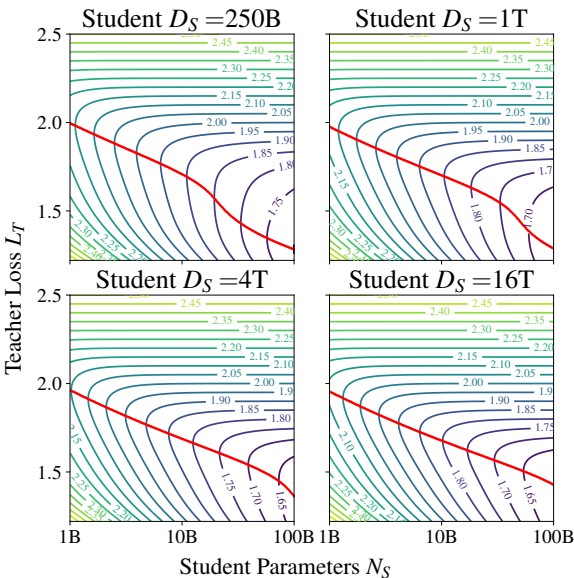

*Figure 7.* **Students given a teacher and token budget.** Contours of student cross-entropy $L_S$ for a range of teachers and students across four distillation token budgets $D_S \in \{250B, 1T, 4T, 16T\}$. The red line indicates the optimal teacher cross-entropy $L_T^*(N_S, D_S) = \arg\min_{L_T} L_S(N_S, D_S, L_T)$ for each student size and distillation token budget.

tokens is shown in Appendix D.3. We see that the optimal teacher loss $L_T^*$ (red line) decreases as a power law with student size $N_S$ until $L_S$ matches $L_T^*$, when there is an inflection point in $L_T^*$, causing the teacher loss de-

crease to sharpen with $N_S$. This generalizes the observation of Zhang et al. (2023a), that *"Optimal teacher scale almost consistently follows a linear scaling with the student scale across different model architectures and data scales."* which is a special case of our finding when the teachers are compute optimal (Figure 36a). Note that our findings consistently show that teacher cross-entropy $L_T$ determines student cross-entropy $L_S$, *not* $N_T$ itself (which leads to a given $L_T$). We investigate a fixed compute budget setting for teacher inference only in Appendix D.3.

### 5.3. Compute Optimal Distillation

We extend the analysis of Hoffmann et al. (2022) to distillation, giving *compute optimal distillation*, determining how to produce the student of a desired size $N_S$ with the lowest cross-entropy given a compute budget $C$

$$D_S^*, N_T^*, D_T^* = \underset{D_S, N_T, D_T}{\arg\min} \; L_S(N_S, D_S, N_T, D_T)$$
$$\text{s.t.} \quad \text{FLOPs} = C, \tag{10}$$

To present the best and worst case for incorporating teacher inference into the compute constraints, we consider *all scenarios* presented in Table 2. We also compare against the optimal *supervised* performance. To find the minima in Equation 10 we perform constrained numerical minimization using Sequential Least SQuares Programming (SLSQP) (Kraft, 1988) in `SciPy` (Virtanen et al., 2019).

**Supervised learning always matches optimal distillation at sufficient compute budget, with the intersection favoring supervised learning increasing as student size grows.** In Figure 8 we see that supervised learning always matches the best case distillation setting at some total compute budget, as anticipated from the asymptotic analysis in Figure 40. The compute transition point at which supervised learning becomes preferable to distillation increases as a function of student size. See also Figure 6. We also observe that *smaller models are more likely to benefit from supervised pretraining*, whereas *larger models are more likely to benefit from distillation*.

**When teacher training is included in the compute, the best student cross-entropy is always higher than in the supervised setting.** This means that if the *only* aim is to produce the best model of a target size and you do not already have access to a teacher, then supervised learning should be used, instead of training a teacher and then distilling. Conversely, if the intention is to distill into a family of models, or use the teacher as a server model, distillation *may* be computationally preferable to supervised learning. On reflection, this finding should be expected, otherwise it would imply that given for a total end-to-end compute, distillation outperforms maximum likelihood optimization.

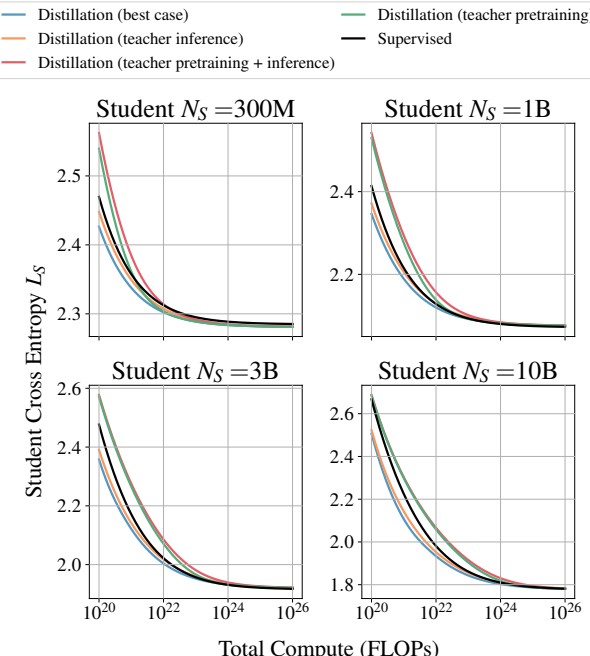

*Figure 8.* **Compute-optimal distilled student performance.** The best cross-entropy students of four sizes $N_S \in \{300M, 1B, 3B, 10B\}$ can achieve in the four distillation scenarios considered (Table 2) and in a supervised baseline, as total compute is varied.

*Table 3.* Optimal compute allocation trends.

| Student size | Compute (FLOPs) | Allocation |
|---|---|---|
| Small ($\lesssim 3B$) | Small ($\lesssim 10^{21}$) | Mostly teacher pretraining. |
| Small ($\lesssim 3B$) | Large ($\gtrsim 10^{25}$) | Evenly divided between student training and teacher inference, much less on teacher pretraining. |
| Large ($\gtrsim 10B$) | Small ($\lesssim 10^{21}$) | Mostly standard student training. |
| Large ($\gtrsim 10B$) | Large ($\gtrsim 10^{25}$) | Equally divided between student training, teacher inference, and teacher pretraining. |

A detailed discussion of the compute optimal configurations that produce $(N_S^*, N_T^*, D_T^*)$ for all scenarios is provided in Appendix D.4.

To build intuition for how quantities interact, we take the most complex scenario, *teacher pretraining + inference*. A view of the optimal distillation setup as compute varies is presented in Figure 9.

Student and teacher tokens scale as a power law, with student tokens scaling at a faster rate. Optimal teacher size increases initially until it is slightly larger than the student, after which it plateaus. This plateau occurs because inference with large teachers is expensive, and with an increase in the number of student tokens, it becomes more efficient to overtrain the teacher.

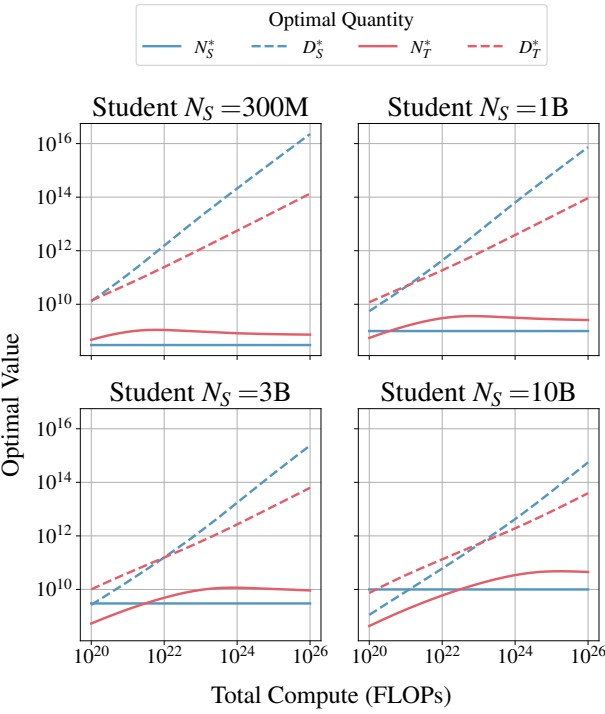

Figure 9. **Optimal configurations accounting for teacher pre-training and teacher logit inference costs.** For student sizes $N_S \in \{300M, 1B, 3B, 10B\}$, the student ($N_S^*$, $D_S^*$), and teacher ($N_T$, $D_T^*$) configurations minimizing the student cross entropy $L_S^*$ subject to a total compute budget that accounts for both teacher pretraining and teacher logit inference costs.

The values in Figure 9 can be recombined to produce the compute terms in Equation 9 as shown in Appendix D.4, Figure 29. We summarize the trend in Table 3.

## 6. Conclusion

We propose a distillation scaling law that estimates distilled model performance based on a compute budget and its allocation between the student and teacher. We then used our law to study practical distillation scenarios, and showed that distillation is only more efficient than supervised learning if (i) the total compute or tokens used for distillation is not larger than a student size-dependent threshold, *and* (ii) a teacher already exists, or the teacher to be trained has applications beyond its use in a single distillation. Moreover, we used this law to determine optimal distillation scenarios that can outperform supervised learning, enabling practitioners to select the best teacher for their use case. This work represents the largest controlled empirical study of distillation we are aware of, with systematic ablations of common distillation techniques. Just as supervised scaling has mitigated risks in supervised pretraining, our find-

ings offer a roadmap for producing smaller, more powerful models with lower inference costs, reducing carbon footprints, and enhancing the feasibility of test-time scaling.

## Acknowledgments

We thank Pierre Ablin, Samira Abnar, Samy Bengio, Miguel Sarabia del Castillo, Federico Danieli, Eeshan Gunesh Dhekane, Angeliki Giannou, Adam Goliński, Tom Gunter, Navdeep Jaitly, Tatiana Likhomanenko, Ian Magnusson, Preetum Nakkiran, Skyler Seto, Josh Susskind, Kunal Talwar, Barry Theobald, Vimal Thilak, Oncel Tuzel, Chong Wang, Jianyu Wang, Luca Zappella, and Shuangfei Zhai for their helpful feedback and critical discussions throughout the process of writing this paper; Okan Akalin, Hassan Babaie, Peter Bukowinski, Denise Hui, Mubarak Seyed Ibrahim, David Koski, Li Li, Cindy Liu, Cesar Lopez Nataren, Ruoming Pang, Rajat Phull, Evan Samanas, Guillaume Seguin, Dan Swann, Shang-Chen Wu, Joe Zhou, Kelvin Zou, and the wider Apple infrastructure and Foundation Model teams for assistance with developing and running scalable, fault tolerant code. Names are in alphabetical order by last name within group.

## Impact Statement

This work shows how to apply the framework of scaling laws to the distillation setting, and investigates distillation as a viable alternative to the overtraining paradigm for producing capable language models. Our findings demonstrate when distillation *should* and *should not* be performed, from a compute efficiency perspective, compared to supervised learning. There are a number of benefits to this:

1. As compute-optimal recipes for distillation are now known, there is greater opportunity for producing powerful models with lower inference costs. Lowering inference costs reduces the largest component of the total carbon footprint of language models (from training to inference).

2. When combined with established scaling laws, there is a larger space of models for which compute-optimal configurations are known. To produce models with a given capability, the compute, hardware and climate costs have been reduced compared to before, thanks to the identification of the optimal recipe.

3. Our distillation scaling law reduces compute usage by eliminating unnecessary experimentation across various hyperparameters and distillation settings. It is now understood that the primary driver of student cross-entropy is teacher cross-entropy, and so teacher size and tokens can be removed as search dimensions.

4. Small powerful models democratize the study of highly capable models, enabling broader participation in the study of their capabilities and safety aspects.

However, there are potential negative consequences:

1. Using distillation as part of a training pipeline introduces new sources of bias. Teacher models may contain bias from their pretraining data. Even if a student is distilled on unbiased data, the bias of the teacher will be inherited by the student.

2. Small powerful language models are more efficient during inference, reducing the amount of resources needed for malicious actors to achieve their goals, such as generating targeted misinformation at scale.

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

# Appendices

# A. Limitations

This work has several limitations that we are aware of:

- Our work is performed in the language modeling setting only. Although there is good evidence that the functional form of scaling laws applies across domains (Henighan et al., 2020), we cannot be absolutely certain that distillation behaves in the way we describe in this work in all domains.

- We perform our analysis on the English subset of C4 dataset (see Appendix I). This means that for our larger token runs, data has been repeated. Although it was shown in Muennighoff et al. (2023b) that on the C4 dataset, repeating data up to 4 times has negligible impact to loss compared to having unique data, this was shown in the supervised setting, and we cannot be absolutely certain that the same applies in the distillation setting.

- A second downside of using the C4 dataset is that we are limited in our ability to analyze downstream evaluations of students resulting from distillation. Our performance over standard English language downstream tasks closely follows cross-entropy, however, C4 is not as well suited for pretraining in order to probe aspects like reasoning performance (see Appendix E.1).

- We focused on distillation as originally defined in Hinton et al. (2015), where the teacher produces a full probability distribution for the student to target. We did this as it is a popular choice for training language models (Rivière et al., 2024; Gunter et al., 2024; Sreenivas et al., 2024). More colloquially, *distillation* has become used to describe the more general process of using a teacher in order to produce a student. One popular approach for training language models is *Sequence-Level Knowledge Distillation* (Kim & Rush, 2016) where the teacher is sampled, e.g. with beam search, in order to produce sequences for training the student on in a supervised way. This technique, also called *synthetic data* or *hard distillation* has been employed to great effect in the LLaMA families (Touvron et al., 2023a) and most recently, the smaller models distilled from DeepSeek-R1 (DeepSeek-AI et al., 2024). On top of these distillation methods are many variations of objectives, such as intermediate layer matching (Romero et al., 2015), modified objectives (Tian et al., 2020) and beyond. While we anticipate that our broader findings should apply in these cases, we cannot be absolutely sure. In particular, we suggest that verifying the scaling properties of Sequence-Level Knowledge Distillation in a controlled, resource constrained manner as we have done here is important for future study.

- Our work exclusively studies transformer style architectures, for both the teacher and student. While supervised cross-entropy is primarily influenced by model size and the amount of training data ((Kaplan et al., 2020)), it is plausible that architectural differences might affect model confidence or knowledge transfer in ways not fully captured by cross-entropy. Evidence for this effect was shown in Furlanello et al. (2018), although in a limited data setting where the teacher behaves as a regularizer *and* as a learning signal, significantly more complicated than our setting. Consequently, a study in *non-repeated data* on i) the influence of architectural *disparities*, and ii) of *non-transformer architectures*, could offer valuable insights.

- Our work exclusively investigates training and distilling on the *same data distribution*. This was done to allow us to isolate and study algorithmic effects, rather than effects from data. Unfortunately, this study design misses one typical distillation workflow, where a user chooses an openly available model trained by another group on a (possibly unknown) source distribution $p_{\text{source}}$, and then distills it on their own target distribution $p_{\text{target}}$. We suspect the following may occur. Consider the case that the teacher is well-trained, that is, $\hat{p}_T(y|x) \approx p_{\text{source}}(y|x)$. The student trained under Equation 7 should then approximate the teacher distribution, i.e. $\hat{q}_S(y|x) \approx \hat{p}_T(y|x) \approx p_{\text{source}}(y|x)$, that is, *on the intersection of the support* of $p_{\text{source}}(x)$ and $p_{\text{target}}(x)$, the student will learn to approximate the next-token distribution of the source domain, and *not* the target domain. Outside of this intersection, the teacher may behave out-of-domain and cease to provide *meaningful* signal for the student. Quantifying the scaling properties as a function of this teacher-student domain difference would be a valuable extension of our study.

- Our Distillation Scaling Law Equation 8 is not universal, that is, the coefficients we observe (Appendix F) are specific to our architecture and dataset choices and are not guaranteed to generalize to other architectures and datasets. Further, although the form of our scaling law has many desired limiting behaviors, it is not derived from first principles, as in e.g. Paquette et al. (2024). As such, we cannot fully guarantee the correctness of the law, and suggest that a formal derivation of the scaling law as valuable future work.

# B. Extended background

## B.1. Knowledge Distillation

Bucila et al. (2006) provided strong evidence that the knowledge gained by a large ensemble of models can be effectively transferred to a single smaller model. Later, Hinton et al. (2015) introduced knowledge distillation, where a smaller *student* network learns from a larger *teacher* network by mimicking its softened output probabilities, improving efficiency and generalization. Building on this, Stanton et al. (2021) studied both fidelity and student generalization, showing that while knowledge distillation often improves generalization, it frequently fails to achieve high fidelity, as student models do not fully match the teacher's predictive distribution. We study fidelity in terms of calibration in Appendix E.8, and show that when the learning signal is consistent with the calibration measure, then the student in our setup is well-calibrated both with respect to the teacher and the actual data. Addressing this, Beyer et al. (2022) demonstrated that knowledge distillation is most effective when the teacher is patient and consistent, providing stable targets over prolonged training to improve student generalization and fidelity. Our Language Model (LM) setup automatically satisfies *consistency*: both the teacher and student see the same data during the student's training. However, our conclusions differ from those of Beyer et al. (2022) in that although distilling a student for longer does improve its performance, unless the teacher is chosen perfectly, distillation becomes less effective than supervised learning in the *patient* setting, see Appendix D.2 for a discussion. Beyond empirical insights, Menon et al. (2020) established a bias-variance tradeoff for the student, quantifying how access to teacher logits can significantly enhance learning. Meanwhile, Pareek et al. (2024) investigated self-distillation, where the student and teacher share the same architecture and size, to assess the potential gains from repeatedly applying knowledge distillation. While most studies assume the teacher is a larger model, recent work explores weak-to-strong generalization, where a weaker model distills knowledge into a stronger one. This concept, introduced by Burns et al. (2024) and studied in LMs, was further analyzed by Ildiz et al. (2024), who extended the theoretical analysis to high-dimensional data and over-parameterized regression. Their findings show that distillation can provably outperform training with strong labels under the same data budget but does not improve the data scaling law. Our distillation scaling law (Equation 8) confirms this finding, which for a fixed teacher cross-entropy does not improve the scaling law compared to the supervised one in Equation 1. Moreover, in many previous works, distillation happens with repeated data, that is, the student sees the same data as the teacher does during its training. In our setup, we do not repeat the data between teacher training and distillation, which allows us to examine only the effect of distillation rather than the possible diminishing returns of repeated data; see Muennighoff et al. (2023a) for more details on the effect of repeating data.

## B.2. Neural Scaling Laws

Predictable scaling trends in neural networks were first empirically observed by Hestness et al. (2017) and later by Kaplan et al. (2020) who established empirical scaling laws for language model performance based on cross-entropy, which led to Hoffmann et al. (2022) and the pursuit of compute-optimal training. Beyond the empirical studies, there have been many theoretical works which provide explanations for why scaling laws should exist (Bahri et al., 2021; Paquette et al., 2024; Havrilla & Liao, 2024). More recent works explore scaling laws across different distributions, closely related to knowledge distillation. Hernandez et al. (2021) derived a scaling law for transfer learning, analyzing effective data transfer in low-data regimes and diminishing returns in high-data regimes. Similarly, Barnett (2024) empirically studied pretraining on one distribution for optimizing downstream performance on another, showing that when the *transfer gap* is low, pretraining is a cost-effective strategy. Finally, Jain et al. (2024) theoretically analyze how additional data from a *surrogate model* affects generalization, demonstrating that surrogate data can reduce test error—even when unrelated—due to Stein's paradox (Stein, 1956), with test error following a scaling law. This setup is related to tuning the coefficient $\lambda$ in our case, where we also observe a U-shape behavior depending on the teacher and student sizes (see Figure 51a). However, we are interested in studying the effect of distillation *only* ($\lambda = 1.0$), which differs from their setup. While these works are closely related to knowledge distillation—since one can compare the distribution of the teacher logits to that of the student—they do not establish a distillation scaling law. Moreover, their setup differs from practical knowledge distillation, as it does not involve training a *new* student model using a teacher but instead studies the effect of transferring training knowledge to a downstream task. Our work is the first to determine and verify a distillation scaling law and examine the regions where one should distill as well as the regions where supervised pretraining outperforms distillation; see Figures 6, 7, and 14 in Appendix D.2 and Section 5.2. Finally, for improving inference cost at a given model capability, the scaling behavior of Mixture of Experts (MoE) (Shazeer et al., 2017; Jelassi et al., 2024) have been investigated in the context of scaling laws (Clark et al., 2022; Ludziejewski et al., 2024; Abnar et al., 2025) as one alternative to knowledge distillation.

## B.3. The Knowledge Distillation Capacity Gap

Despite extensive research on knowledge distillation, a persistent challenge is the curse of capacity gap, where a larger teacher does not necessarily produce a superior student compared to a smaller teacher. This occurs because a large gap in model capacity makes it harder for the student to effectively learn from the teacher's outputs. As a result, there exists an optimal teacher size along the scaling trajectory that maximizes student performance. Our distillation scaling law in Equation 8 confirms this, revealing a u-shaped trend in the scaling law and validating the existence of an optimal teacher. However, our results further indicate that the capacity gap is influenced not only by the size of the teacher but also by its training tokens and, more generally, its loss. A theoretical analysis in the kernel regression setup (Appendix C) supports these findings. Lukasik et al. (2022) showed that distillation gains are not uniform and can even degrade performance when small teacher errors are amplified by the student. Similarly, Nagarajan et al. (2023) found that deviations in predictive probabilities cause students to exaggerate the teacher's confidence levels. Several works (Peng et al., 2024; Zhang et al., 2023a; Rawat et al., 2024) observed the capacity gap in pre-training distillation for Large Language Model (LLM)s, affecting both large-to-small and small-to-large distillation. Notably, Zhang et al. (2023a) proposed an empirical law of the capacity gap, showing that the optimal teacher scale follows an approximately linear relationship with the student's scale. However, our findings suggest that scaling alone is insufficient—one must account for the complexity of the effective hypothesis space (Equation 8) and we show that Zhang et al. (2023a) is a special case of our work when the teachers are compute-optimal from a supervised perspective (see Section 5.3). To address this issue, various strategies have been explored. Yuan et al. (2024) studied temperature scaling, which simplifies the teacher's output into more learnable representations, aiding student generalization. We analyzed the effect of temperature and learning rate in distillation (Figures 52 and 53) and found that, contrary to existing literature, the optimal temperature is one. We hypothesize that this discrepancy arises because previous studies used repeated tokens, whereas our setup does not involve repeated data. Additionally, Cho & Hariharan (2019) found that early stopping of the teacher's training mitigates the capacity gap, while Mirzadeh et al. (2020) proposed progressive distillation, where knowledge is transferred through intermediate models to improve student learning. Further, Fan et al. (2024) looked at the effect of knowledge distillation from distributional differences using calibration, and found that teacher miscalibration is a primary source of poor student performance and a capacity gap. We study calibration in Appendix E.8 and show that our teachers are *well-calibrated*, and that poor calibration cannot be the *only* source of the capacity gap. (Lee et al., 2022) focuses on the calibration of the student rather than teacher, and develop a modified training procedure that swaps between teacher and data supervision, improving student generalization. Amara et al. (2022) investigated further modifications of the objective, using a sample-wise adaptive balance between forward and reverse KL divergence, reducing Expected Calibration Error (ECE) and reducing the capacity gap.

From a theoretical perspective, Harutyunyan et al. (2023) analyzed the capacity gap in distillation using supervision complexity in kernel classifiers. Their findings highlight a trade-off between teacher accuracy, student margin with respect to teacher predictions, and teacher complexity, explaining why some teachers are easier for the student to learn. Earlier, Lopez-Paz et al. (2016) studied generalization error in distillation, proving that learning from a teacher can be beneficial under certain conditions, particularly when the teacher's capacity is small. Using similar techniques in LMs, Zhang et al. (2023b) demonstrated that among students of different capacities distilled from the same teacher, smaller students suffer from higher generalization error and lower performance, while larger teachers provide lower generalization error, reinforcing the trade-off in teacher-student capacity. Our distillation scaling law (Equation 8) also confirms this trend, and we observe the effect of capacity gap in our scaling law terms, see Section 4.3 for more details.

Foundation models were initially undertrained (Brown et al., 2020), then followed the compute-optimal scaling law carefully (Hoffmann et al., 2022; Pearce & Song, 2024; Besiroglu et al., 2024), and soon after started overtraining heavily (Sardana et al., 2024; Bi et al., 2024; Hu et al., 2024; Mesnard et al., 2024; Jiang et al., 2023). The LLaMA family (Touvron et al., 2023a;b; Dubey et al., 2024) and Phi line (Li et al., 2023; Abdin et al., 2024b;a) is following the same trend, where smaller models are overtrained according to the original Chinchilla scaling laws. In all these cases, the models are designed to be best possible foundation model that is still cheap and fast to run on lower end hardware. Besides overtraining, more recently, smaller foundation models tend to be distilled from larger models (Gunter et al., 2024; Rivière et al., 2024; Reid et al., 2024) to further increase performance. In these cases, the large model either specifically trained with the sole purpose of being a distillation teacher, or an existing model is re-used. In both these cases, there are no reports of how the exact teacher size is decided when taking total compute into account. Determining the optimal allocation of compute in distillation is one of the primary contributions of our work (see Section 5.3).

## C. Teacher Student Capacity Gaps

In this section, we examine the capacity gap in two settings: kernel regression and a synthetic example using Multi-Layer Perceptron (MLP) for a mapping problem. The kernel regression setup provides a theoretical and analytically tractable perspective on the capacity gap. The MLP-based synthetic example allows us to study the capacity gap in a more practical, learnable function approximation scenario. By analyzing these two setups, we aim to better understand the fundamental limitations of distillation when there is a significant mismatch between teacher and student capacities.

### C.1. Kernel Regression

One of our main contributions is that the student loss follows a broken power law, where the transition between the two power law regions occur when the student becomes a stronger learner than the teacher (Equation 8). This implies that making the teacher too capable (relative to the student) reduces student performance. In this section we show how a capacity gap provably degrades student performance in the setting of kernel regression. While simple, we believe the underlying principle causing the student performance degradation in this case carry over to much more general settings involving neural networks.

#### C.1.1. SETUP

Let $\mathcal{H}$ denote a Hilbert space spanned by orthonormal bases functions $\{\phi_i\}_{i=1}^{\infty}$ such that $\langle \phi_i, \phi_j \rangle_{\mathcal{H}} = \delta_{ij}$. Let $f^* \in \mathcal{H}$ denote the *target function*, identified by a set of coefficients $\boldsymbol{\alpha} = \{\alpha_i\}_{i=1}^{\infty} \in \mathbb{R}$, $\|\alpha\| = M < \infty$ such that:

$$f^{\star}(x) = \sum_{i=1}^{\infty} \alpha_i \phi_i(x). \tag{11}$$

Let $\mathcal{H}_t^m, \mathcal{H}_s^n$ denote the teacher and student Hilbert spaces respectively:

$$\mathcal{H}_t^m = \text{Span}\{\phi_1, \phi_2, ..., \phi_m\}, \tag{12}$$

$$\mathcal{H}_s^n = \text{Span}\{\phi_1, \phi_2, ..., \phi_n\}, \tag{13}$$

which are the hypothesis spaces of the teacher and student. Note that while the Hilbert space $\mathcal{H}$ is spanned by an infinite orthonormal basis, the teacher and student spaces are *finite* and spanned by $m$ and $n$ basis functions respectively, where $|m - n|$ represents the teacher and student capacity gap.

The process of training the teacher and student models involves solving the following constrained optimization problems:

$$g^{\star} = \min_{g \in \mathcal{H}_t^m} \|g - f^{\star}\|_{\mathcal{H}} \quad \text{s.t} \quad \|g\|_{\mathcal{H}} \leq T, \tag{14}$$

$$h^{\star} = \min_{h \in \mathcal{H}_s^n} \|h - g^{\star}\|_{\mathcal{H}} \quad \text{s.t} \quad \|h\|_{\mathcal{H}} \leq D, \tag{15}$$

where $g^{\star}, h^{\star}$ are the optimal teacher and student respectively, and $D \leq T < M$. Note that we assume the teacher and student are exposed to an infinite amount of training data, hence our analysis is carried over entirely in function space.

**Lemma C.1.** *The optimal teacher $g^{\star}$ is given by:*

$$g^{\star}(x) = C(m, T) \sum_{i=1}^{m} \alpha_i \phi_i(x), \quad C(m, T) = \begin{cases} 1 & \sqrt{\sum_{i=1}^{m} \alpha_i^2} \leq T \\ \frac{T}{\sqrt{\sum_{i=1}^{m} \alpha_i^2}} & \textit{otherwise.} \end{cases} \tag{16}$$

*The teacher error $e_{teacher}^{\star}(m, T)$ is given by:*

$$e_{teacher}^{\star}(m, T) = \|g^{\star} - f^{\star}\|_{\mathcal{H}} = \sqrt{(C(m, T) - 1)^2 \sum_{i=1}^{m} \alpha_i^2 + \sum_{i=m+1}^{\infty} \alpha_i^2}. \tag{17}$$

*Proof.* By construction we may assume the teacher model takes the form $g^{\star} = \sum_{i=1}^{m} \beta_i \phi_i$. where $\sqrt{\sum_{i=1}^{m} \beta_i^2} \leq T$. We can write the error of $g^{\star}$ using:

$$e_{\text{teacher}}(m, T, \boldsymbol{\beta}) = \left\| \left( \sum_{i=1}^{m} (\beta_i - \alpha_i)\phi_i + \sum_{i=m+1}^{\infty} \alpha_i \phi_i \right\|_{\mathcal{H}} = \sqrt{\sum_{i=1}^{m} (\beta_i - \alpha_i)^2 + \sum_{i=m+1}^{\infty} \alpha_i^2}. \tag{18}$$

Note that the minimizing coefficients $\boldsymbol{\beta}^\star$ of Equation 18 must take the form $\boldsymbol{\beta} = C\boldsymbol{\alpha}$ for some coefficient $C$. Considering the norm constraint on $g$, the constant $C$ takes the form in Equation 16. Plugging the resulting $g^\star$ into the expression for $e_{\text{teacher}}(m, T, \boldsymbol{\beta}^\star)$ completes the proof. $\qquad\square$

Notably and intuitively, teacher error decreases monotonically as $m$, representing the teacher model capacity, increases.

### C.1.2. DISTILLING THE TEACHER

We now pick our student function $h^\star$ by mimicking the teacher subject to a norm constraint:

$$h^\star(x) = \min_{h \in \mathcal{H}_t^n} \|h - g^\star\|_{\mathcal{H}} \quad \text{s.t.} \quad \|h\|_{\mathcal{H}} \le D. \tag{19}$$

**Lemma C.2.** *Let $k = \min(m, n)$ be the smaller of the teacher and student capacities. The optimal student $h^\star$ is given by:*

$$h^\star = Q(m, k, T, D)C(m, T)\sum_{i=1}^{k} \alpha_i \phi_i \tag{20}$$

$$Q(m, k, T, D) = \begin{cases} 1 & C(m,T)\sqrt{\sum_{i=1}^{k}\alpha_i^2} < D \\ \frac{D}{C(m,T)\sqrt{\sum_{i=1}^{k}\alpha_i^2}} & \text{otherwise.} \end{cases} \tag{21}$$

*The student error with respect to the target function is then:*

$$e_{student}(m, n, T, D) = \|h^\star - f^\star\|_{\mathcal{H}} = \sqrt{(C(m,T)Q(m,k,T,D) - 1)^2 \sum_{i=1}^{k}\alpha_i^2 + \sum_{i=k+1}^{\infty}\alpha_i^2} \tag{22}$$

*Proof.* The proof follows the exact same logic as in Lemma C.1. i.e, we can assume the optimal student is given by $h^\star = \sum_{i=1}^{n} \gamma_i \phi_i$. From the distillation loss, the optimal coefficients must match the teacher coefficients for the basis functions $\{\phi_i\}_{i=1}^n$, perhaps rescaled due to the norm constraint $\sqrt{\sum_{i=1}^{n}\gamma_i^2} \le D$. This rescaling then gives rise to the additional $Q(m, k, T, D)$ multiplier in Equation 21. $\qquad\square$

### C.1.3. U-SHAPE IN THE STUDENT ERROR

We will prove that the map

$$m \longmapsto e_{\text{student}}(m, n, T, D)$$

is comprised of two distinct segments: i) where the student error monotonically decreases for $m < n$, and ii) where it monotonically increases for $m \ge n$, establishing a U-shape in the student error echoing the trend seen in Figures 3 and 4.

**Case 1: $m < n$. (Student error is non-increasing in $m$)**

*Claim.* For $1 \le m < n$, we have

$$e_{\text{student}}(m + 1, n, T, D) \quad \le \quad e_{\text{student}}(m, n, T, D).$$

In words, when $m < n$, the error does not increase (and typically decreases) as the teacher capacity $m$ increases.

*Proof.*

Let $\mathcal{H}_t^{m,T} \subseteq \mathcal{H}_t^m$ denote the space of functions in $\mathcal{H}_t^m$ that are norm constrained by $D$. i.e:

$$\mathcal{H}_t^{m,T} = \{f \in \mathcal{H}_t^m \;:\; \|f\|_{\mathcal{H}} \le T\}. \tag{23}$$

Since $\mathcal{H}_t^{m,T} \subseteq \mathcal{H}_t^{m+1,T}$, it follows that $g_m^\star \in \mathcal{H}_t^{m+1,T}$, which implies that the teacher error cannot increase as $m$ increases, hence it monotonically decreases. Now, let $h_m^\star$ denote the optimal student given the teacher $g_m^\star$. Since $D \le T$, then for any $m < n$, we can equivalently write the optimal student $h_m^\star$ as the solution to the following optimization problem:

$$\forall_{m \le n} h_m^\star = \min_{h \in \mathcal{H}_s^n} \|h - g_m^\star\|_{\mathcal{H}} \quad \text{s.t} \quad \|h\|_{\mathcal{H}} \le D \tag{24}$$

$$= \min_{h \in \mathcal{H}_t^m} \|h - f^\star\|_{\mathcal{H}} \quad \text{s.t} \quad \|h\|_{\mathcal{H}} \le D, \tag{25}$$

which corresponds exactly to the objective of finding the optimal teacher with with a norm constraint set to $D$. Therefore, from the fact that the teacher error monotonically decreases we can conclude that the student error monotonically decreases as well in the regime $m < n$.

**Case 2: $m \geq n$. (Student error eventually increases in $m$)**

*Claim.* For $m \geq n$:

$$e_{\text{student}}(m+1, n, T, D) \quad \geq \quad e_{\text{student}}(m, n, T, D).$$

Hence once $m$ exceeds $n$ the student error cannot decrease any further, the error eventually starts to rise.

*Proof.*

Let $\boldsymbol{\beta}^\star_m = \{\beta_1, ..., \beta_m\}$ denote the coefficients of the optimal teacher $g^\star_m$. Note that in the regime $m \geq n$, as long as $\sqrt{\sum_{i=1}^{n} \beta_i^2} \geq D$ (i.e the norm of the coefficients corresponding to the basis $\{\phi_1, ..., \phi_n\}$ is smaller than $D$), we have from Equation 21 that $Q(m, k, T, D) = 1$, which means that the optimal student doesnt change, hence its error remains constant. If however $\sqrt{\sum_{i=1}^{n} \beta_i^2} < D$, then we have from Equation 21:

$$1 > Q(m, k, T, D) \geq Q(m+1, k, T, D), \tag{26}$$

where the second inequality becomes strict if $\alpha_{m+1}^2 > 0$. A strict inequality (i.e $Q(m, k, T, D) > Q(m+1, k, T, D)$) implies the optimal student is further scaled down due to the teacher having to "spread its capacity" to additional basis functions that are not learnable by the student, thereby strictly increasing its error. Hence for $m \geq n$, we get

$$e_{\text{student}}(m+1, n, T, D) \quad \geq \quad e_{\text{student}}(m, n, T, D),$$

demonstrating that the error increases monotonically with $m$ once $m \geq n$. $\qquad\square$

**Conclusion (U-shaped trend).** Combining these two cases:

$$\begin{cases} \text{For } 1 \leq m < n: & e_{\text{student}}(m, n, T, D) \text{ monotonically decreasing in } m, \\ \text{For } m \geq n: & e_{\text{student}}(m, n, T, D) \text{ monotonically increasing in } m. \end{cases}$$

Therefore, as a function of $m$, the student error $e_{\text{student}}(m, n, T, D)$ *first decreases* and *then increases* (for $m \geq n$) (for $m \leq n$), giving a u-shape in student error due to a capacity gap between the teacher and the student. $\qquad\square$

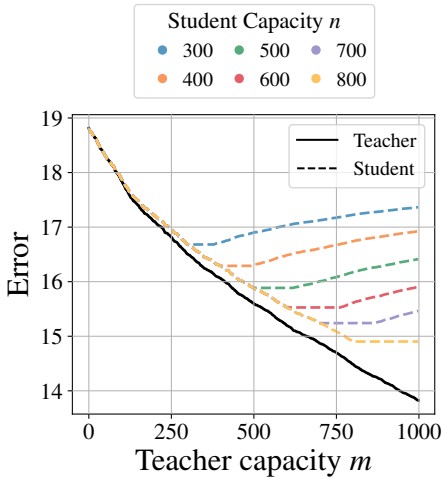

*Figure 10.* **Distillation in kernel regression.** We randomly sample the $\alpha = \{\alpha_1, ..., \alpha_{1000}\}$ coefficients of the target function uniformly in the range $[-1, 1]$. We fix $T = 5, D = 4.5$ and compute the optimal student and teacher errors according to Lemmas C.1 and C.2 for various values of $n$ (dashed curves), and for $m \in [1...1000]$. The student error exhibits a U shaped error curve as predicted, where the error starts to increase when $m \geq n$. The black solid line indicates the teacher error, which always decreases with increasing $m$.

We present an empirical verification of these conclusions in Figure 10.

The above theoretical analysis points to an intuitive interpretation of the potentially adverse effect of a large teacher-student capacity gap; the degradation in student performance is due to the teacher learning basis functions that are unreachable by the student, at the expense of basis functions that are reachable by the student. In the following we provide empirical evidence in support of this picture in a controlled yet more realistic setting.

### C.2. MLPs on the Mapping Problem

#### C.2.1. PROBLEM DEFINITION

Here we show a synthetic setting which exhibits the U-shape phenomenon. Matching the kernel regression analysis (Appendix C.1), we find that the synthetic problem must include a class of problems that are easy for the student to learn, and ones that are harder, in order for the U-shape to appear.

The problem setting is the *Mapping Problem*, and is similar in spirit to Pointer Value Retrieval (Zhang et al., 2021), Here, the input is composed of small integers in {0,1,2}. The label for each sample is given by the code below, which shows the two cases: i) one where the label is simply given by a one-hot position, and ii) one where the label is given by the location of a matching element in the context portion of the input.

```
def find(vector, value):
    """Find locations of value in vector."""
    return np.where(vector == value)[0]

def remove(vector, value):
    """Find value from vector."""
    return np.delete(vector, find(vector, value))

def label(vector: np.ndarray, num_classes: int) -> np.ndarray:
    """Return the label in [0, num_classes) for vector."""
    assert len(vector) == 2 * num_classes
    one_hot = vector[num_classes:]
    context = vector[:num_classes]
    i = find(one_hot, 1)
    if context[i] == 0:
        return i
    else:    # remapping
        c = context[i]
        return remove(find(context, c), i)
```

```
Examples:
----------------------------
2020210001000000, label = 1
    context [2 0 2 0 2 1 0 0]
    one-hot [0 1 0 0 0 0 0 0]
----------------------------
1210120000000100, label = 2
    context [1 1 2 0 1 2 0 0]
    one-hot [0 0 0 0 0 1 0 0]
----------------------------
0122221201000000, label = 6
    context [0 1 2 2 2 2 1 2]
    one-hot [0 1 0 0 0 0 0 0]
----------------------------
```

#### C.2.2. EXPERIMENTAL FINDINGS

We train MLPs with two hidden layers of equal width, all non-linearities are Rectified Linear Units (ReLUs). Teachers and students of different sizes are produced by varying the hidden layer width only.

All model are trained with Adam (Kingma & Ba, 2015) using a peak learning rate of $3 \times 10^{-4}$, a single cycle cosine learning rate schedule with a linear warmup of $5\%$ of the total training steps. A batch size of 512 is used for all models. Training samples are never repeated. Unless explicitly stated, model are trained on $500 \times 512$, or $20N$ samples, where $N$ is the number of model parameters, whichever is larger.

In Figure 11, we look at varying the size of the teacher. For the width 256 model, student performance improves as the teacher size increases to a point, and then student performance worsens. This is observable in both the student cross-entropy (Figure 11a) and accuracy (Figure 11b). Aligning with theory and large-scale experiments, the student cannot learn if it is too small, and learns to match the teacher model when the student is large enough. In the intermediate regime, where distillation is often used, we see an optimal teacher size and a capacity gap phenomenon.

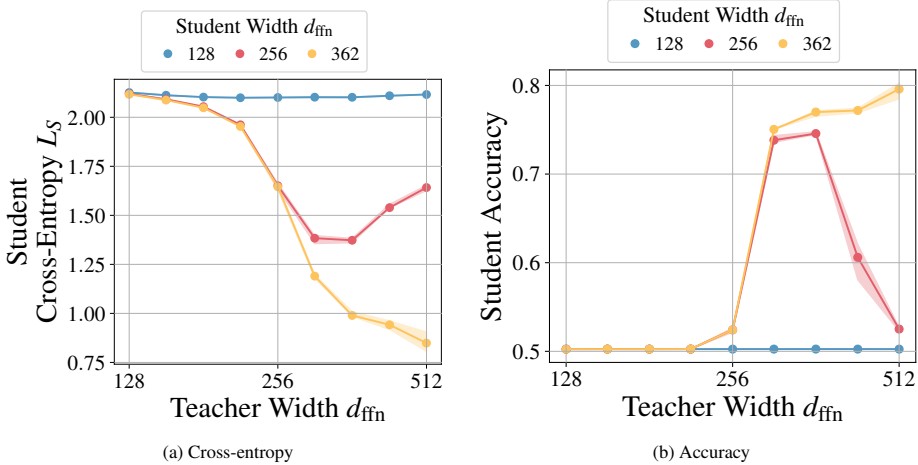

(a) Cross-entropy

(b) Accuracy

*Figure 11.* **Student performance when varying teacher width.** **(a)** Student cross-entropy as teacher width $d_{\text{ffn}}$ is varied. **(b)** Student accuracy as teacher width $d_{\text{ffn}}$ is varied. Bands show the (25%,75%) values across four trials.

In Figure 12, a similar effect can be seen, when a large teacher ($d_{\text{ffn}} = 512$) is trained with on different amounts of data. This observation aligns with the idea that it is the teacher's completeness in modeling the problem that eventually harms the performance of a student with lesser capacity, and *not* only the teacher size.

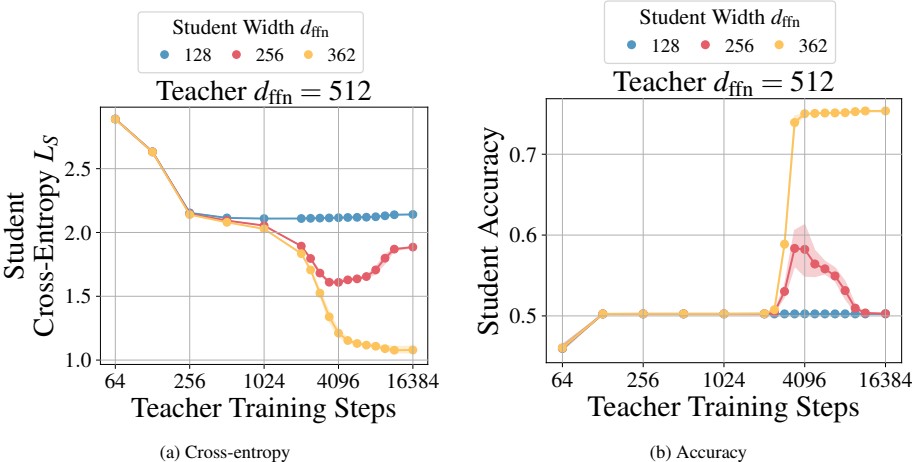

(a) Cross-entropy

(b) Accuracy

*Figure 12.* **Student performance when varying teacher training data.** **(a)** Student cross-entropy as teacher training data is varied. **(b)** Student accuracy as teacher training data is is varied. Bands show the (25%,75%) values across four trials.

## D. Distillation scaling law applications (additional results)

In this section, we present results referenced in Section 5. We explore the best-case scenario for distillation under fixed student tokens or compute, as well as under fixed teacher size or compute, while accounting for teacher inference. These results provide further insights into the optimal distillation strategies in different resource-constrained settings.

### D.1. Experimental differences resolving the apparent contradiction with patient teachers

Beyer et al. (2022) showed in computer vision that a good teacher is:

1. *Patient:* Distillation works best when training for a large number of epochs, and

2. *Consistent:* The teacher and the student see the *same views* of the data under an augmentation policy.

Our setting automatically satisfies *consistency* as there is no augmentation policy. There is a remaining question about

patience, which in our scenario corresponds to the large $D_S$ limit. We observe that for a given student size:

1. If the teacher is optimally chosen for the student, distilling on a large number of tokens produces the same result as training the model in a supervised way on the same number of tokens (Appendix E.6).

2. Otherwise supervised learning outperforms distillation (Section 5.3).

The second statement implies that the student should not be trained for *too long*, appearing to contradict patient teachers.

To resolve the contradiction, first we note that the modes in Beyer et al. (2022) are trained on a large, diverse dataset, e.g. ImageNet21k (Kolesnikov et al., 2020) and then fine-tuned on target datasets (e.g. Flowers102 (Nilsback & Zisserman, 2008), or ImageNet1k (Deng et al., 2009)). Students are distilled on the target datasets and only access the teacher's training distribution indirectly, i.e.

1. The students in Beyer et al. (2022) do not see the teacher training distribution directly, whereas ours do.

2. There is *no supervised baseline* where a supervised model has access to both ImageNet21k *and* the target dataset.

The absence of a supervised baseline means that Beyer et al. (2022) were unable to observe the point at which supervised learning becomes preferred to distillation as a function of compute or training data. This was not the focus of their work.

In our setting, we do have a supervised baseline, and see that at some amount of compute, supervised learning becomes more efficient than (or equally efficient as) distillation, leading us to upper-bound the length one should distill for. We also do see that distilling for longer improves the distilled model performance, i.e. patient teaching *does work*. However, we additionally note that patient teaching can be compute-suboptimal compared to supervised learning, depending on the specific setting (see Appendix D.4).

Additional differences in our experimental setups beyond the ones mentioned above, are summarized in Table 4.

*Table 4.* Experimental setting differences between Beyer et al. (2022) and ours.

| Component | Beyer et al. (2022) | Ours |
|---|---|---|
| Data repetitions | Many repetitions | Minimal repetitions |
| Data diversity | Low number of unique tokens | Large number of unique tokens |
| Domain | Vision | Language |
| Objective | Fewer categories, more unimodal | Many categories, highly multimodal |
| Architecture | Different computer vision architectures | Maximal Update Parameterization ($\mu$P) optimized homogeneous transformers |

### D.2. Fixed tokens or compute (best case)

**Distillation can outperform supervised learning given enough teacher training tokens or compute.** As shown in Figures 13a and 13b, when the teacher size, student size, and number of student tokens are held constant, increasing the number of teacher training tokens makes distillation more favorable than supervised learning. This advantage arises because the teacher, with access to more training tokens, can better learn the approximation of the language distribution. As a result, the teacher's learned distribution become more informative for the student to follow, thus improving the student's performance. Note that for a fixed student size and compute, the teacher must be sufficiently large and well-trained; otherwise, supervised learning will outperform distillation. Without adequate teacher size or training, the student may not benefit from the distillation process, leading to inferior performance compared to direct supervised learning.

We also see that the scatter data matches up well with the contour colors, despite these contour being a difference of two scaling laws, providing a verification of our setup.

**Supervised learning always outperforms distillation given enough student compute or tokens.** The trend observed in Figure 14 mirrors that of Section 5.1. It demonstrates that, for a fixed teacher size and compute, supervised learning can outperform distillation when the student's compute is sufficiently large. With enough resources allocated to the student, it can learn more effectively from the data directly, making distillation less advantageous in comparison. This advantage only happens at a compute budget that grows with student size.

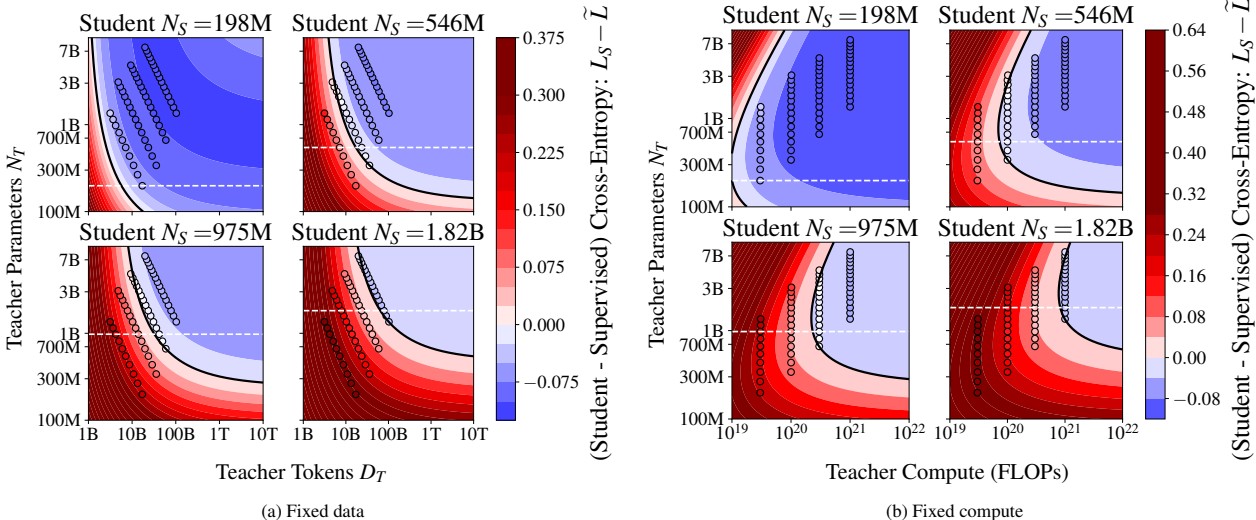

(a) Fixed data

(b) Fixed compute

*Figure 13.* **IsoFLOP Teacher Contours with Fixed $M$ students. (a)** For a given teacher size $N_T$, for a given teacher token $D_T$, what is the difference between the loss achieved by distillation and supervised learning. Blue indicates distillation outperforms supervised learning, and red indicates when supervised learning outperforms distillation. The white horizontal dashed line indicates the student size. **(b)** For a given teacher size $N_S$, for a given teacher compute budget, what is the difference between the loss achieved by distillation and supervised learning. Blue indicates distillation outperforms supervised learning, and red indicates when supervised learning outperforms distillation. The white horizontal dashed line indicates the student size.

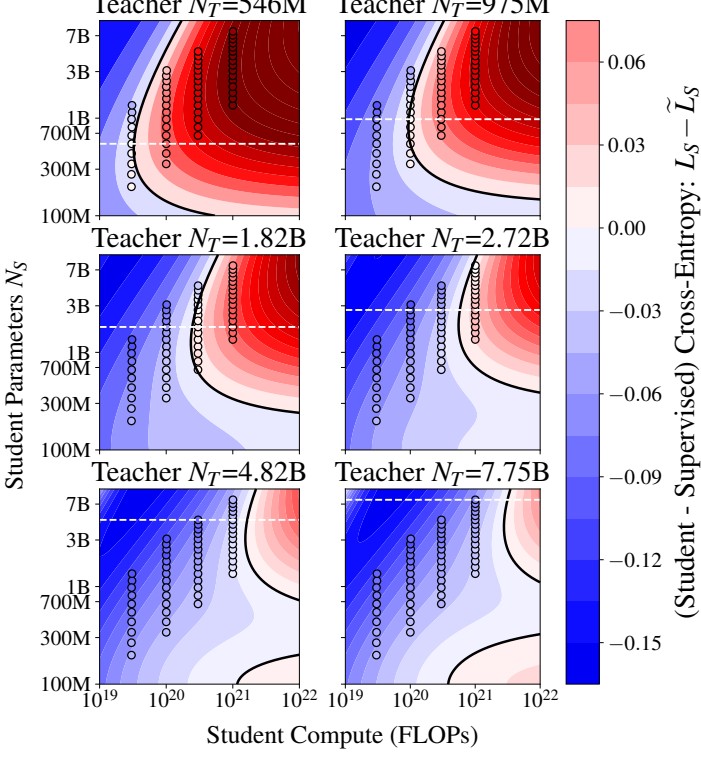

*Figure 14.* **Fixed $M$ Teacher Contours with IsoFLOP students (compute).** For a given student size and student compute budget, the difference between the loss achieved by distillation and supervised learning. Blue indicates distillation outperforms supervised learning, and red indicates when supervised learning outperforms distillation. The white horizontal dashed line indicates the teacher size.

### D.3. Fixed size or compute (teacher inference)

**Fixed student size** For a fixed student size, as the number of student tokens increases, the optimal teacher cross-entropy decreases slightly; see Figure 15. This observation highlights an asymmetry between the growth of student size and student tokens (or their rates in the scaling law), as the behavior here differs from that observed in Section 5.1. Notably, when the student size is sufficiently large, such as $N_S = 30B$, increasing the student tokens initially leads to a decrease in the teacher's loss, followed by a saturation point and a slow decrease in the optimal teacher's loss.

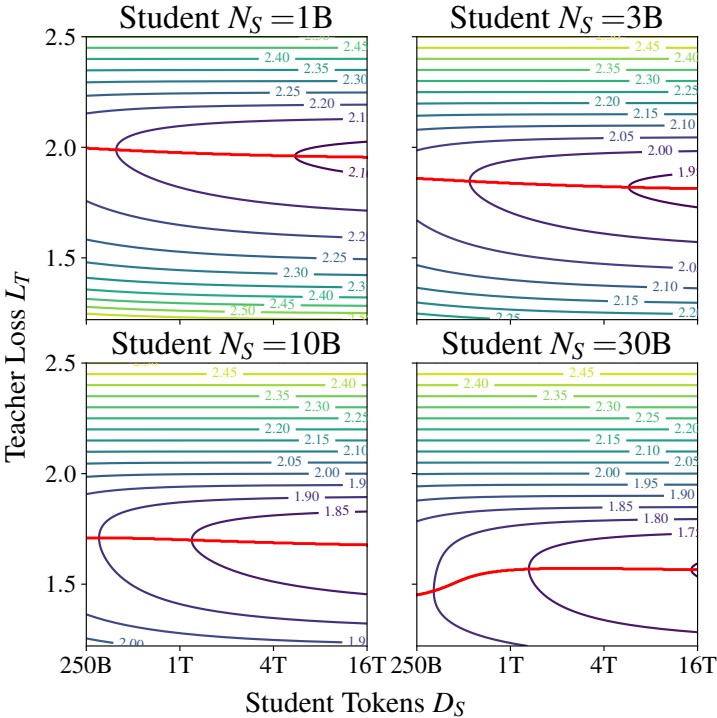

*Figure 15.* **Student performance given a teacher varying distillation tokens.** For four distillation student sizes $N_S \in \{1B, 3B, 10B, 30B\}$ the validation loss achieved by a students distilled on $D_S \in [250B, 16T]$ tokens under a teacher with loss $L_T \in [E, 2.5]$. The red line indicates the value of the teacher loss resulting in the best performing student, and the vertical dashed line indicates the number of tokens at which supervised pretraining outperforms distillation.

**Fixed compute budget** Given an inference budget $N_S$, a set of teachers $\{(L_T^{(i)}, N_T^{(i)})\}_{i=1}^n$ and a total compute budget $C_{\text{Total}}$, the number of distillation tokens is determined from Equation 9

$$D_S = C_{\text{Total}}/(3F(N_S) + \delta_{\text{T}-\text{Logits}}F(N_T)), \tag{27}$$

where $F(N)$ is the forward Floating Operations (FLOPs) per token of a model of size $N$ (see Appendix H). If $\delta_{\text{T}-\text{Logits}} = 0$ then there is no price to pay for a larger teacher, and the conclusions are identical to those of the fixed token analysis of Section 5.2. In the worst case scenario, $\delta_{\text{T}-\text{Logits}} = 1$, then using a larger teacher will mean fewer distillation tokens are available for the student. Due to the capacity gap phenomenon, at small compute budgets, this means it is actually better to use a *large weak teacher* rather than a *large strong teacher*. Once compute is sufficient to allow enough distillation tokens, a stronger teacher can be used for all student sizes (see Figure 16).

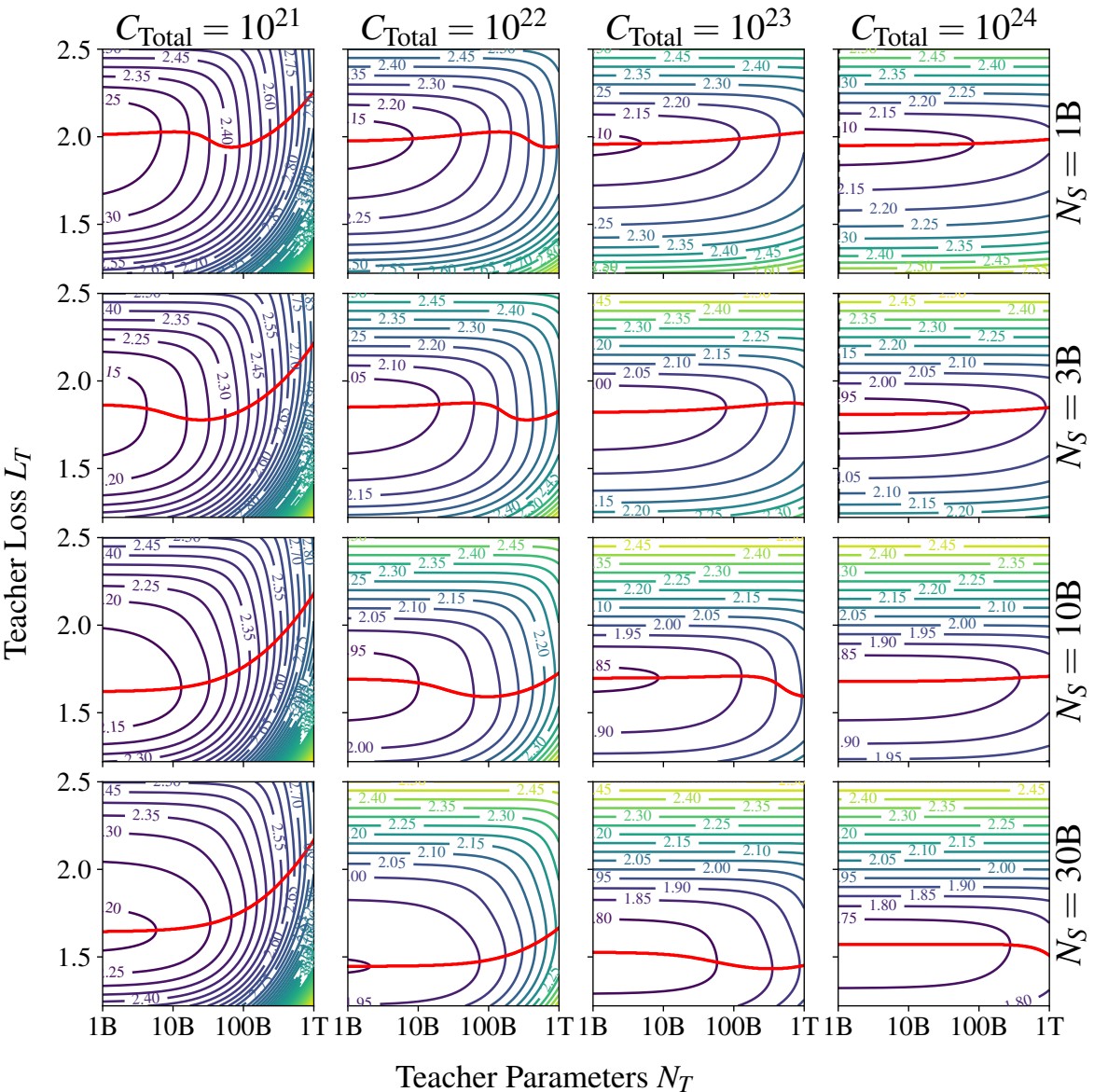

*Figure 16.* **Fixed compute distillation strategy.** The student performance obtained for four total compute budgets $C_{\text{Total}} \in \{10^{21}, 10^{22}, 10^{23}, 10^{24}\}$ FLOPs and four student sizes $N_S \in \{1B, 3B, 10B, 30B\}$ under a teacher of size $N_T \in [1B, 1T]$ and teacher loss $L_T \in [E, 2.5]$. The red line indicates the value of teacher loss $L_T^*(N_T)$ that results in the best student performance for each teacher size $N_T$.

*Table 5.* Scenarios considered in our scaling law applications. Same as Table 2.

| Compute Scenario | $\delta_T^{\text{Lgt}}$ | $\delta_T^{\text{Pre}}$ | Description |
|---|---|---|---|
| Best case (fully amortized teacher) | 0 | 0 | The teacher produces no additional FLOPs and so we are free to choose the teacher $L_T^*$ that minimizes the student cross-entropy. |
| Teacher inference | 1 | 0 | We don't account for the teacher cost because the teacher already exists, or we intend to use the teacher as e.g. a server model. We still need to pay to use it for distilling a student. |
| Teacher pretraining | 0 | 1 | The teacher needs training, but we store the logits for re-use, either during training, or after training for distilling into sufficiently many students. |
| Teacher pretraining + inference | 1 | 1 | The teacher needs training and we pay for distilling into one student, the worst case scenario. |

## D.4. Compute optimal distillation

### D.4.1. SETUP

The solutions resulting in the losses give guidance on how to scale depending on the use case, and are the result of constrained optimization

$$D_S^*, N_T^*, D_T^* = \underset{D_S, N_T, D_T}{\arg\min} \ L_S(N_S, D_S, N_T, D_T) \qquad \text{s.t.} \qquad \text{FLOPs}(N_S, D_S, N_T, D_T) = C, \tag{28}$$

where $L_S(N_S, D_S, N_T, D_T)$ is the distillation scaling law (Equation 8), and

$$\text{FLOPs}(N_S, D_S, N_T, D_T) \approx \underbrace{3F(N_S)D_S}_{\substack{\text{Student} \\ \text{Training}}} + F(N_T)(\underbrace{\delta_T^{\text{Lgt}} D_S}_{\substack{\text{Teacher} \\ \text{Logits}}} + \underbrace{\delta_T^{\text{Pre}} 3D_T}_{\substack{\text{Teacher} \\ \text{Training}}}) \tag{29}$$

is the total number of floating operations performed in the entire distillation setup. $F(N)$ is the forward FLOPs per token of a model of size $N$ (see Appendix H), and $\delta_T^{\text{Lgt}}, \delta_T^{\text{Pre}} \in [0, 1]$ indicate if we account for the cost of teacher logit inference for the student targets and teacher pretraining cost in the total compute budget. For convenience, we restate our compute scenarios of interest in Table 5). Constrained numerical minimization using Sequential Least SQuares Programming (SLSQP) (Kraft, 1988) in SciPy (Virtanen et al., 2019). We allow numerical solutions for model sizes and tokens $N_T, D_S, D_T \in [1M, 100P]$. While this token upper-limit is larger than available resources (Epoch AI, 2023), it simplifies discussions when comparing to supervised learning at large compute budgets, which otherwise, for smaller students, would only by using a fraction of the available compute.

We begin by looking at the student cross-entropy achievable in each compute scenarios alongside the corresponding teacher cross-entropies in Appendix D.4.2. We then investigate the compute-optimal distillation configurations for each scenario that produce those cross-entropies. We look at *best case* distillation in Appendix D.4.3, *teacher inference* in Appendix D.4.4, *teacher pretraining* in Appendix D.4.5, and *teacher pretraining + inference* in Appendix D.4.6. Finally, to aid comparisons across methods, we present the token and parameter configurations for all methods in Appendix D.4.7 and Appendix D.4.8 respectively. For completeness, in the following sections, some of the findings of Section 5.3 are restated.

### D.4.2. CROSS-ENTROPY

In Figure 17 we show the student cross-entropies achieved in the compute optimal case for each scenario in Table 5, and the teacher cross-entropies that enable those student cross-entropies in Figure 18.

**Distillation and supervised learning produce the same student at large compute.** The first thing to note in Figure 17 is that at low compute, in the *best case* and *teacher inference* scenarios, distillation outperforms supervised learning, consistent with our expectations from distillation and the existing literature (see Appendix B.1). However, once enough the compute is *large enough*[6], distillation and supervised learning produce models with the same cross-entropy, i.e. in general,

---

[6]The level of compute at which this happens is larger for larger models, see Figure 17 for specific values.

*distillation does not allow us to produce better models that supervised learning does*, however, *distillation does produce better models than supervised learning with modest resources*. This behavior is consistent with the asymptotic analysis in Appendix E.6, and can be understood through noting that although distillation modifies the learning process the student undergoes, *distillation does not alter the hypothesis space of the student*, which is tied to the student size $N_S$, is the same hypothesis space in the supervised and distillation settings, and can be explored in the limit of infinite compute or data.

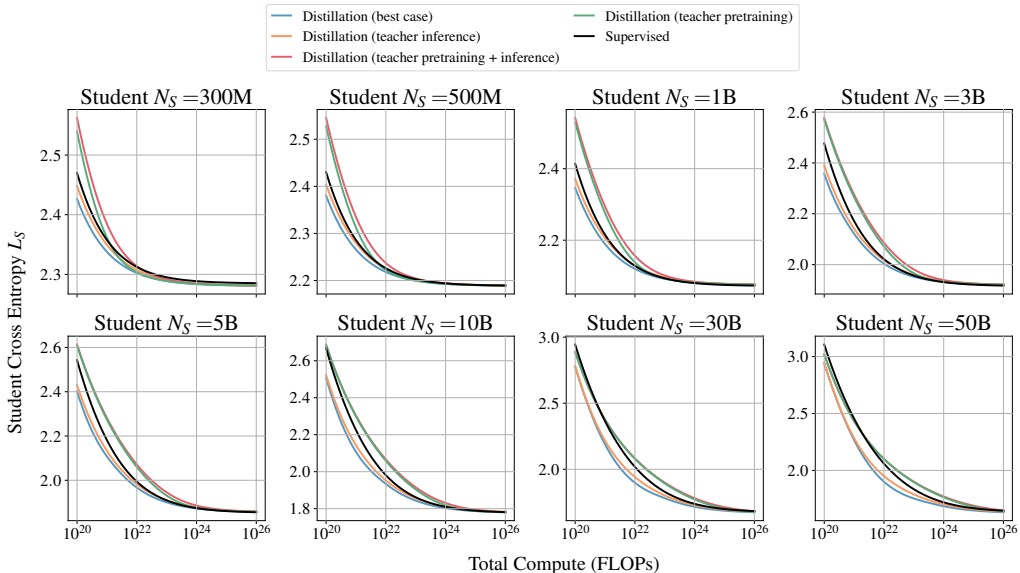

*Figure 17.* **Compute optimal distillation student cross-entropies.** For eight student sizes, the optimal student validation cross-entropy $L_S^*$ in each of the distillation scenarios considered as the total compute is varied.

**The compute at which distillation and supervised learning produce similar models grows with student size.** Continuing the previous observation, we see in Figure 17 that supervised cross-entropy approaches the *best case* and *teacher inference* student cross-entropies at a value of compute which increases with compute, meaning that *larger students benefit from distillation for larger compute budgets than supervised learning*. This implies that if your target student size is small and your compute budget is large, then supervised learning is more likely to be beneficial than if your target student size is larger. The phenomenon happens because larger supervised models saturate in performance at larger values of $D$ (Equation 1), and distillation accelerates progress towards this saturation with the correct choice of teacher (Equation 8), with more capable teachers producing more gains per token.

**Including teacher training in compute produces student cross-entropies higher than in the supervised setting.** In Figure 17 supervised cross-entropy is always below the *teacher pretraining* and *teacher pretraining + inference* scenarios, except at very large compute budgets, when supervised learning and these distillation scenarios produce similar student cross-entropies. This means that if your *only* aim is to produce the model of a target size with the lowest cross-entropy and you do not have access to a teacher, then you should choose supervised learning, instead of training a teacher and then distilling. Conversely, if the intention is to distill into a family of models, or use the teacher as a server model, distillation *may* be more computationally beneficial than supervised learning. This finding aligns with expectations, the alternative implies distillation can outperform direct maximum likelihood optimization given fixed compute.

**The optimal teacher cross-entropy decreases with increasing total compute.** As shown in Figure 18, the optimal teacher cross entropy loss has a decreasing trend with respect to the total compute. However, in the *best case* scenarios, at low compute for larger student, where the number of student tokens is lower than the Chinchilla rule of thumb, an inflection point happens in optimal teacher compute.

We now turn to investigating the optimal distillation configurations that achieve these student cross-entropies.

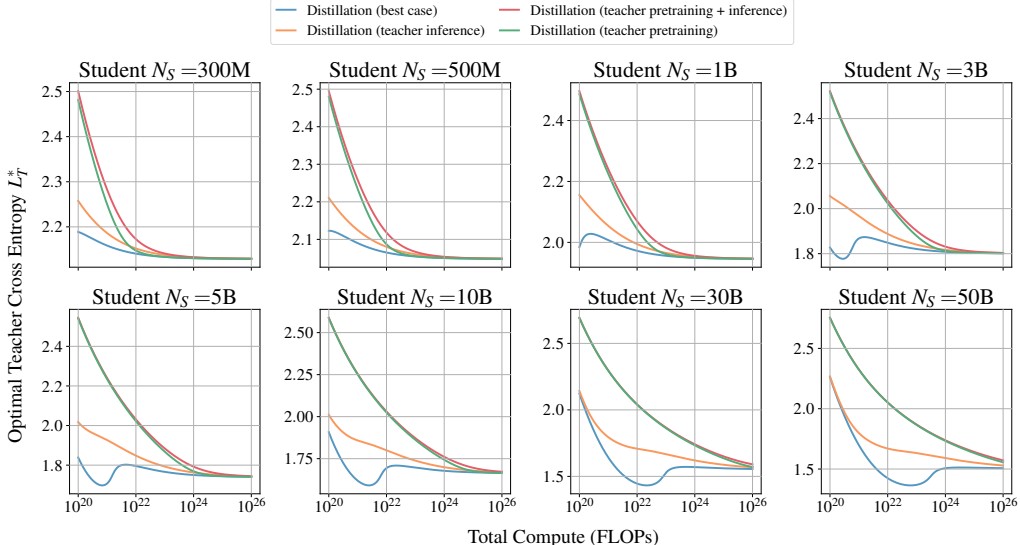

*Figure 18.* **Compute optimal distillation teacher cross-entropies.** For eight student sizes, the optimal teacher validation loss $L_T^*$ resulting in lowest student validation loss $L_S^*$ in each of the distillation scenarios considered (Table 5) the total compute is varied.

### D.4.3. DISTILLATION (BEST CASE)

In the *distillation (best case)* scenario, $\delta_T^{\text{Lgt}} = \delta_T^{\text{Pre}} = 0$, which means that we only account for compute associated with the standard supervised learning case

$$\text{FLOPs}(N_S, D_S, N_T, D_T) \approx \underbrace{3F(N_S)D_S}_{\substack{\text{Student} \\ \text{Training}}}. \tag{30}$$

We call this *best case* as the scenario reflects a freedom to choose *the best* distillation setting for a given student size $N_S$, with all of the compute being put into training the student for as long as possible (maximal $D_S$). In this sense we can consider this the *upper bound* in performance for distillation in our experimental setting.

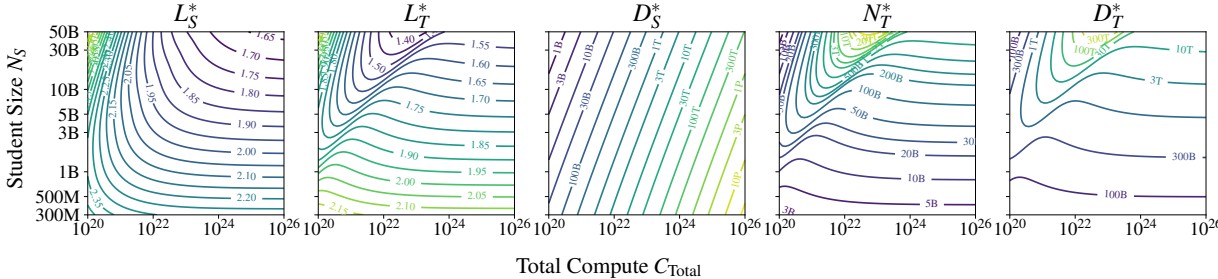

*Figure 19.* **Compute optimal configuration contours for distillation (best case).** The compute optimal quantities $(D_S^*, N_T^*, D_T^*)$ giving rise to the student cross entropies for *best case* in Figure 17 for a range of student sizes. $(N_T^*, D_T^*)$ are the supervised compute optimal combination giving rise to $L_T^*$ in Figure 18.

This scenario represents the setting where a teacher already exists, or we will use the teacher for another purpose, for example a server model. In these scenarios, we do not need to worry about the teacher pretraining cost. Additionally, this teacher may be used to produce the logits for many different students, or we may have saved the logits from the teacher *during its training*. In these cases, the cost for producing the student logits can also be ignored.

The optimal quantities $(D_S^*, N_T^*, D_T^*)$ giving rise to the cross entropies in Figure 17 are shown in Figures 19 and 20. In the *best case* scenario, $L_T^*$ is determined, however $N_T^*$ and $D_T^*$ are not determined because they do not enter into the compute constraint, yielding a one-dimensional family $(N_T(L_T^*, D_T), D_T)$ of valid solutions to the minimization problem (Equation 28). To provide some guidance for producing $L_T^*$, in Figure 18 we present the supervised compute optimal $(N_T(L_T^*, D_T), D_T)$, i.e. the combination that minimizes FLOPs $\propto F(N_T)D_T$ subject to $L(N_T, D_T) = L_T$.

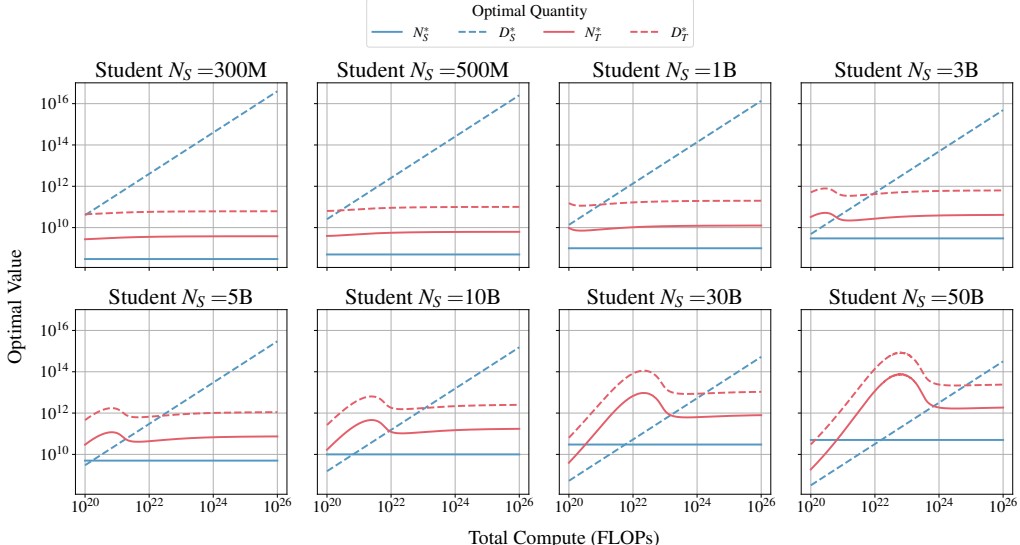

*Figure 20.* **Compute optimal configurations for distillation (best case).** For eight student sizes, the compute optimal quantities ($D_S^*$, $N_T^*$, $D_T^*$) giving rise to the student cross entropies for *best case* in Figure 17. ($N_T^*$, $D_T^*$) are the supervised compute optimal combination giving rise to $L_T^*$ in Figure 18. This is a one-dimensional slice of Figure 19.

In this scenario, all the compute goes into student tokens, and so in Figure 20 we see optimal student tokens $D_S^*$ increases with compute at the same rate as we could for the supervised model, which is higher for smaller students. The optimal teacher parameters $N_T^*$ and tokens $D_T^*$ move together to produce the $L_T^*$ in Figure 18. Again, the exact values of $N_T^*$, $D_T^*$ in Figure 20 represent the *supervised compute optimal* solution for producing the $L_T^*$, but are not the only solution in this compute scenario, since $N_T^*$, $D_T^*$ are not uniquely determined by the compute constraint.

### D.4.4. DISTILLATION (TEACHER INFERENCE)

In the *distillation (teacher inference)* scenario, $\delta_T^{\text{Lgt}} = 1$, $\delta_T^{\text{Pre}} = 0$, which means that we account for compute associated with the standard supervised learning case as well as the cost for producing the logits for the student

$$\text{FLOPs}(N_S, D_S, N_T, D_T) \approx \underbrace{3F(N_S)D_S}_{\substack{\text{Student} \\ \text{Training}}} + \underbrace{F(N_T)D_S}_{\substack{\text{Teacher} \\ \text{Logits}}}. \tag{31}$$

This scenario represents the setting where a teacher already exists, but logits for the distillation still need producing. The optimal quantities ($D_S^*$, $N_T^*$, $D_T^*$) giving rise to the cross entropies in Figure 17 are shown in Figures 21 and 22.

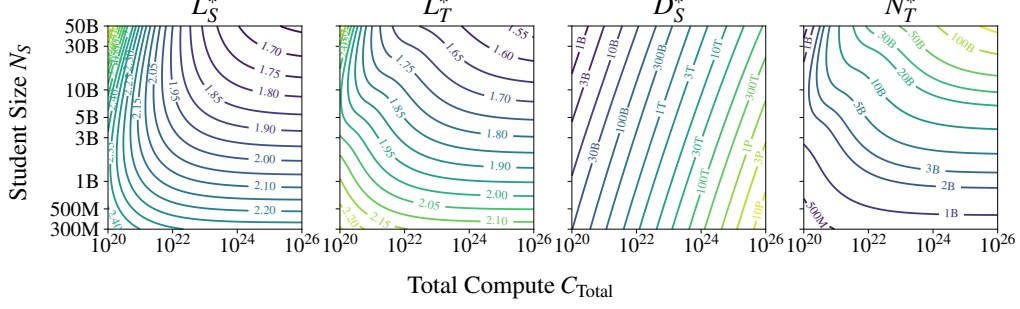

*Figure 21.* **Compute optimal configuration contours for distillation (teacher inference).** The compute optimal quantities ($D_S^*$, $N_T^*$, $D_T^*$) giving rise to the student cross entropies for *teacher inference* in Figure 17.

**The teacher should be overtrained.** In the *teacher inference* scenario, $D_T^*$ does not contribute directly to compute but instead indirectly $N_T^*$ subject to $L_T^*$. To minimize $N_T^*$ at a given $L_T^*$, the solution is to maximize $D_T^*$ as is seen in Figure 22;

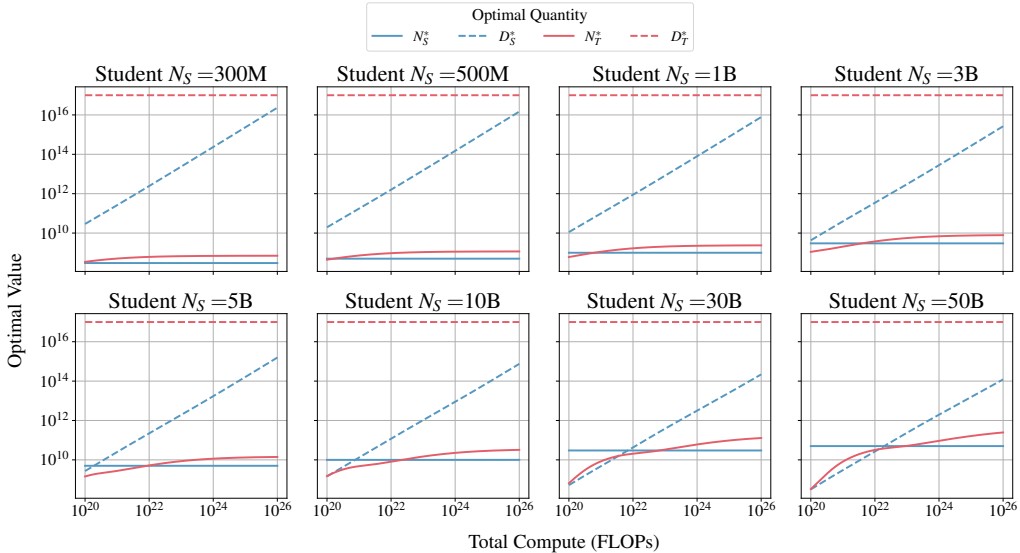

*Figure 22.* **Compute optimal configurations for distillation (teacher inference).** For eight student sizes, the compute optimal quantities $(D_S^*, N_T^*, D_T^*)$ producing the student cross entropies for *teacher inference* in Figure 17. This is a one-dimensional slice of Figure 21.

$D_T^*$ takes the largest value allowed in our numerical optimization, $10^{17}$ tokens. Although not surprising, this demonstrates the benefit of producing *overtrained teachers*, instead of taking the tempting strategy of using compute optimal teachers followed by a long distillation process into a smaller student model.

**As compute is increased, relatively less should be spent on student training, and more on teacher logit inference.** The compute allocations resulting from the optimal combination are shown in Figure 23. We see that in all cases, the student training term (blue) decreases as compute increases, whereas the teacher logits (orange) increases. This happens because as compute increases: i) optimal student tokens increases at a rate approximately independent of compute, ii) the teacher size increases with compute to provide a stronger signal, while iii) the student size is fixed (see Figure 22).

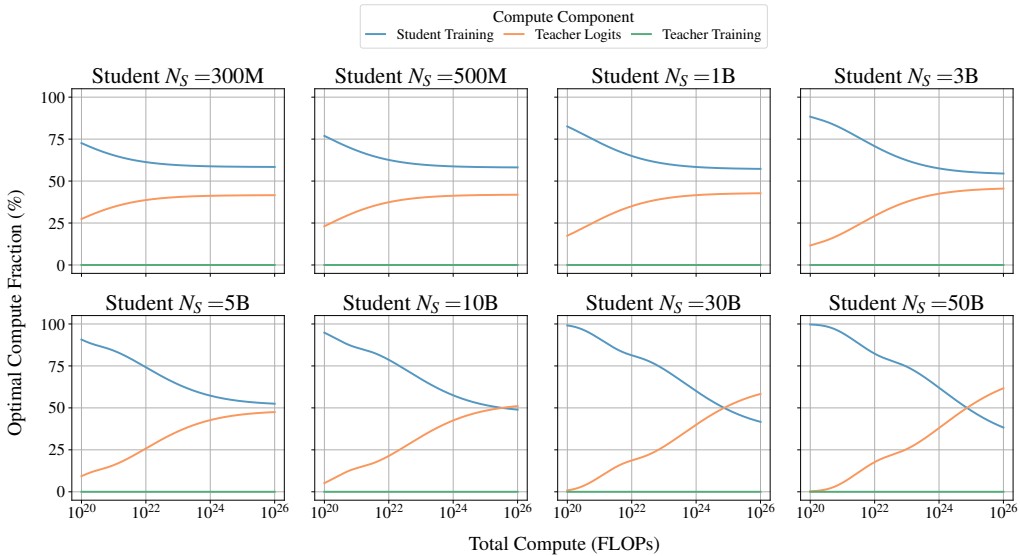

*Figure 23.* **Compute optimal allocations for distillation (teacher inference).** For eight student sizes, the compute optimal allocations corresponding to the terms in Equation 29 for the compute optimal values in Figure 22.

### D.4.5. DISTILLATION (TEACHER PRETRAINING)

In the *distillation (teacher pretraining)* scenario, $\delta_T^{\text{Lgt}} = 0$, $\delta_T^{\text{Pre}} = 1$, which means that we account for compute associated with training the teacher, in addition to the standard training cost of the student, but *not* the cost of producing the logits

$$\text{FLOPs}(N_S, D_S, N_T, D_T) \approx \underbrace{3F(N_S)D_S}_{\substack{\text{Student} \\ \text{Training}}} + \underbrace{3F(N_T)D_T}_{\substack{\text{Teacher} \\ \text{Training}}}. \tag{32}$$

This scenario represents when we want to figure out which teacher to produce to distill into sufficiently many different students, storing the teacher logits for reuse, effectively ammortizing the cost of producing the logits. Here, contrary to the previous two scenarios (Appendices D.4.3 and D.4.5), the teacher size $N_T$ *and* teacher tokens $D_T$ contribute directly to the compute accounting (Equation 32). The optimal quantities ($D_S^*$, $N_T^*$, $D_T^*$) giving rise to the cross entropies in Figure 17 are shown in Figures 24 and 25.

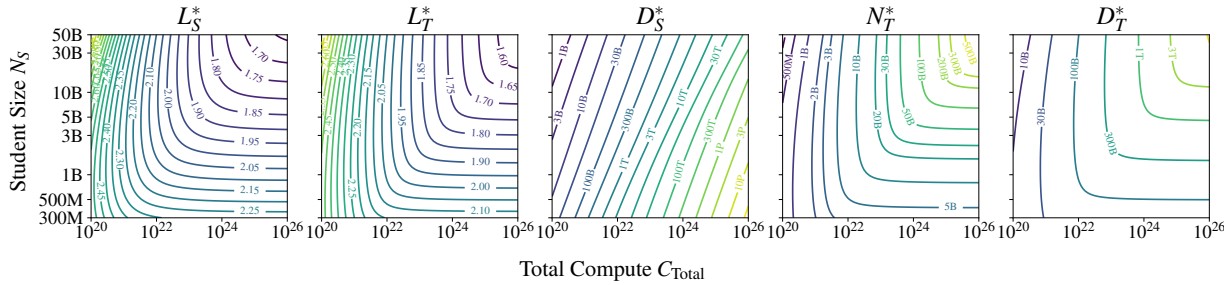

*Figure 24.* **Compute optimal configuration contours for distillation (teacher pretraining).** The compute optimal quantities ($D_S^*$, $N_T^*$, $D_T^*$) giving rise to the student cross entropies for *teacher pretraining* in Figure 17.

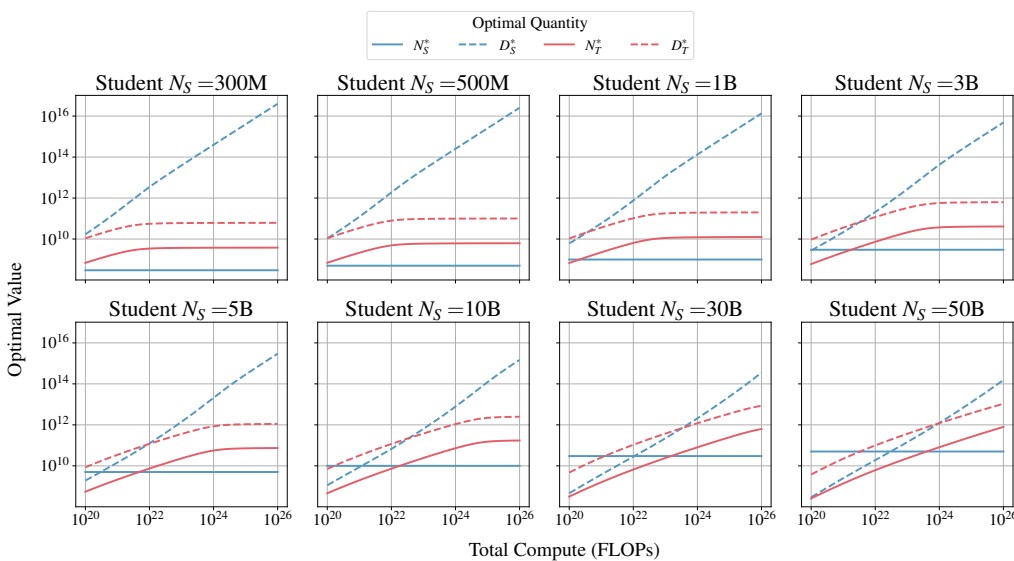

*Figure 25.* **Compute optimal configurations for distillation (teacher pretraining).** For eight student sizes, the compute optimal quantities ($D_S^*$, $N_T^*$, $D_T^*$) giving rise to the student cross entropies for *teacher pretraining* in Figure 17. This is a one-dimensional size of Figure 24.

**The compute optimal teacher for distillation is a supervised compute optimal teacher.** In Figure 25 we see that the $M_T \equiv D_T/N_T$ ratio of the teacher is constant for all values of compute, and can be compared to the ratio in Figure 19. This can be understood as there is no inference cost to pay for making the teacher large; we are only minimizing the training compute budgets of two models, and the most efficient way to produce a teacher with a given cross-entropy $L_T$ is a teacher

that is compute-optimal in a supervised sense. Note that this conclusion is the *opposite* to the finding in Appendix D.4.4. There, the inference is expensive, and so the teacher should be *overtrained*. Here, teacher training is expensive, so teacher training should be *compute optimal*.

**As compute is increased, relatively less should be spent on teacher training, and more on student training.** In Figure 26 we see the compute allocations for the configurations shown in Figure 25, and see that student training relative compute (blue) increases with increasing compute budget, while the teacher training (green) decreases with increasing compute budget. This happens because, as in all compute scenarios, with increasing compute, the optimal student tokens $N_S^*$ increases (Figure 25). Teacher size and tokens are also increasing with increasing compute, providing a stronger signal for the student with more tokens to learn. However, this increase in teacher size and tokens plateaus, while the student tokens continues to increase. This is because here the teacher is compute optimal, and so the amount of compute needed to improve the learning signal for the student is much less than the amount of compute needed to train the student for to make use of that signal, due to the stronger diminishing returns with respect to $D_S$ at a fixed $N_S$ (Equation 8).

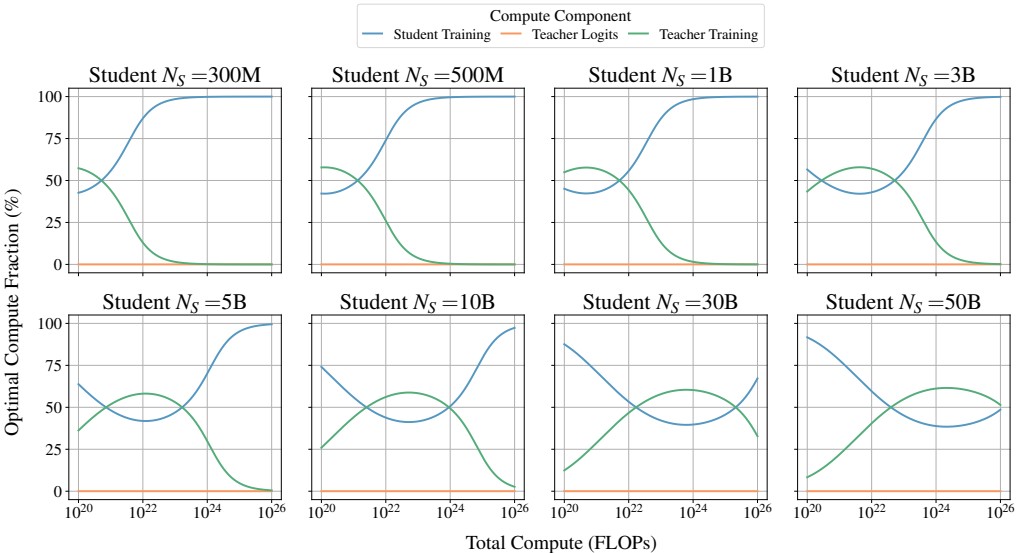

*Figure 26.* **Compute optimal allocations for distillation (teacher pretraining).** For eight student sizes, the compute optimal allocations corresponding to the terms in Equation 29 for the compute optimal values in Figure 25.

### D.4.6. DISTILLATION (TEACHER PRETRAINING + INFERENCE)

In the *distillation (teacher pretraining + inference)* scenario, $\delta_T^{\text{Lgt}} = \delta_T^{\text{Pre}} = 1$, which means that we account for all costs associated with distilling a single student

$$\text{FLOPs}(N_S, D_S, N_T, D_T) \approx \underbrace{3F(N_S)D_S}_{\substack{\text{Student} \\ \text{Training}}} + \underbrace{F(N_T)D_S}_{\substack{\text{Teacher} \\ \text{Logits}}} + \underbrace{3F(N_T)D_T}_{\substack{\text{Teacher} \\ \text{Training}}}. \tag{33}$$

This scenario can be thought of as the compute optimal *worst case* scenario for distillation, i.e. *one teacher* is trained *only* for the purposes of *one* student. As in Appendix D.4.4, teacher size $N_T$ *and* teacher tokens $D_T$ contribute directly to the compute accounting (Equation 33). The optimal quantities $(D_S^*, N_T^*, D_T^*)$ giving rise to the cross entropies in Figure 17 are shown in Figures 27 and 28.

**Compute optimal teachers should be used for lower compute budgets and overtrained teachers should be used for larger compute budgets.** In Figure 28 we see a teacher configuration that interpolates between the *teacher pretraining* (Appendix D.4.5) and *teacher inference* (Appendix D.4.4) compute scenarios. At low compute, the optimal number of student tokens $D_S^*$ is not too large, this means there is little penalty to increasing the teacher size, resulting in an approximately supervised compute-optimal teacher given a teacher compute budget. Once the optimal number of student tokens

becomes higher than the optimal number of teacher tokens, there is significant penalty to increasing the teacher size. At this point, the teacher solution starts to become the overtrained solution seen in *teacher inference*, the optimal teacher tokens continue to increase polynomially, but this is not followed with an increase in the teacher size. For sufficiently high compute, corresponding to a large number of student distillation tokens, the compute penalty for teacher size is so large that optimal teacher size decreases with compute.

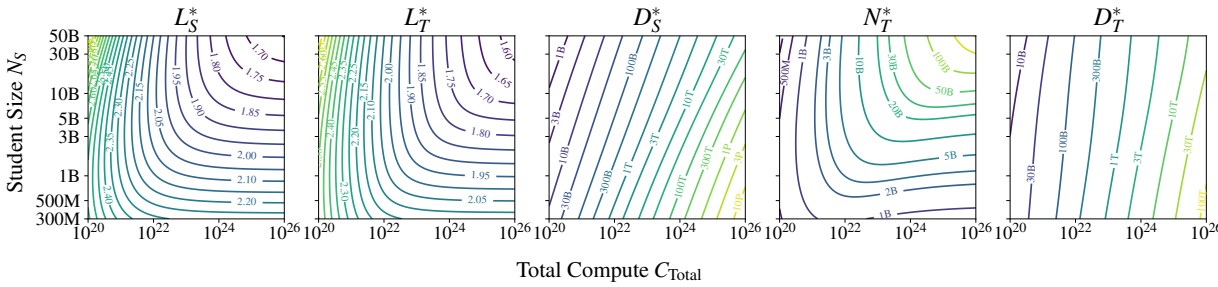

*Figure 27.* **Compute optimal configuration contours for distillation (teacher pretraining + inference).** The compute optimal quantities ($D_S^*$, $N_T^*$, $D_T^*$) giving rise to the student cross entropies for *teacher pretraining + inference* in Figure 17.

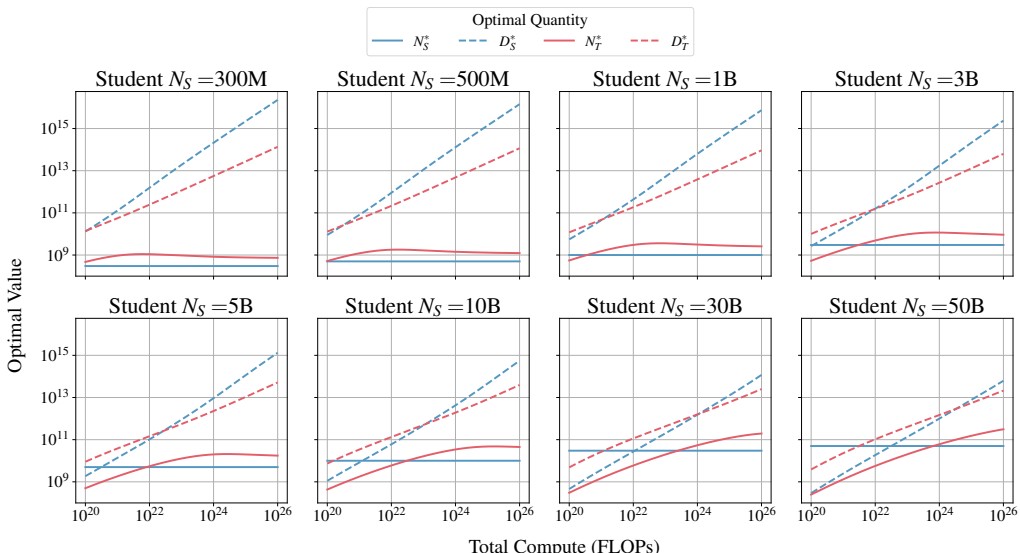

*Figure 28.* **Compute optimal configurations for distillation (teacher pretraining + inference).** For eight student sizes, the compute optimal quantities ($D_S^*$, $N_T^*$, $D_T^*$) giving rise to the student cross entropies for *teacher pretraining + inference* in Figure 17. This is a one-dimensional size of Figure 27.

**For small students, as compute grows, more should be spent on training the student and producing logits for the student.** In Figure 29 we see the compute allocations for the configurations shown in Figure 28. Compute optimal smaller models tend to have smaller teachers, and optimal teacher tokens always grow at a slower rate than student tokens, and so teacher the training cost is relatively small. As compute grows, the student is distilled on more tokens, and the teacher always becomes slightly larger than the student, which gives rise to most compute being allocated to standard student training compute component and producing the logits for this training.

**For large students, as compute grows, more should be spent on training the teacher, until a transition happens where more should be spent on training the student and producing logits for the student.** The explanation for the phenomenon is as above, except that the larger students need a more capable teacher to learn from as compute grows, and so initially compute needs to bused to produce the teachers required. After a certain amount of compute, the large number of

optimal student distillation tokens moves the optimal solution towards an overtrained teacher scenario, and more compute being allocated to student training and logit production.

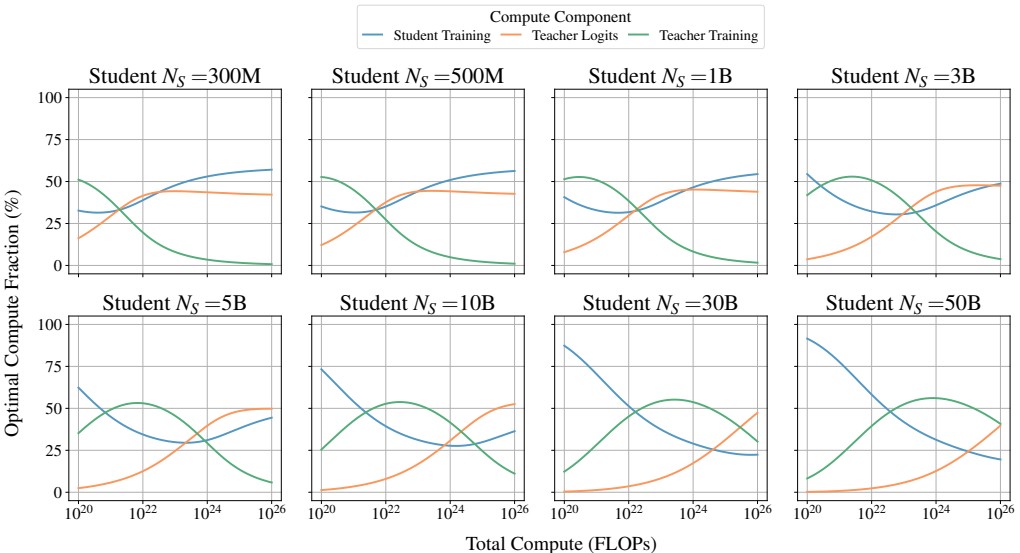

*Figure 29.* **Compute optimal allocations for distillation (teacher pretraining).** For eight student sizes, the compute optimal allocations corresponding to the terms in Equation 29 for the compute optimal values in Figure 28.

### D.4.7. OPTIMAL TEACHER TRAINING AND STUDENT DISTILLATION TOKENS

To aid in comparing the different compute strategies presented in Appendices D.4.3 to D.4.6, we now present each compute optimal value for all strategies, including supervised. Here, we show compute-optimal distillation student tokens $D_S^*$ in Figure 31 and compute-optimal teacher pretraining tokens $D_T^*$ in Figure 31.

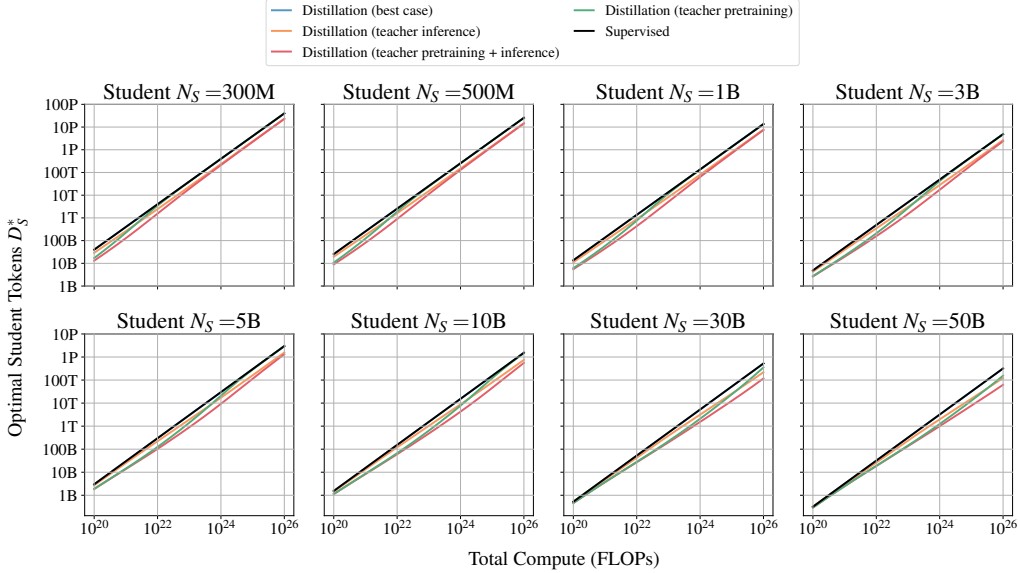

*Figure 30.* **Compute optimal distillation student tokens.** For eight student sizes, the compute optimal student tokens $D_S^*$ giving rise to the student cross-entropies for all compute scenarios, including supervised.

**In all scenarios, student tokens should be increased with compute similar to in the supervised case.** We see in Figure 30 that, as in Chinchilla (Hoffmann et al., 2022), supervised tokens are increased polynomially with compute. *Dis-*

*tillation (best case)* follows the exact same allocation, as does *distillation (pretraining)* with asymptotically large compute. All other methods follow the same increase rate, but with scenario-dependent offsets.

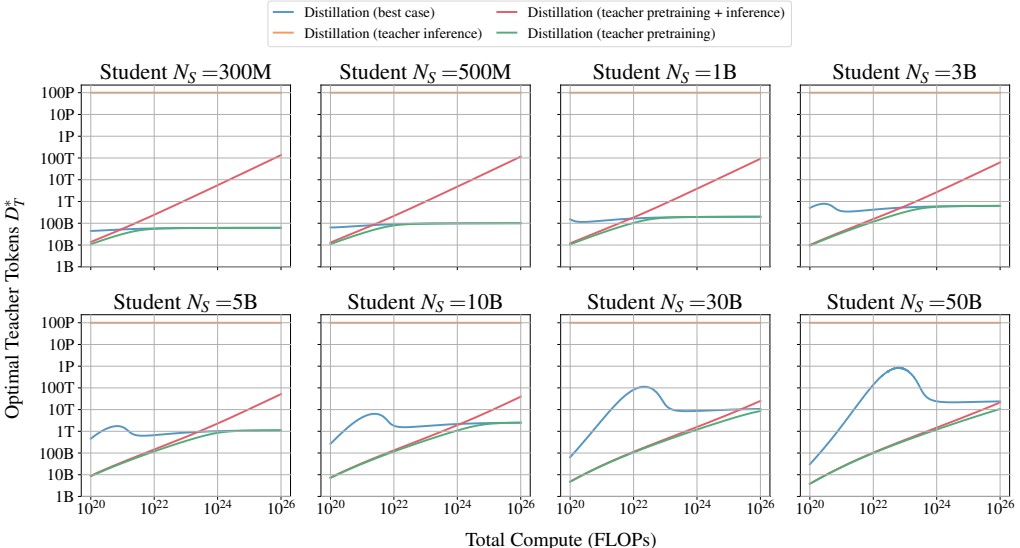

*Figure 31.* **Compute optimal distillation teacher tokens.** For eight student sizes, the compute optimal teacher tokens $D_T^*$ giving rise to the student cross-entropies for all compute scenarios.

**Optimal teacher tokens interpolate between scenarios based on compute allocation.** In Figure 31 we can see more clearly the interpolation behavior discussed in Appendix D.4.6. At low compute, *teacher pretraining* and *teacher pretraining + inference* share optimal solutions because the number of student tokens $N_S^*$ is small. At high compute, *teacher pretraining + inference* approaches *teacher inference*, while *teacher pretraining* approaches *best case*, as $N_S^*$ is large, and costs associated with teacher pretraining become less important.

D.4.8. OPTIMAL TEACHER SIZE

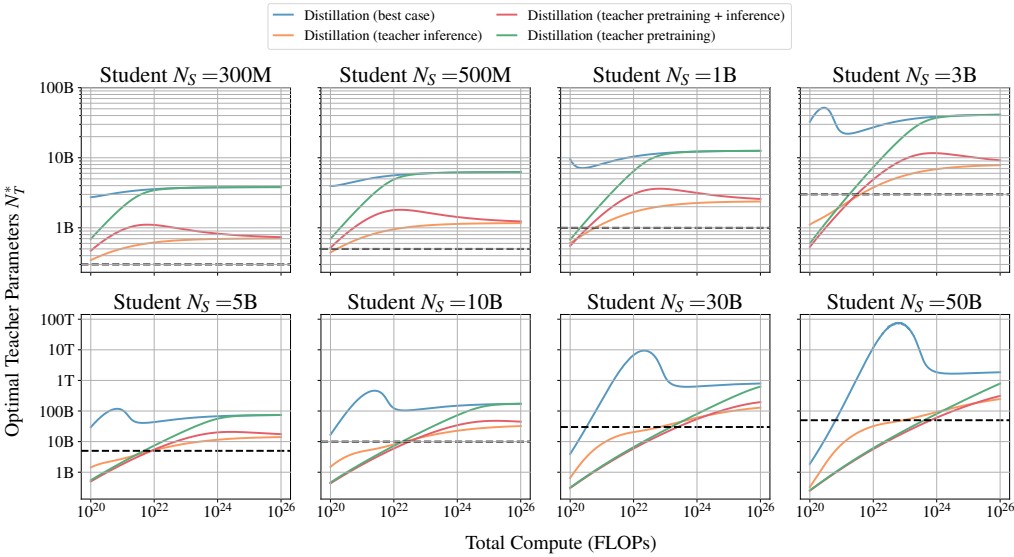

*Figure 32.* **Compute optimal distillation teacher size.** For eight student sizes, the compute optimal teacher size $N_T^*$ giving rise to the student cross-entropies for all compute scenarios.

**Optimal teacher size interpolate between scenarios based on compute allocation.** As in the optimal teacher tokens $N_T^*$ in Figure 31, the same mechanism causes interpolation behavior in optimal teacher size (see Figure 32).

## D.5. Compute and data efficiency gains for distillation compared to supervised learning

In this final section, we use the compute-optimal strategies developed through Appendices D.4.3 to D.4.6 and understand, for each distillation compute scenario (Table 5) if it is more compute and/or data efficient to use distillation compared to supervised learning in order to produce a desired model (i.e. of a given size $N_S$ with a desired performance, measured in cross-entropy $L_S$).

In Figure 33 we show the amount of compute needed to distill a student of a given size to a given cross-entropy as a multiple of the compute that supervised learning needs to produce the same result. We do this for for each of the distillation compute scenarios, whose optimal configurations are given in Appendices D.4.3 to D.4.6. In Figure 34 we show the same, except we show the number of tokens needed to distill a student of a given size to a given cross-entropy as a multiple of the number of tokens that supervised learning needs to produce the same result. Our distillation token accounting depends on compute scenario:

$$D_{\text{Dist.}} = D_S + \delta_T^{\text{Pre}} D_T, \tag{34}$$

i.e. we only count teacher tokens if the teacher pretraining cost is also included in the compute cost (see Equation 29).

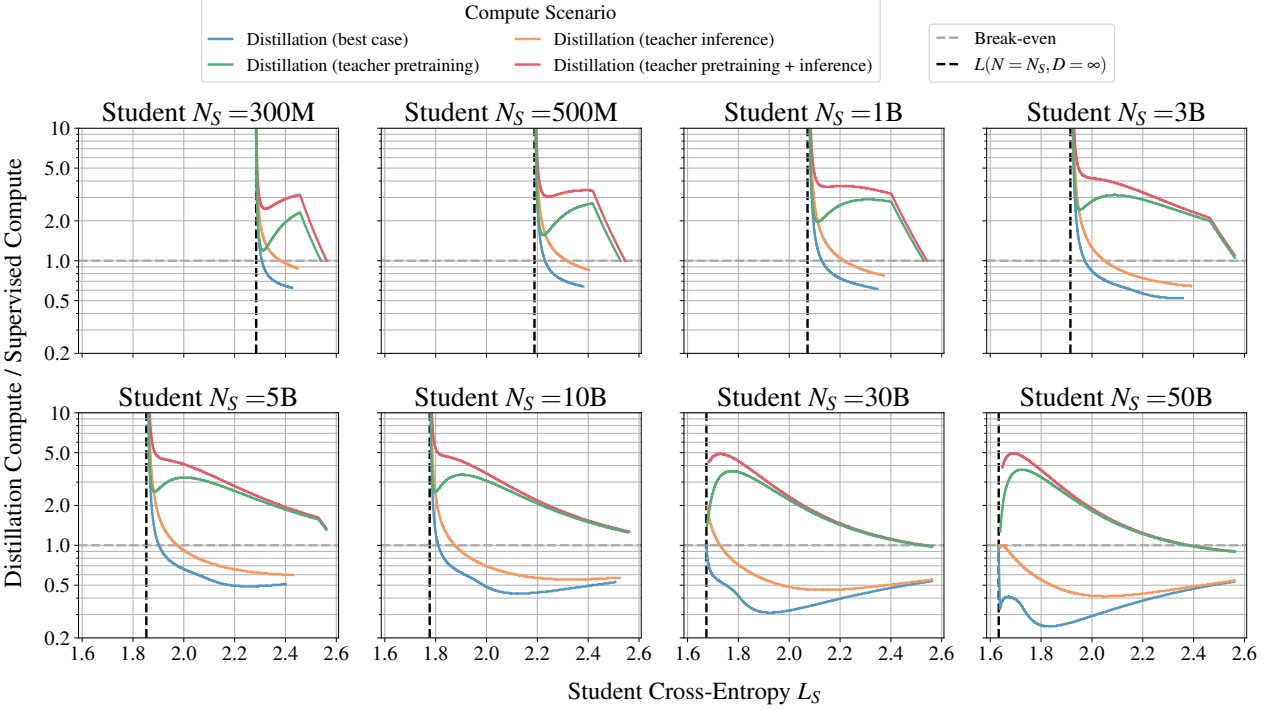

*Figure 33.* **Compute optimal distillation compute ratios.** For eight student sizes, the amount of supervised compute needed to produce a student of the indicated size and cross-entropy. The horizontal dashed line indicates the break-even point, when doing supervised learning is as computationally efficient as the corresponding distillation compute scenario. Values greater (less) than one indicate distillation is more (less) expensive than supervised learning for producing a model of the indicated size and cross-entropy. The vertical dashed line indicates the lowest cross-entropy achievable by that student.

**When teacher training is discounted, distillation is often more efficient.** In Figure 33, the *base case* (blue) and *teacher inference* (orange) compute scenarios are below the grey dashed line for cross-entropies slightly above the lowest possible cross-entropy (vertical grey dashed line), meaning less compute is needed for distillation than supervised learning. This compute efficiency translates into data efficiency (see Figure 34).

**To produce the strongest student possible, supervised learning is more efficient.** In Figures 33 and 34, the *base case* (blue) and *teacher inference* (orange) compute scenarios attain values larger than one as the target cross-entropy $L_S$ approaches the limiting value $L(N = N_S, D = \infty)$ for each student size $N_S$, (vertical dashed line). This suggests i) the existence of a more efficient training strategy where distillation is used as an initial training stage, with a transition to

supervised learning based on a token or cross-entropy threshold, and ii) potentially increased importance of data mixtures ($\lambda \leq 1$, see Appendix G.1) when distilling with significant token and/or compute budgets. We leave this for future work.

**In situations where teacher training is required, supervised learning is more efficient.** As observed in Appendix D.4.2, for all student sizes, if teacher pretraining is included in the computational cost of producing a student, supervised learning is always more efficient than distilling. This can be seen from Figure 33 as the *teacher pretraining* (green) and *teacher pretraining + inference* (red) compute scenarios are above the grey dashed line, which means more compute is needed for distillation than supervised learning in those compute scenarios. This compute efficiency translates into data efficiency (see Figure 34).

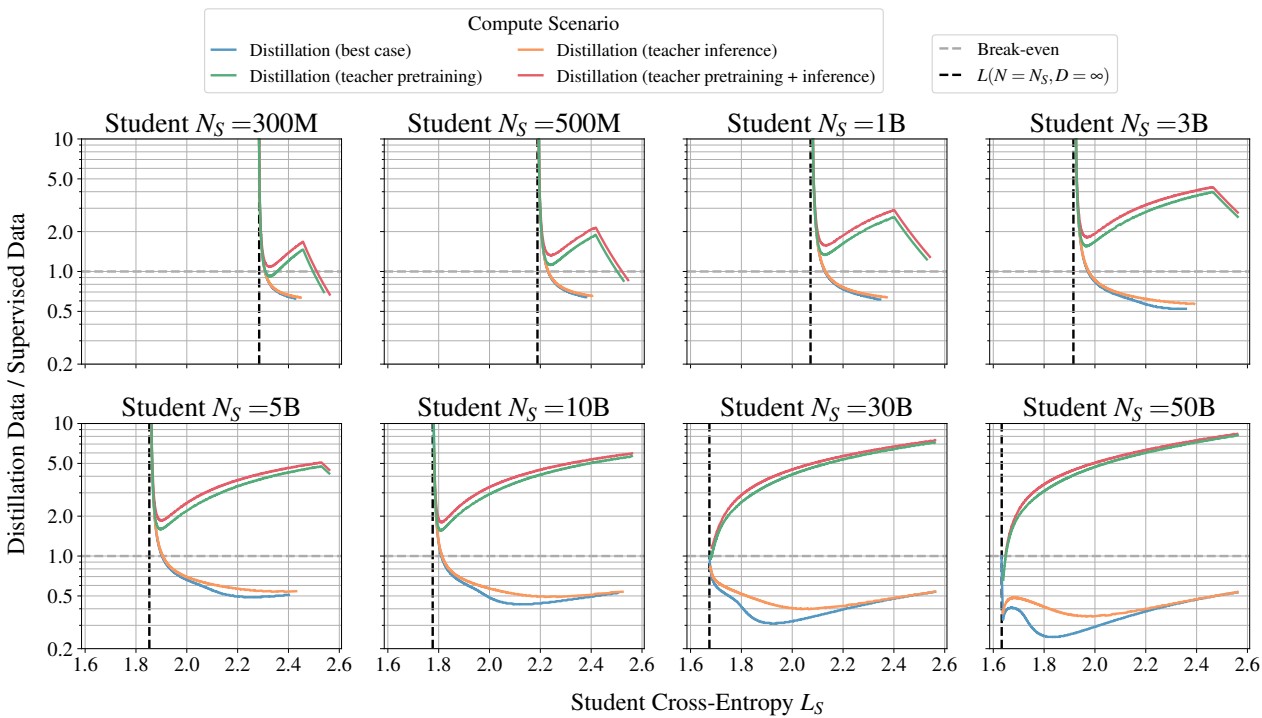

*Figure 34.* **Compute optimal distillation data ratios.** For eight student sizes, the number of tokens compute needed to produce a student of the indicated size and cross-entropy. The horizontal dashed line indicates the break-even point, when doing supervised leaning is as data efficient as the corresponding distillation compute scenario. Values greater (less) than one indicate distillation is more (less) expensive than supervised learning for producing a model of the indicated size and cross-entropy. The vertical dashed line indicates the lowest cross-entropy achievable by that student.

**Distillation is more efficient for larger students.** In Figure 33 we see in the *pretrain + inference* scenario, producing a $N_S$ =500M student with a cross-entropy of 2.4 has roughly 3/4 the compute cost of producing the same model with supervised learning, whereas producing a $N_S$ =10B student with a cross-entropy of 2.2 has roughly 1/2 the compute cost of producing the same model with supervised learning. In terms of data (Figure 34), the 500M and 10B configurations use roughly 2/3 and 1/2 the number of tokens of their supervised counterparts respectively. *The efficiency gains from distillation are potentially greater for larger students when considering compute or data.*

## E. Additional Results

In this section, we provide an extensive list of studies, including downstream evaluations of distillation. We cover the models used as teachers, examine the Kullback-Leibler Divergence (KLD) between teacher and student in fixed token-to-size ratios, and present supplementary materials to Section 4.1. Additionally, we investigate the limiting behavior of our scaling law, weak-to-strong generalization, and conduct a model calibration study to assess fidelity. These analyses offer a comprehensive view of the factors influencing distillation performance and the behavior of our proposed scaling laws.

### E.1. Downstream evaluations

In all settings, we optimize for and predict validation cross-entropy. To confirm that the validation cross-entropy is a good proxy for the downstream evaluation that is the ultimate interest, in Figure 35 we show evaluations for the supervised teachers and the distilled students on downstream evaluation tasks. ARC Easy (Bhakthavatsalam et al., 2021), ARC Challenge (Bhakthavatsalam et al., 2021), HellaSwag (Zellers et al., 2019), Piqa (Bisk et al., 2020), Sciq (Welbl et al., 2017), WinoGrande (Sakaguchi et al., 2021) and Lambada OpenAI (Paperno et al., 2016) are zero-shot tasks. TriviaQA (Joshi et al., 2017) and WebQS (Berant et al., 2013) are one-shot tasks. TriviaQA evaluation is on the larger and more challenging *Web* split. CoreEn is the average of both the zero-shot and one-shot tasks.

We have included GSM8K (Cobbe et al., 2021) and MMLU (Hendrycks et al., 2021b;a). GSM8K is used in an 8-shot chain of thought setting, following LLaMA (Touvron et al., 2023a;b; Dubey et al., 2024). MMLU is used in a five-shot setting. These perform near-random for most of the models, and only show a slightly upwards trend for models with low cross-entropy. This near-random performance is due to the use of the C4 dataset in training, and we note that we do not aim for competitive downstream evaluation results.

Finally, we note that the relation between cross-entropy and downstream performance for the supervised and distilled models is similar. We suspect this is because the student behaves like a low variance expectation of a biased teacher in the KL-matching distillation scenario (Menon et al., 2020), and we anticipate that the relationship between cross-entropy and downstream performance may be different for alternative distillation strategies.

All models are evaluated using an internal version of the open-source `lm-evaluation-harness` (Gao et al., 2024).

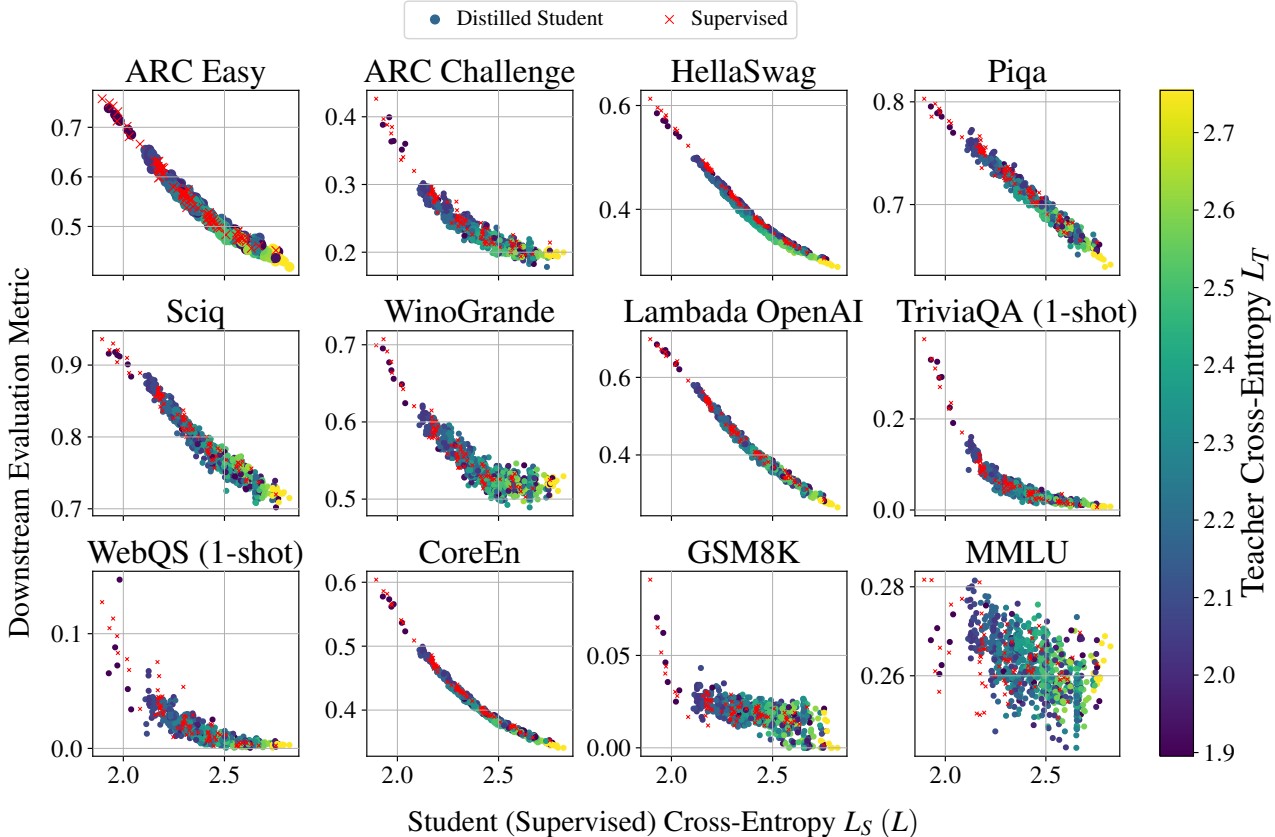

*Figure 35.* **Model downstream evaluations.** Each scatter point is a different model. The circular points correspond to distilled students, whose color indicates the cross-entropy of the teacher used for that distillation process. The red crosses correspond to the supervised models (i.e. the teachers). For a discussion of the individual metrics and datasets, see Appendix E.1.

### E.2. Teachers used in distillation

In Figure 36 we show the cross-entropies of the models used as teachers in Section 4.2, and for fitting the supervised scaling law: i) eleven of fixed-$M$ ratio models following the Chinchilla rule of thumb $D/N = M^* \approx 20$ (Hoffmann et al., 2022), ii) six models on $D = 512B$ tokens (Figure 36a), and iii) four IsoFLOP profiles (Figure 36b). Together this produces 74 runs corresponding to tuples of $(N, D, L)$.

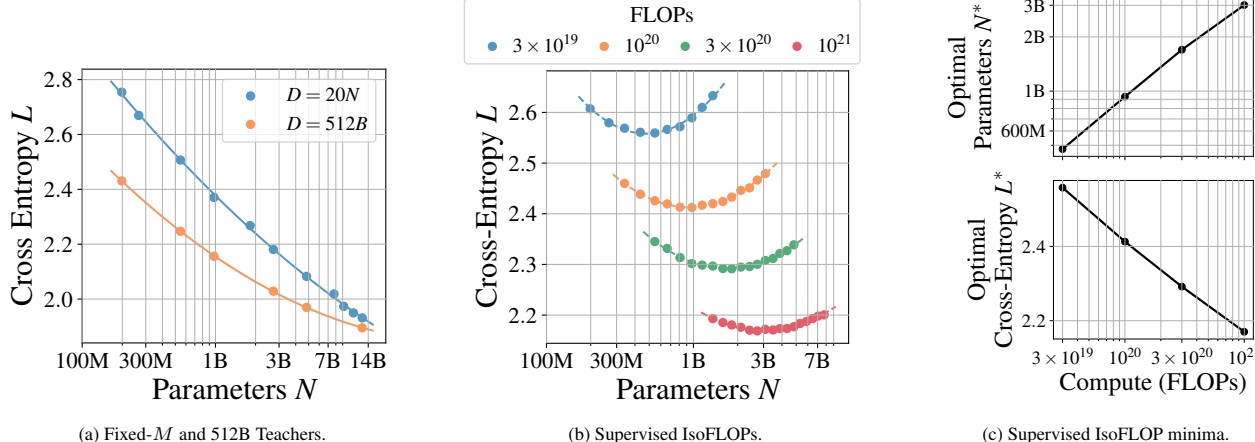

(a) Fixed-$M$ and 512B Teachers.    (b) Supervised IsoFLOPs.    (c) Supervised IsoFLOP minima.

*Figure 36.* **Supervised IsoFLOPs.** **(a)** The cross-entropy of supervised models trained with either a Chinchilla optimal $M = D/N \approx 20$ or on 512B tokens. **(b)** The cross-entropy supervised models trained with four ISOFLOP profiles $C \in \{3 \times 10^{19}, 10^{20}, 3 \times 10^{20}, 10^{21}\}$. **(c)** The optimal supervised parameters $N^*(C) = \arg\min_N L(C)$ for each IsoFLOP profile, and the loss $L^*(C)$ achieved by that model.

Coefficient estimation (Appendix F.1) yields the scaling coefficients shown in Table 6, and a scaling law which has $\lesssim 1\%$ relative prediction error, including when extrapolated from weaker to stronger models (see Figure 5a).

### E.3. Fixed-$M$ teacher/fixed-$M$ students and the capacity gap

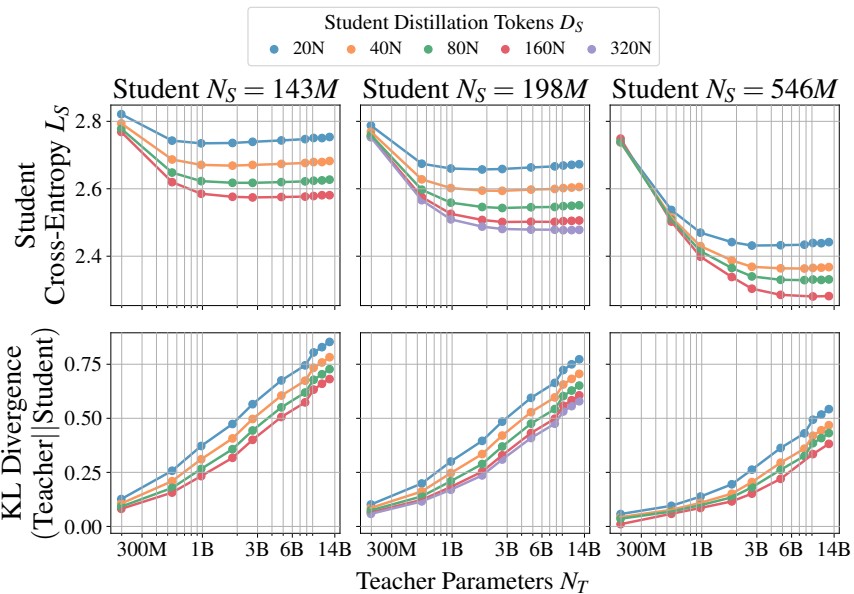

*Figure 37.* **Fixed $M$ Teacher/Fixed $M$ Student.** Students of three sizes trained with different $M_S = D_S/N_S = 20$ ratios are distilled from teachers with $M_T = D_T/N_T \approx 20$. This is a more complete version of Figure 3.

In Figure 37, the *capacity gap* in knowledge distillation can be seen. Improving a teacher's performance does not always improve a student's, and even reduces the performance after a certain point. The KLD between teacher and student is an increasing function of teacher size in all cases, which means as the teacher improves its own performance, the student finds

the teacher more challenging to model, which eventually prevents the student from taking advantage of teacher gains. See Appendix E.8.2 for an investigation using calibration to understand where this mismatch occurs.

### E.4. Full distillation scaling law IsoFLOP profiles

In Figure 38a we provide the full six fixed $M$ Teacher/IsoFLOP Student profiles, only two of which were shown in Figure 2. These experiments enable the reliable determination of $\alpha', \beta', \gamma', A'$ and $B'$. In Figure 38b we provide the full four IsoFLOP teacher/ fixed $M$ student, only two of which were shown in Figure 3. These experiments enable the reliable determination of $c_0, c_1, f_1$ and $d_1$.

**Strong-to-weak generalization occurs.** For the weaker teachers ($N_T \leq 2.72B$), The horizontal dashed line in each pane shows the cross-entropy achieved by the teacher (Appendix E.2). we see that for students larger than the teacher ($N_S > N_T$) and for sufficiently large compute budgets, *the student is able to outperform the teacher* (see Appendix E.7 for a detailed one-dimensional slice).

**A stronger teacher signal is needed in order for stronger students to outperfom the supervised baseline.** The horizontal dashed line in each pane shows the cross-entropy achieved by the student if trained using supervised learning (Appendix E.2). We see that weaker students benefit more from distillation, as e.g. the 198M student has all observed data below this dashed line, meaning all distillations outperform the supervised baseline. However, for the 1.82B student, only $10^{21}$ FLOP teachers produce distilled students that outperform the supervised baseline.

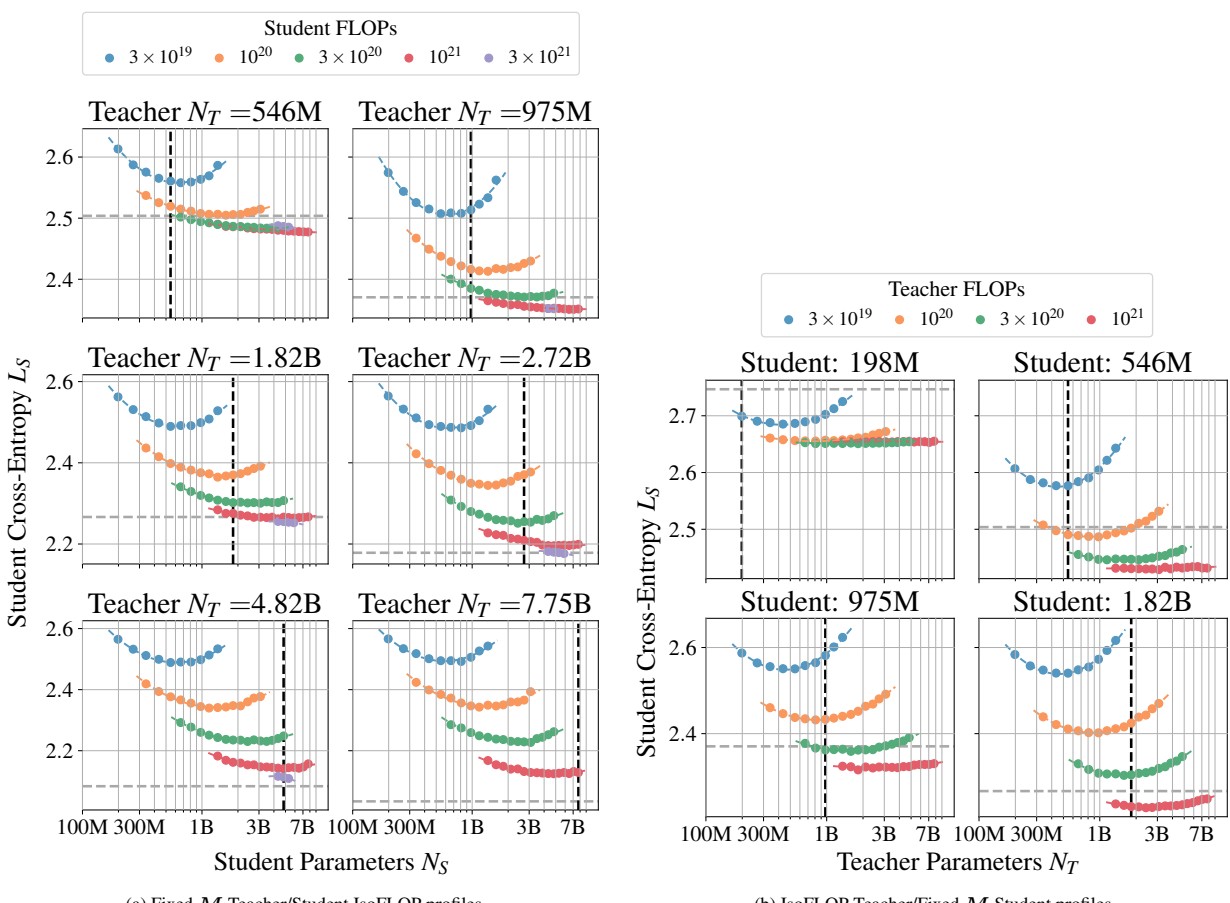

(a) Fixed $M$ Teacher/Student IsoFLOP profiles.

(b) IsoFLOP Teacher/Fixed $M$ Student profiles.

*Figure 38.* **Supervised IsoFLOPs.** **(a)** Teachers of six sizes with $M_T = D_T/N_T \approx 20$ are distilled into Students with four IsoFLOP profiles, and a small number with $C_S = 3 \times 10^{21}$. The horizontal grey and vertical black dashed lines indicate teacher cross entropy $L_T$ and size $N_T$ respectively. **(b)** Students of four sizes trained with a $M = D_S/N_S = 20$ are distilled from teachers with four IsoFLOP profiles. Horizontal (vertical) dashed lines indicate student supervised cross entropy $\widetilde{L}_S$ (student size $N_S$).

### E.5. Distillation scaling law IsoFLOP optima

The optimal loss values of each IsoFLOP in Figure 38a are shown in Figure 39.

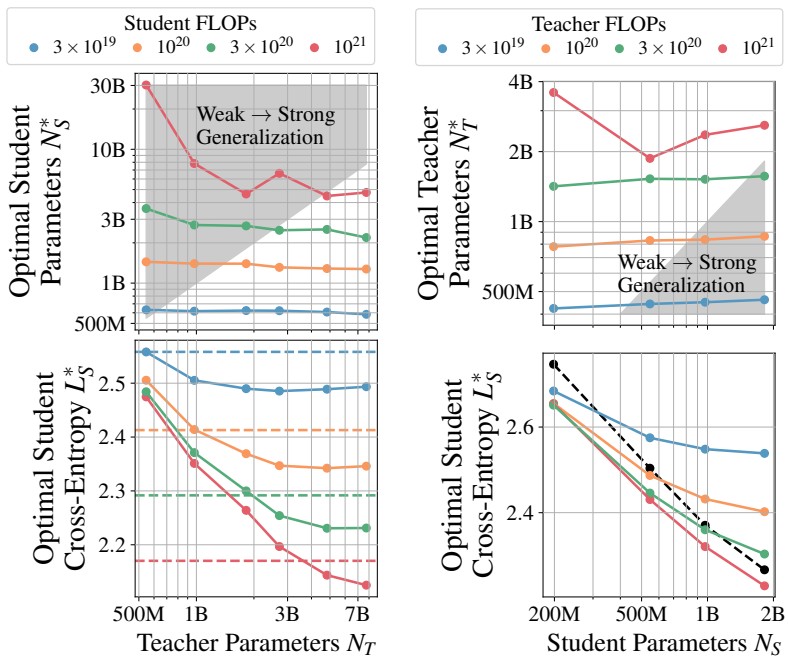

(a) Fixed $M$-Ratio Teacher/Student ISOFlop optima.    (b) Fixed $M$-Ratio Student/Teacher ISOFlop optima.

*Figure 39.* **ISOFlop optima. a)** The optimal student parameters $N_S^* = \arg\min_{N_S} \mathcal{L}(N_S)$ that give the lowest student validation loss for each teacher-student combination shown in Figure 38a. The dashed lines correspond to the validation loss of the optimal supervised models trained with the four corresponding compute budget. **b)** The optimal teacher parameters $N_T^* = \arg\min_{N_T} \mathcal{L}(T_S)$ that give the lowest student validation loss for each teacher-student combination shown in Figure 3. The black dashed line correspond to the validation loss of a $M = D/N = 20$ supervised model of the indicated student size. In both figures, the shaded region corresponds to where *weak to strong generalization* may occur, as $N_S > N_T$ (see Appendix E.7).

### E.6. Distillation with infinite data

From the supervised scaling law (Equation 1) a model with $N$ parameters has a cross-entropy lower bound

$$L(N) \equiv L(N, D = \infty) = E + (AN^{-\alpha})^\gamma \tag{35}$$

which represents the best solution to the training objective subject to constraints from that model's hypothesis space (Hoffmann et al., 2022) and is achieved when the number of training tokens is large ($D \to \infty$). As the hypothesis space of a model is independent of the procedure used to find the solutions, we anticipate that the student with $N_S$ parameters has a cross-entropy lower bound that is the same as the supervised one Equation 35. However, it not immediately clear if this is true in practice, since

$$L_S(N_S) \equiv L_S(N_S, D_S = \infty, L_T = L_T^*) \tag{36}$$

$$= L_T^* + \frac{(A'N_S^{-\alpha'})^{\gamma'}}{(L_T^*)^{c_0}} \left(1 + \left(\frac{L_T^* d_1^{-1}}{L(N_S)}\right)^{1/f_1}\right)^{-c_1 f_1}, \tag{37}$$

where $L_T^* = \arg\min_L(N_S, D_S = \infty, L_T)$ is the teacher cross-entropy that minimizes Equation 8. Upon checking numerically, we do find that Equation 35 is consistent with Equation 37 for a range of models $N, N_S \in [100M, 100B]$ (Figure 40). We stress that unlike our three motivations for the equation properties (Section 4.3), this infinite data limit was imposed added by hand, and is only true for certain values scaling coefficients. This lower bound consistency is evidence

that that our distillation scaling law has desired behavior far outside of observed models, at least along the data and teacher axes. We also note that only the optimal teacher for each student size produces a student cross-entropy lower bound that is consistent with the supervised one. Any other choice produces higher student cross-entropies, either because the teacher is too weak, or due to the capacity gap.

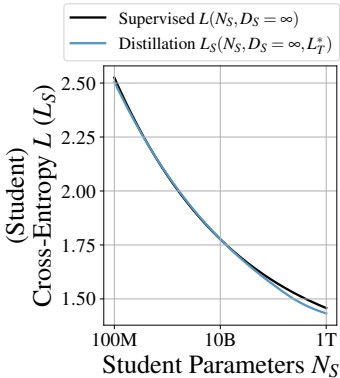

*Figure 40.* **Scaling behavior in the infinite data regime.** For the *optimal* choice of teacher, the loss achieved by all student sizes under distillation is consistent with the loss achievable by supervised learning. This is *not* true for *any* choice of teacher, *only* the optimal one, which can be determined through numerical optimization of the provided distillation scaling laws (see Section 5).

### E.7. Weak-to-strong generalization

In Figure 41 we see that weak-to-strong generalization (Burns et al., 2024; Ildiz et al., 2024) occurs *only in the finite distillation data regime*, and when the number of tokens is sufficiently large, the student cross-entropy increases again, eventually matching the teacher cross-entropy. This can be understood in the following way: i) when the student is larger than the teacher, the student contains in its hypothesis space the function represented by the teacher, ii) when the student is shown the teacher outputs on enough of the data manifold, it eventually matches what the teacher does on the whole data manifold. We note this doesn't explain how and why the student outperforms its teacher, and only constrains its asymptotic (low and high distillation data) behaviors.

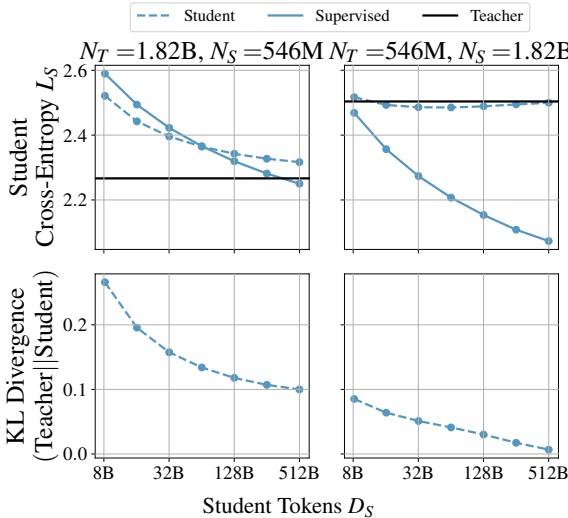

*Figure 41.* **Fixed M-Ratio Teacher varying student data.** We look at *strong to weak* generalization (left) and *weak to strong* (right) distillation, varying distillation tokens $D_S \in [8B, 512B]$.

## E.8. Model calibration

Calibration in LMs refers to the alignment between the model's confidence in its predictions and the actual correctness of those predictions. Well-calibrated models provide confidence scores that accurately reflect their probability of correctness, enabling more decision-making. ECE is a common metric to quantify miscalibration, and measures the difference between *predicted confidence* and *actual accuracy* across multiple confidence intervals

$$\text{ECE} = \sum_{m=1}^{M} \frac{|\mathcal{B}_m|}{N_{\text{Samples}}} \left| \text{Accuracy}(\mathcal{B}_m) - \text{Confidence}(\mathcal{B}_m) \right|, \tag{38}$$

where $M$ is the number of bins, $\mathcal{B}_m$ is the set of samples whose confidence scores fall into the $m$-th bin, $|\mathcal{B}_m|$ denotes the number of samples in bin $\mathcal{B}_m$, $N_{\text{Samples}} = \sum_{m=1}^{M} |\mathcal{B}_m|$ is the total number of samples, $\text{Accuracy}(\mathcal{B}_m)$ and $\text{Confidence}(\mathcal{B}_m)$ are the empirical accuracy and average confidence of the model being evaluated in bin $m$ respectively. Lower ECE indicates better model calibration.

To measure ECE, we use $M = 21$ bins uniformly partitioned across the output probability space. Accuracy and confidence are computed in the standard manner: the predicted label is determined via the argmax over the output probabilities for each prediction, and the confidence is defined as the maximum probability assigned to the predicted label. Accuracy is then measured as the proportion of instances where the predicted label matches the ground truth. Notably, this approach focuses solely on the maximum probability prediction, disregarding the calibration of lower-probability predictions. To assess calibration across the entire output distribution rather than just the top prediction, alternative metrics could be considered.

### E.8.1. TEACHERS

In Figure 42, we see the ECE for different sizes of teachers. For all models, ECE is between $0.4\%$ and $0.6\%$, suggesting that the models' confidence estimates closely align with their actual accuracies. We also observe that the blue points, i.e. , the teacher's actual accuracy for predictions falling into specific confidence intervals, closely follow the diagonal, indicating that the models are well-calibrated. This well-calibrated nature can be surprising, as large models can be overconfident. For example, Mukhoti et al. (2020) indicates the overconfidence of large models observed in (Minderer et al., 2021) arises from overfitting, regardless of the training set correctness.

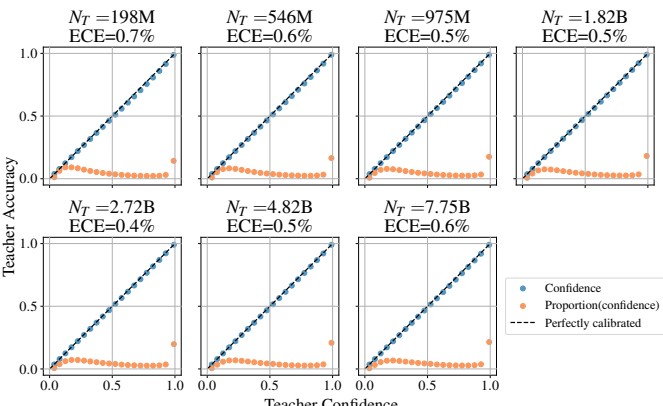

*Figure 42.* **Teacher calibration.** The calibration of teachers of seven different sizes. The $x$-axis shows the teacher probability assigned to the most confident class, and the $y$-axis is the empirical accuracy of predictions within each confidence bin. Blue points represent the teacher accuracy for predictions falling into specific confidence intervals. Orange points represent the proportion of samples in each confidence bin (helpful for understanding sample distribution across confidence levels). The dashed line represents perfect calibration, where confidence matches empirical accuracy. The ECE (Equation 38) for each teacher is shown as the title of each plot.

The primary distinction in our setup is that: i) our models are underparameterized ($N < D$), and ii) data is not repeated. Consequently, overfitting to the training set does not occur (Aitchison, 2024), so model overconfidence does not arise to the same extent as in many prior calibration studies. *Instead*, in our setting, increasing model size $N$ or training tokens $D$, improves the approximation of the seen distribution with minimal generalization gap, yielding better calibration (Carrell et al., 2022; Blasiok et al., 2023). Our observation of good calibration in large models aligns with prior calibration findings

for language model calibration (Zhu et al., 2023; Kadavath et al., 2022; OpenAI, 2023).

### E.8.2. 198M STUDENTS TRAINED ON 20N TOKENS

In this section we consider students trained on the teacher distribution, as in our main study. We also study students trained on the teacher top-1 distribution, as described in Appendix G.4, as the qualitative difference in behavior can be informative for student design.

Evaluating the calibration of a student can be done in a number of ways:

1. We can compare student outputs relative ground-truth data, as in Appendix E.8.1 for the teachers.

2. We can compare student outputs with the outputs of its teacher.

**Calibration against ground-truth.** First, let's consider comparison against ground truth data. In Figure 43 we show student calibration with respect to the dataset labels for both *teacher distribution* distillation and *teacher top-1* distillation.

1. *Distilled on the full teacher distribution.* In Figure 43a, we observe that the student is well-calibrated against ground truth data. Similar to the teacher's calibration plot in Figure 42, we see a small discrepancy at very low and very high confidence values, and the ECE value is low.

2. *Distilled on teacher top-1.* In Figure 43b, we see that *a student trained only on its teacher's top-1 prediction, is not calibrated against ground truth data.* The blue points below the dashed line indicate an overconfident student, i.e. , its predicted confidence is higher than the actual accuracy in that confidence range. This is because training the student on top-1 assigns the student to the most plausible outcome rather than all the plausible outcomes with correct frequencies. Confidence proportions are low for all bins that are not the most confident bin, and ECE is high, although decreases with increasing teacher size $N_T$.

Figure 43 shows that training the student on the teacher's distribution results in a calibrated student, whereas training on the teacher top-1 does not. Indeed, optimizing against the teacher's top-1 is not a proper scoring metric, and that teacher top-1 is *not* an *unbiased* estimator for the data, while the teacher distribution is.

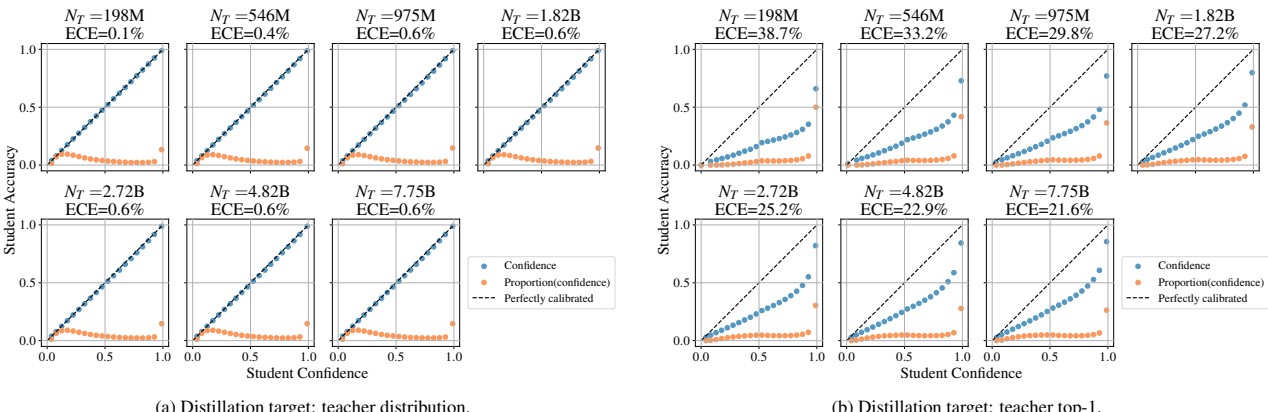

*Figure 43.* **Student calibration (data).** Calibration of the student with respect to the actual data labels, trained with different teacher sizes ($N_T$), on **(a)** the teacher distribution and **(b)** the teacher's top-1. For axis definitions and the figure legend, refer to Figure 42. Blue points below the dashed line indicate student overconfidence.

**Calibration against teacher top-1.** Next we investigate the first student calibration against the teacher. In Figure 44 we show student calibration with respect to the teacher's top-1 label. That is, the next-token label used for accuracy computation, and extract the students confidence is the most probable next-token according to the teacher, instead of the label from data. Here no next token labels are used at all. These teacher top-1 labels are also used for the ECE calculation, which is still computed using Equation 38.

1. *Distilled on the full teacher distribution.* We see in Figure 44a that when distilled from the full teacher distribution, the student is *not* calibrated against the teacher top-1. The blue points are above the dashed line, which means that the empirical accuracy is higher than the model's predicted confidence, i.e. with respect to the teacher top-1, the student is *underconfident*. This can be understood by noting that the top-1 objective is an easier objective than modeling the full vocabulary at each step.

2. *Distilled on teacher top-1.* In Figure 44b we observe that a student is distilled from its teacher's top-1 *is calibrated with respect to teacher's top-1*.

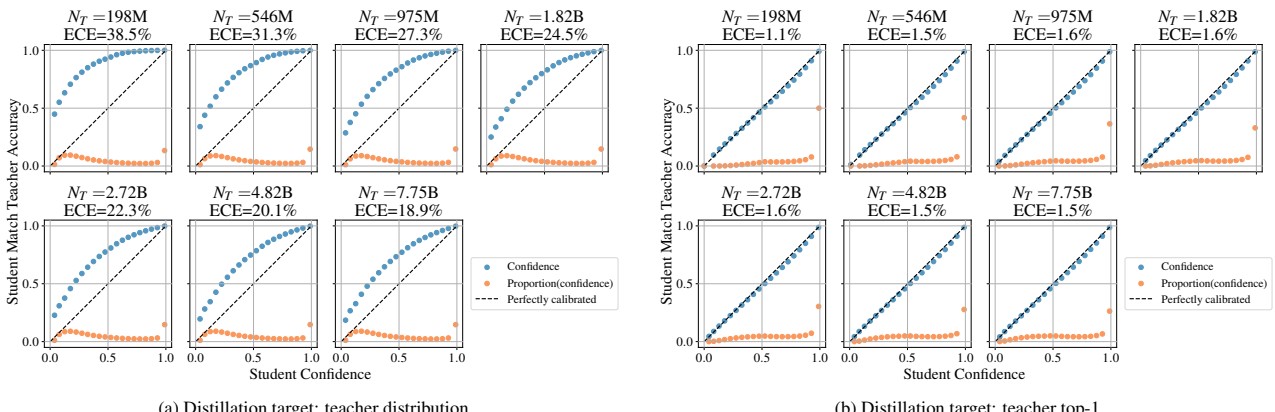

(a) Distillation target: teacher distribution.    (b) Distillation target: teacher top-1.

*Figure 44.* **Student calibration (teacher top-1).** Calibration of the student with respect to the teacher's top 1, trained with different teacher sizes ($N_T$), on **(a)** the teacher distribution and **(b)** the teacher's top-1. For axis definitions and the figure legend, refer to Figure 42. Blue points above the dashed line indicate the student is *underconfident*.

Figure 44 shows that training the student on teacher top-1 results in calibration against teacher top-1, whereas a model trained on data, or distilled on the full teacher distribution is not calibrated against teacher top-1. As above, this can be understood as now teacher's top-1 is now a proper scoring metric, and teacher top-1 is an unbiased estimator for itself.

**Calibration against teacher distribution.** Here we develop a modified calibration measure that will help us understand if the student matches the teacher in a distributional sense. As we have two distributions to compare, we can ask, for a given teacher confidence, what is the expected student confidence. This leads to $\text{ECE}_{\text{Dist}}$, a distributional form of ECE:

$$\text{ECE}_{\text{Dist}}(A, B) = \sum_{m=1}^{M} \frac{|\mathcal{B}_m|}{N_{\text{Samples}}} \left|\text{Confidence}(\mathcal{B}_m; A) - \text{Confidence}(\mathcal{B}_m; B)\right|, \tag{39}$$

and is similar in spirit to divergence measures like KLD. $\mathcal{B}_m$, $|\mathcal{B}_m|$, and $N_{\text{Samples}}$ are defined as before, and $\text{Confidence}_S(\mathcal{B}_m; A|B)$ is the average confidence of model $A$ or $B$ in bin $m$ respectively. The bins $\mathcal{G}_m$ are always witin the bins of confidence of model $B$. In the current evaluation, we take $A$ as the teacher and $B$ as the student, and we are measuring the average confidence of the teacher is measured within a student's confidence bin.

1. *Distilled on the full teacher distribution.* In Figure 45a, we see that when the student is confident, it matches the teacher confidence. However, as the teacher model grows in size, when the student is less confident, it it systematically underestimates its confidence. This suggests that the student has not effectively learned low-probability outcomes, or that these outcomes are particularly challenging for the student to replicate. The underconfidence in these regions may be a result of the distillation process not providing sufficient learning signal for these difficult cases, or the inherent difficulty of capturing the uncertainty associated with low-confidence predictions. This observation of confidence mismatch helps indicate which parts of the distribution the student finds challenging to model, giving rise to the increasing KLD and capacity gap observed in Figure 4 and Appendix E.3.

2. *Distilled on teacher top-1.* In Figure 45b, for small teachers, we observe student overconfidence. As the teacher increases in size, the student's overconfidence in low-confidence bins transitions to underconfidence. At the same time,

the student's overconfidence in high-confidence bins improves, leading to an overall reduction in distributional ECE. This pattern of overconfidence in the student is similar to what we saw in Figure 43b, but the change in behavior at low-confidence bins as the teacher's size varies is different. This shift in the student's calibration behavior, especially in low-confidence bins, aligns with findings from Figure 45a and may highlight the difficulty the small student faces in learning rare events.

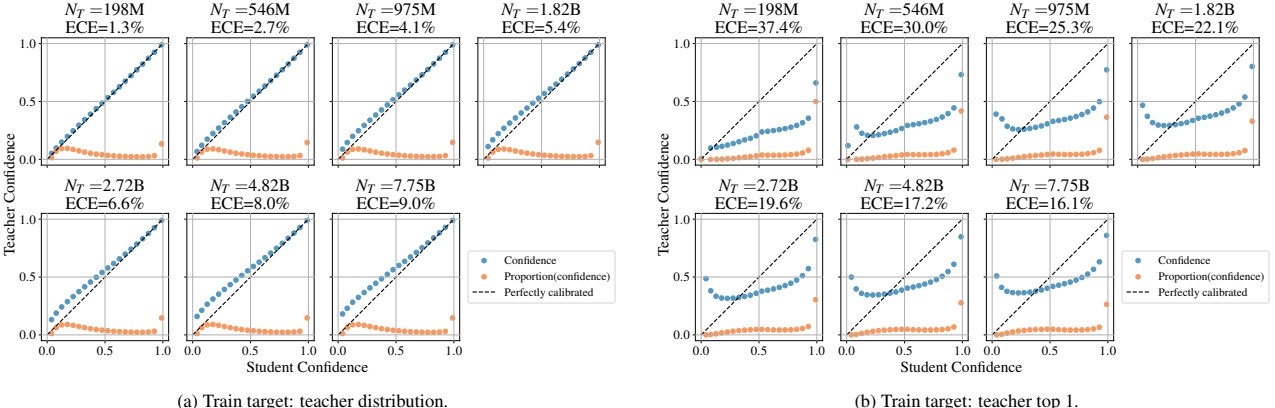

(a) Train target: teacher distribution.    (b) Train target: teacher top 1.

*Figure 45.* **Student calibration (teacher distribution).** Calibration of the student with respect to the teacher's distribution, trained with different teacher sizes ($N_T$), on **(a)** the teacher distribution and **(b)** the teacher's top-1. For ECE calculation on the full distribution, see Equation 39. For axis definitions and the figure legend, refer to Figure 42. Blue points below the dashed line indicate student overconfidence, while points above the dashed line indicate underconfidence.

We can also inspect the student confidences within a bin of teacher confidences, and compute the distributional ECE (Equation 39), swapping the roles of teacher and student (see Figure 46).

1. *Distilled on the full teacher distribution.* In Figure 45a we complete the picture from Figure 45a and see that the part of the distribution the student struggles to model is actually the place where teacher is most confident.

2. *Distilled on teacher top-1.* In Figure 45b we see that the student is systematically overconfident for all values of teaacher confidence, except for the largest teachers, where the student is underconfident when those teachers are most confident.

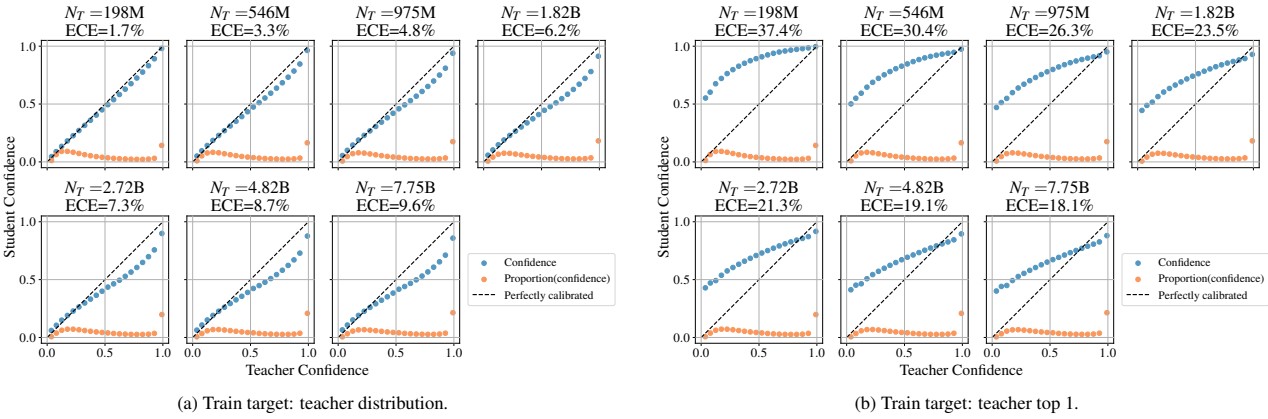

(a) Train target: teacher distribution.    (b) Train target: teacher top 1.

*Figure 46.* **Student calibration (under teacher confidence bins).** Calibration of the student with respect to the teacher's confidence bins, trained with different teacher sizes ($N_T$), on **(a)** the teacher distribution and **(b)** the teacher's top-1. For ECE calculation on the full distribution, see Equation 39. For axis definitions and the figure legend, refer to Figure 42. Blue points below the dashed line indicate the teacher is less confident than the student.

### E.8.3. 198M STUDENTS TRAINED ON 128B TOKENS

In this section, we study the effect of increasing the number distillation tokens in Appendix E.8.2 from $D_S \approx 20N_S$ to $D_S \approx 512B$. Here, we reserve discussion for the observed differences compared to Appendix E.8.2.

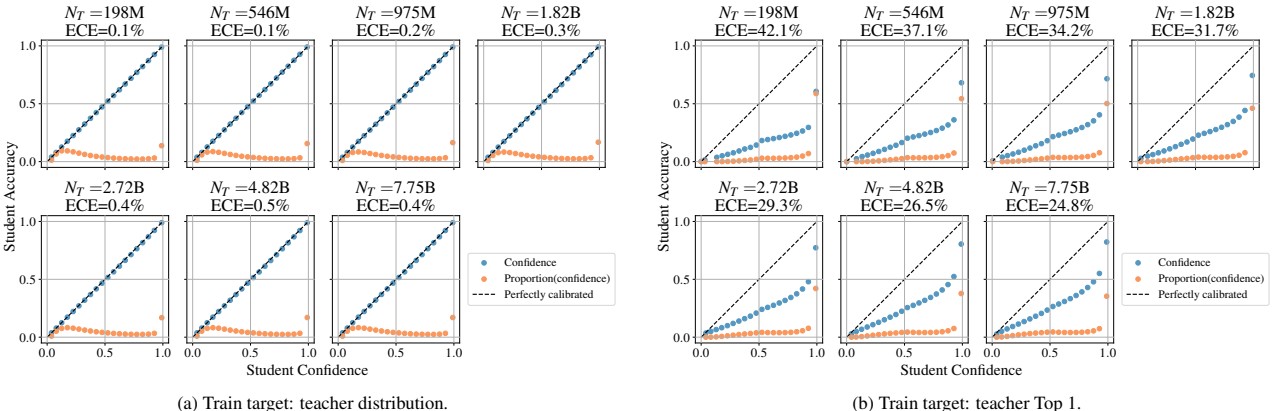

(a) Train target: teacher distribution.

(b) Train target: teacher Top 1.

*Figure 47.* **Student calibration (data).** Calibration of the student with respect to the actual data labels with increased training tokens. Compare to Figure 43 for the effect of tokens and refer to Figure 42 for legend and axis explanations.

**Calibration against ground-truth.** As the number of distillation tokens increases, we observe a consistent decrease in the ECE when the student is trained on the teacher's distribution, as shown by the comparison between Figure 47a and Figure 43a across different teacher sizes. However, when the student is trained on the teacher's top-1 predictions, increasing the number of tokens *negatively* impacts ECE, as evidenced by the comparison between Figure 47b and Figure 43b. This suggests that the teacher's top-1 predictions are not a reliable, unbiased estimator of the actual data, and increasing the number of training tokens only exacerbates this issue. See Appendix G.4 for further discussion.

**Calibration against teacher top-1.** Increasing the number of distillation tokens leads to worse calibration between the student and the teacher's top-1 predictions when the student is trained on the full distribution. This change primarily occurs in the low-confidence bins, and results in a higher ECE (compare Figure 48a and Figure 44a). However, when comparing the ECEs for the student trained on the teacher's top-1 predictions (Figures 44b and 48b), there is an improvement across all teacher sizes. When the student is trained and evaluated using the same metric, increasing the training tokens helps improve calibration, demonstrating consistency between the learning objective and the evaluation metric.

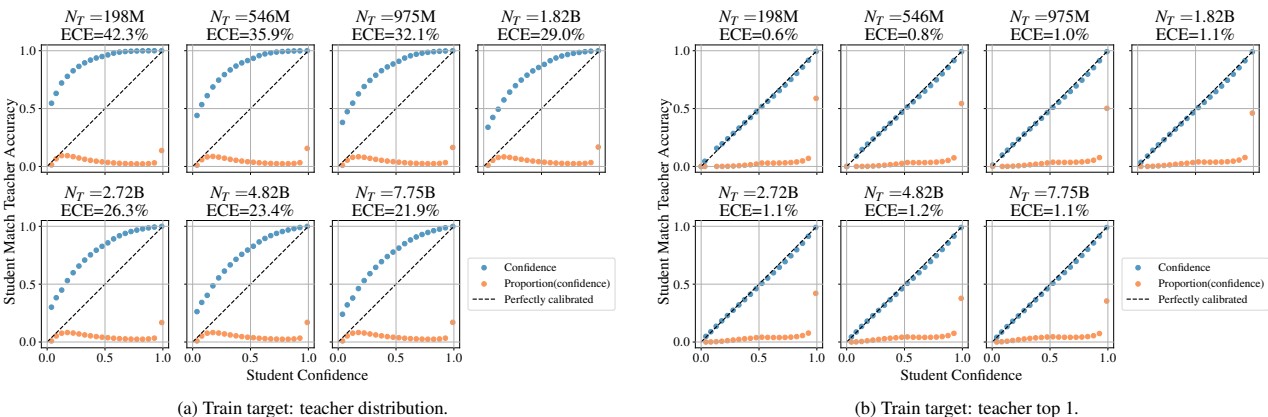

(a) Train target: teacher distribution.

(b) Train target: teacher top 1.

*Figure 48.* **Student calibration (teacher top 1).** Calibration of the student with respect to the teacher's top 1 when the training tokens have increased. Compare to Figure 44 for the effect of tokens and refer to Figure 42 for legend and axis explanations.

**Calibration against teacher distribution.** A comparison between Figure 49a and Figure 45a shows that when the student is trained on the teacher's full distribution and evaluated against the full distribution using Equation 39, increasing the number of training tokens consistently improves calibration across all teacher sizes. However, when the student is trained on the teacher's top-1 predictions, a quick comparison between Figure 49b and Figure 45b reveals worse calibration uniformly across all confidence bins.

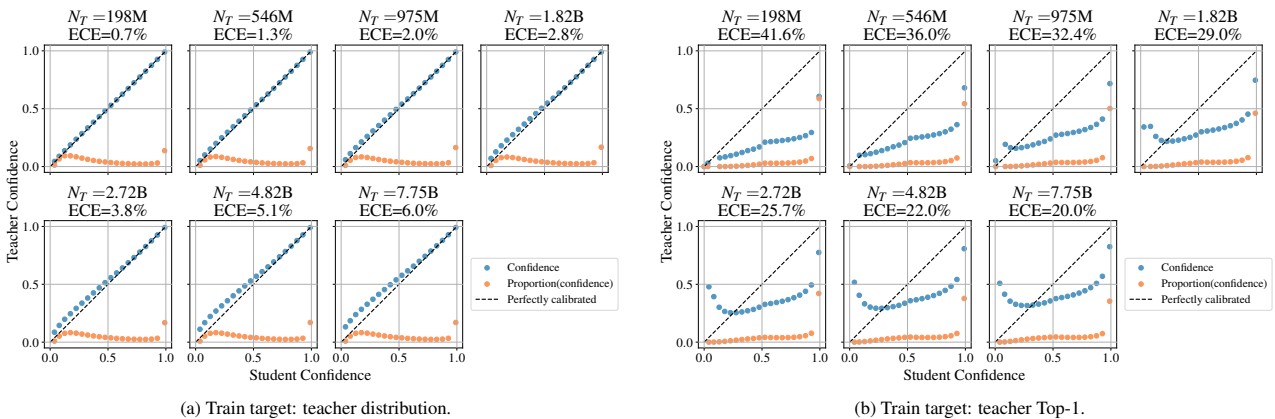

(a) Train target: teacher distribution.  (b) Train target: teacher Top-1.

*Figure 49.* **Student calibration (teacher distribution).** Calibration of the student with respect to the teacher's distribution as the number of training tokens increases. Compare to Figure 45 for the effect of tokens and refer to Figure 42 for legend and axis explanations.

Similarly, when comparing within teacher confidence bins (Figure 50) increasing the number of distillation tokens from 20N to 128B primarily amplifies the observed phenomena at lower distillation token budgets, and improving calibration in cases where there is a proper scoring metric present (Figure 50a).

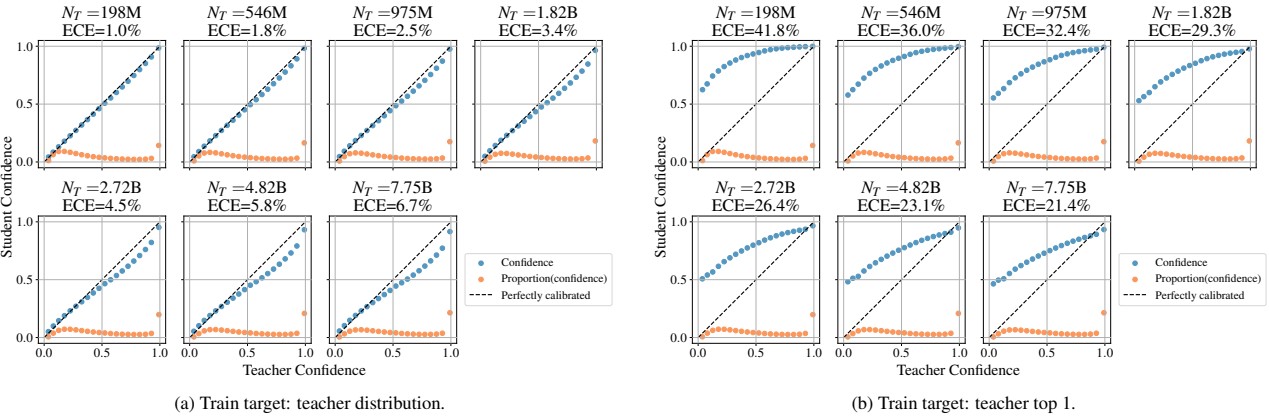

(a) Train target: teacher distribution.  (b) Train target: teacher top 1.

*Figure 50.* **Student calibration (teacher distribution).** Calibration of the student with respect to the teacher' confidence bins distribution as the number of training tokens increases. Compare to Figure 46 for the effect of tokens.

In general, increasing the number of training tokens has a positive effect when the training metric is an unbiased estimator of the actual data or the measured calibration quantities (see Figures 47a, 48b, and 49a) and reduces the ECE, while it has a negative impact when there is a mismatch between the learned and measured quantities (see Figures 47b, 48a, and 49b).

# F. Scaling coefficients

In this section, we analyze the process of deriving the coefficients for our scaling law. We follow the procedure outlined in (Hoffmann et al., 2022; Besiroglu et al., 2024), while incorporating our modified scaling laws

## F.1. Supervised scaling law coefficient estimation

First, let's tackle the supervised scaling law Equation 1 restated for convenience

$$L(N, D) = E + \left( \frac{A}{N^\alpha} + \frac{B}{D^\beta} \right)^\gamma. \tag{40}$$

To aid numerical stability, we write this expression in log space. First note that for $a, b > 0$

$$\log(a + b) = \log\left(\exp\log a + \exp\log b\right) = \mathrm{LSE}(\log a, \log b), \tag{41}$$

where LSE is the log-sum-exp operator. We can now proceed to write the supervised scaling law in log form

$$\log L(N, D; A, B, E, \alpha, \beta) = \log\left[ E + \left( \frac{A}{N^\alpha} + \frac{B}{D^\beta} \right)^\gamma \right] \tag{42}$$

$$= \mathrm{LSE}\left[ \log E, \gamma \log\left( \frac{A}{N^\alpha} + \frac{B}{D^\beta} \right) \right] \tag{43}$$

$$= \mathrm{LSE}\left[ \log E, \gamma\, \mathrm{LSE}\left( \log A - \alpha N, \log B - \alpha D \right) \right]. \tag{44}$$

We make no assumptions about the relationships between the values (i.e. *no parameter tying*) and optimize

$$(A^*, B^*, E^*, \alpha^*, \beta^*, \gamma^*) = \underset{\{A, B, E, \alpha, \beta, \gamma\}}{\arg\min} \sum_i \mathrm{Huber}_\delta\left( \log L(N^{(i)}, D^{(i)}; A, B, E, \alpha, \beta) - L^{(i)} \right) \tag{45}$$

with a Huber $\delta = 10^{-4}$, where $N^{(i)}$, $D^{(i)}$ and $L^{(i)}$ are the model size, number of training tokens and loss achieved by the $i$-th run. We fit on 73 samples over a grid of L-BFGS-B initializations given by: $\log A \in \{0., 5., 10., 15., 20.\}$, $\log B \in \{0., 5., 10., 15., 20.\}$, $\log E \in \{-1., -0.5., 0., 0.5, 1., 1.5.\}$, $\alpha \in \{0., 0.5, 1., 1.5\}$, $\beta \in \{0., 0.5, 1., 1.5\}$, $\gamma \in \{0., 0.5, 1., 1.5\}$. The $L \geq 2.2$ case corresponds to 48 samples.

## F.2. Distillation scaling law coefficient estimation

Next, let's address the distillation scaling law Equation 8 restated for convenience

$$L_S(N_S, D_S, L_T) = L_T + \frac{1}{L_T^{c_0}} \left( 1 + \left( \frac{L_T}{\widetilde{L}_S d_1} \right)^{1/f_1} \right)^{-c_1 * f_1} \left( \frac{A'}{N_S^{\alpha'}} + \frac{B'}{D_S^{\beta'}} \right)^{\gamma'}. \tag{46}$$

As in Appendix F.1, to aid numerical stability during optimization, we write this in log space

$$\log L_S(N_S, D_S, L_T; \theta) = \log\left[ L_T + \frac{1}{L_T^{c_0}} \left( 1 + \left( \frac{L_T}{\widetilde{L}_S d_1} \right)^{1/f_1} \right)^{-c_1 * f_1} \left( \frac{A'}{N_S^{\alpha'}} + \frac{B'}{D_S^{\beta'}} \right)^{\gamma'} \right] \tag{47}$$

$$= \mathrm{LSE}\left[ \log L_T, -c_0 \log L_T - c_1 f_1 \log\left( 1 + \left( \frac{L_T}{d_1 \widetilde{L}_S} \right)^{1/f_1} \right) + \gamma \log\left( \frac{A'}{N_S^\alpha} + \frac{B'}{D_S^\beta} \right) \right] \tag{48}$$

$$= \mathrm{LSE}\left[ \log L_T, \left( -c_0 \log(L_T) - c_1 f_1\, \mathrm{LSE}\left( 0, \frac{1}{f_1}\left( \log L_T - \log \widetilde{L}_S - \log d_1 \right) \right) \right. \right.$$

$$\left. \left. + \gamma\, \mathrm{LSE}\left( \log A' - \alpha' \log N_S, \log B' - \beta' \log D_S \right) \right) \right], \tag{49}$$

where $\theta = \{A', B', \alpha', \beta', c_0, c_1, f_1, d_1\}$. We make no assumptions about the relationships between the values and optimize

$$\theta^* = \arg\min_\theta \sum_i \text{Huber}_\delta \left( \log L_S(N_S^{(i)}, D_S^{(i)}, L_T^{(i)}; \theta) - L_S^{(i)} \right) \tag{50}$$

with a Huber $\delta = 10^{-4}$, where $N_S^{(i)}$, $D_S^{(i)}$, $L_T^{(i)}$ and $L_S^{(i)}$ are the student model size, number of training distillation tokens, the teacher pretraining loss and the student validation loss on the data achieved by the $i$-th run. We fit on 697 samples over a grid of L-BFGS-B initializations given by: $\log A' \in \{0., 5., 10., 15., 20.\}$, $\log B' \in \{0., 5., 10., 15., 20.\}$, $\alpha' \in \{0., 0.5, 1.\}$, $\beta' \in \{0., 0.5, 1.\}$, $\gamma' \in \{0., 0.5, 1.\}$, $c_0 \in \{0., 0.5, 1., 1.5\}$, $c_1 \in \{0., 0.5, 1., 1.5\}$, $f_1 \in \{0., 0.5, 1., 1.5\}$, $\log d_1 \in \{-1., -0.5, 0., 0.5, 1.\}$. The $L_S \geq 2.3$ case corresponds to 551 samples.

### F.3. Scaling law coefficients parameteric fit

The fitting procedure outlined in Appendices F.1 and F.2 applied to data described in Section 4.2 yields the scaling coefficients and associated confidence intervals shown in Table 6. Note in the supervised case, our values of $a$ and $b$ are consistent with those of Hoffmann et al. (2022).

*Table 6.* Scaling law parameter estimates accompanied by 90% confidence intervals obtained by bootstrapping (4096 resamples) following the procedure of Besiroglu et al. (2024). $a = \beta/(\alpha + \beta)$ and $b = \beta/(\alpha + \beta)$ are the supervised compute optimal scaling estimates for $N$ and $D$ respectively (Hoffmann et al., 2022).

|              | Supervised            | Distillation          |
| ------------ | --------------------- | --------------------- |
| $A^{(\prime)}$    | 3355 (3346, 3360)     | 2243 (2227, 2255)     |
| $B^{(\prime)}$    | 18186 (18157, 18236)  | 24181 (24084, 24266)  |
| $E$          | 1.220 (1.190, 1.247)  |                       |
| $\alpha^{(\prime)}$ | 0.408 (0.405, 0.411)  | 0.321 (0.319, 0.324)  |
| $\beta^{(\prime)}$  | 0.431 (0.428, 0.433)  | 0.637 (0.634, 0.640)  |
| $\gamma^{(\prime)}$ | 0.452 (0.442, 0.461)  | 0.764 (0.732, 0.788)  |
| $c_0$        |                       | 2.549 (2.425, 2.615)  |
| $c_1$        |                       | 522.6 (522.6, 522.6)  |
| $f_1$        |                       | 0.090 (0.088, 0.093)  |
| $d_1$        |                       | 1.315 (1.302, 1.327)  |
| $a^{(\prime)}$    | 0.513 (0.513, 0.513)  | 0.664 (0.662, 0.665)  |
| $b^{(\prime)}$    | 0.486 (0.486, 0.486)  | 0.335 (0.334, 0.337)  |
| Runs         | 73                    | 697                   |

We also note that our irreducible error term is lower than the one in Hoffmann et al. (2022). We suspect this is due to our use of $\mu$P (Yang & Hu, 2021; Yang & Littwin, 2023; Yang et al., 2022; Wortsman et al., 2023; Yang et al., 2023).

## G. Distilling language models in practice

In the following analyses, we explore the sensitivity of student performance under modification of distillation hyperparameters. We demonstrate that the pure distillation setting ($\lambda = 1$, Appendix G.1), unit temperature ($\tau = 1$, Appendix G.2), and learning rate $\eta = 0.01$ (Appendix G.3) under $\mu$P (Yang & Hu, 2021; Yang & Littwin, 2023; Yang et al., 2022; Wortsman et al., 2023; Yang et al., 2023) provides robust performance across model scales, while distribution truncation methods (Top-$k$, Top-$p$) degrade performance *unless combined with ground-truth next-token prediction* (Appendix G.4). Finally, we verify that forward KL divergence distillation, $D_{\text{KL}}(\hat{p}_T || \hat{q}_S)$, consistently outperforms reverse KL (Appendix G.5).

For ease of reference, we restate the components of the token-level loss for the student:

$$\mathcal{L}_{\text{NTP}}(x^{(i)}, \boldsymbol{z}^{(i)}) = -\sum_{a=1}^{V} \boldsymbol{e}(x^{(i)})_a \log \sigma_a(\boldsymbol{z}^{(i)}), \qquad \text{(Next-token prediction)} \quad (51)$$

$$\mathcal{L}_Z(\boldsymbol{z}^{(i)}) = \|\log Z(\boldsymbol{z}^{(i)})\|_2^2 = \left\|\log \sum_{a=1}^{V} \exp(z_a^{(i)})\right\|_2^2, \qquad \text{(Z-loss)} \quad (52)$$

$$\mathcal{L}_{\text{KD}}(\boldsymbol{z}_T^{(i)}, \boldsymbol{z}_S^{(i)}) = -\tau^2 \sum_{a=1}^{V} \sigma_a\left(\frac{\boldsymbol{z}_T^{(i)}}{\tau}\right) \log \sigma_a\left(\frac{\boldsymbol{z}_S^{(i)}}{\tau}\right), \qquad \text{(Distillation loss)} \quad (53)$$

$$\mathcal{L}_S(x^{(i)}, \boldsymbol{z}_T^{(i)}, \boldsymbol{z}_S^{(i)}) = (1-\lambda)\,\mathcal{L}_{\text{NTP}}(x^{(i)}, \boldsymbol{z}_S^{(i)}) + \lambda\,\mathcal{L}_{\text{KD}}(\boldsymbol{z}_T^{(i)}, \boldsymbol{z}_S^{(i)}) + \lambda_Z\,\mathcal{L}_Z(\boldsymbol{z}_S^{(i)}). \qquad \text{(Student loss)} \quad (54)$$

See Section 2 for a discussion of each of the terms.

### G.1. Mixing coefficient ($\lambda$) sensitivity analysis

The distillation process combines two loss components: knowledge transfer from the teacher, $\lambda\mathcal{L}_{\text{KD}}(\boldsymbol{z}_T^{(i)}, \boldsymbol{z}_S^{(i)})$, and direct learning from data, $(1-\lambda)\mathcal{L}_{\text{NTP}}(x^{(i)}, \boldsymbol{z}_S^{(i)})$, weighted by the mixing coefficient $\lambda$ (Equation 7). Our distillation scaling law analysis is performed in the *pure distillation* setting ($\lambda = 1$). Here we show this simple choice provides robust performance across a wide range of configurations.

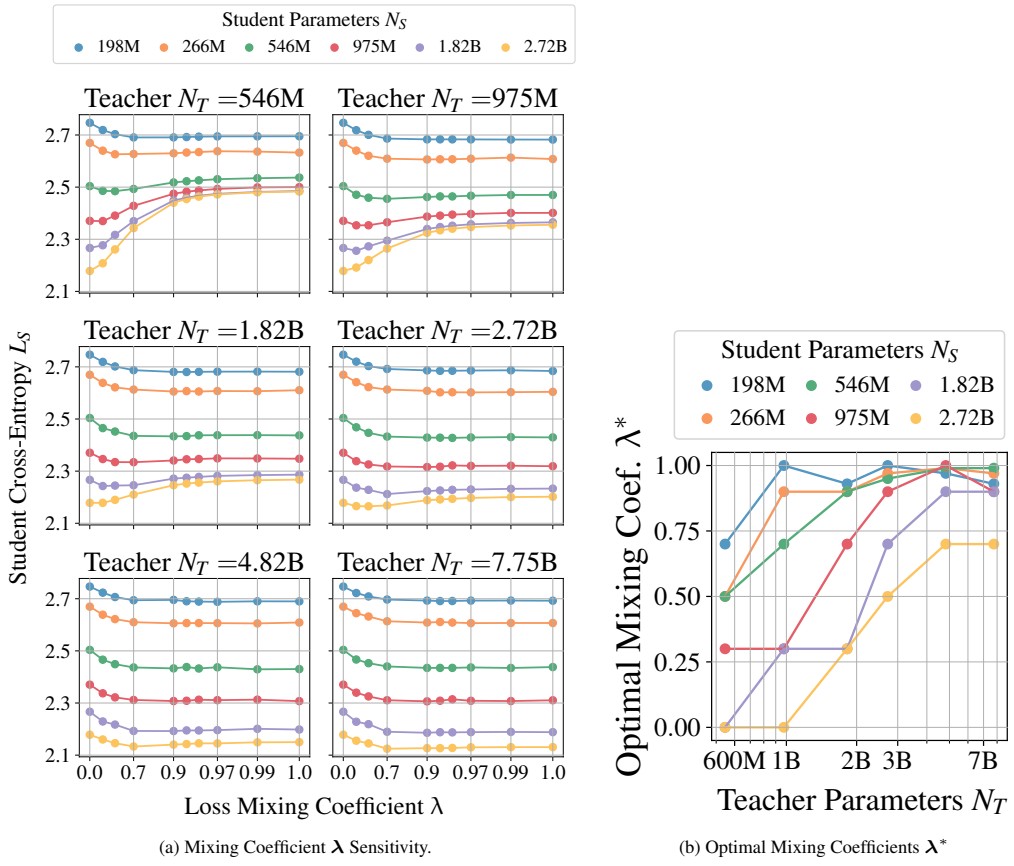

(a) Mixing Coefficient $\boldsymbol{\lambda}$ Sensitivity.

(b) Optimal Mixing Coefficients $\boldsymbol{\lambda}^*$

*Figure 51.* **Mixing Coefficients $\boldsymbol{\lambda}$.** **(a)** Students of six sizes $N_S \in \{198M, 266M, \ldots, 2.72B\}$ trained with a $M = D_S/N_S = 20$ ratio are distilled from teachers of size sizes $N_T \in \{546M, 975M, \ldots, 7.75B\}$ trained with a $M = D_T/N_T = 20$ ratio with different values of loss mixing coefficient $\lambda \in [0,1]$. $\lambda = 0$ and $\lambda = 1$ correspond to supervised training and pure distillation cases respectively. **(b)** The mixing coefficients $\lambda^* = \arg\min_\lambda \mathcal{L}(\lambda)$ that give the lowest student validation loss for each teacher-student combination shown in Figure 51a.

We examine various $\lambda$ values across different teacher-student configurations in Figure 51a and find that while the optimal mixing coefficients $\lambda^*$ vary based on the specific teacher-student combinations (Figure 51b), the student cross-entropy $L_S$ remains mostly flat for choices of $\lambda > 0.5$, with lower values of $\lambda$ only preferred in the cases where the teacher is particularly weak and where the supervised signal is more informative. From Figure 51a it is also possible to get a sense of when distillation $\lambda > 0$ generally outperforms supervised learning $\lambda = 0$ under the same token budget.

To guide practitioners, Figure 51b shows empirically derived optimal mixing coefficients, $\lambda^*$, though the simplicity and robustness of pure distillation makes it a reliable default choice for practical use and study.

### G.2. Temperature ($\tau$) sensitivity analysis

In distillation, the temperature $\tau$ controls the entropy of teacher predictions by scaling logits $z_T^{(i)}/\tau$ and $z_S^{(i)}/\tau$ in the knowledge distillation loss $\mathcal{L}_{\text{KD}}$ (Equations 7 and 53). This scaling modulates the transfer of *dark knowledge* (Hinton et al., 2015) – the log-probability ratios between incorrect categories encode the teacher's understanding of relationships between those categories. Our analysis across $\tau \in [0.5, 10]$ (Figure 52) reveals that higher temperatures ($\tau > 3$) reduces performance by attenuating these ratios in $\sigma_a(z_T^{(i)}/\tau)$, particularly harming smaller students that rely heavily on this signal. Lower temperatures ($\tau < 1$) similarly reduce effectiveness by concentrating probability mass on argmax tokens, diminishing the transfer of relationships between lower-ranked predictions.

We find optimal performance at $\tau = 1$ across all model scales, suggesting this temperature best preserves log-probability structure. Unlike the original distillation setting, which relied on dark knowledge to represents hierarchical relationships between incorrect classification predictions in the presence of a *true label*, language modeling is inherently ambiguous and complex, with many valid continuations. *It is precisely the understanding of the ambiguity of language we want to transfer to the student*, which is supported by our finding that maintaining the teacher's original probability ratios ($\tau = 1$) produces the lowest student cross-entropies.

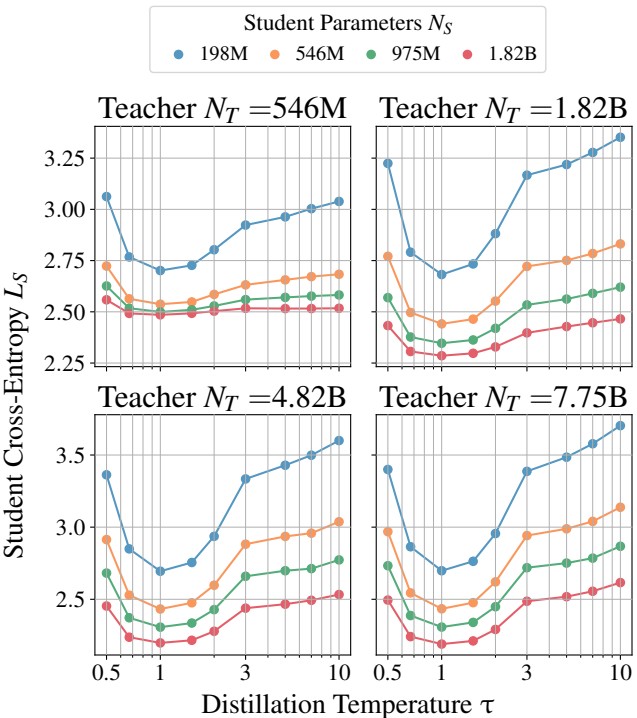

*Figure 52.* **Temperature $\tau$ Sensitivity Analysis.** Students of four sizes $N_S \in \{198M, 546M, 975M, 1.82B\}$ trained with a $M = D_S/N_S = 20$ ratio are distilled from teachers of sizes $N_T \in \{546M, 1.82B, 4.82B, 7.75B\}$ trained with a $M = D_T/N_T = 20$ ratio with different distillation temperatures $\tau \in [0.5, 10]$.

### G.3. Learning rate ($\eta$) sensitivity analysis, verification of $\mu$P for distillation

The peak learning rate $\eta$ determines the scale of student parameter updates in distillation. In our experiments we use a simplified version of $\mu$P (Yang & Hu, 2021; Yang & Littwin, 2023; Yang et al., 2022; Wortsman et al., 2023; Yang et al., 2023), described as $\mu$P (simple) in (Wortsman et al., 2024).

In the supervised case, in addition to improving the performance lower bound compared to the standard parameterization, $\mu$P simplifies experimental settings as it enables *hyperparameter transfer*; the optimal peak learning rate $\eta$ and initialization scales found for a reference model size can be reused when changing model size[7].

Here we validate that the optimal peak learning rate $\eta^* = 0.01$ determined in the supervised case transfers to the distillation setting. Sweeping values $\eta \in [0.001, 0.1]$ (Figure 53) reveals that $\mu$P achieves optimal performance at $\eta = 0.01$ uniformly across all configurations, from $198M$ to $1.82B$ parameter students and $546M$ to $7.75B$ parameter teachers, consistent with the optimal peak learning rate in the supervised setting.

Performance varies smoothly and modestly around this optimum, with cross-entropy changing by less than 0.1 nats over one order of magnitude in learning rate. This consistency validates $\mu$P's guarantee of scale-invariant training dynamics for distillation, confirming that our experimental setting for determining our distillation scaling law operates at the optimal learning rate or sufficiently close to it in all of our settings. The observed moderate learning sensitivity in distillation partially alleviates the requirement for careful learning rate tuning, showing that in practice the reference learning rate found in the supervised setting can be safely reused in the distillation setting.

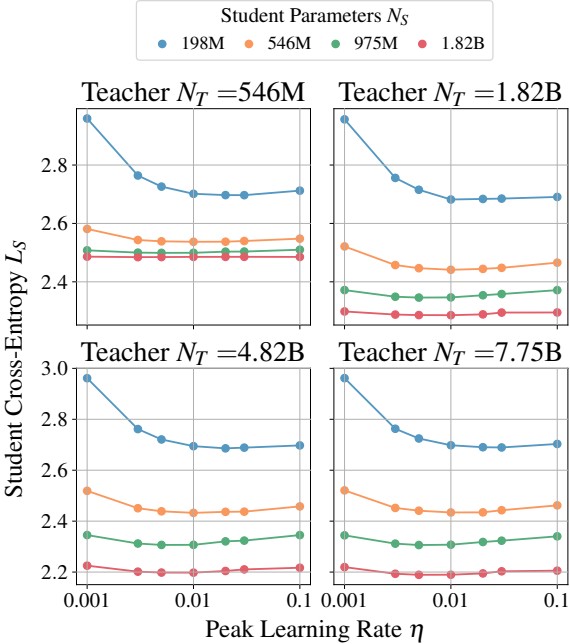

Figure 53. **Learning Rate $\eta$ Sensitivity Analysis.** Students of four sizes $N_S \in \{198M, 546M, 975M, 1.82B\}$ trained with a $M = D_S/N_S = 20$ ratio are distilled from teachers of sizes $N_T \in \{546M, 1.82B, 4.82B, 7.75B\}$ trained with a $M = D_T/N_T = 20$ ratio with different learning rates $\eta \in [0.001, 0.1]$.

### G.4. Distribution truncation methods: Top-$k$ and Top-$p$ sensitivity

We investigate how the truncation of the teacher distributions affects student performance. For these methods, when the teacher produces a distribution $\hat{p}_T(x^{(i)} = a|\boldsymbol{x}^{(<i)})$, $a \in \{1, \dots, V\}$ over the vocabulary for the student to match, only some entries in the distribution are used. This is done primarily to reduce repeated inference and storage costs in the case teacher outputs are being stored for re-use in the multiple distillations scenario discussed in Section 5.3. In our case, the

---

[7]$\mu$P only guarantees learning rate optimality when varying widths. Empirically, the learning rate is also stable when changing the model depth within a reasonable range (Yang et al., 2022). To *guarantee* transfer *across model depths* one can additionally employ depth-$\mu$P (Yang et al., 2024), although we do not use depth-$\mu$P here.

vocabulary size $V = 32168$, so assuming storage in `float32`, means each token requires $32168 \times 4\,\text{bytes} \approx 129\text{KB}$, and storing all of C4 (approximately 2T tokens) would take approximately 260 Petabytes, a significant amount of data, roughly the total amount collected during the first ten years of the Large Hadron Collider (LHC) (CERN, 2018).

Given a truncation method $\mathcal{M}$, can a *truncated* teacher output $\hat{p}_T^{(\mathcal{M})}$ can be stored whilst still achieving the gains of distillation? Concretely, the truncation $p^{(\mathcal{M})}(x|c)$ of a distribution $p(x|c)$ with a truncation method $\mathcal{M}$ is

$$p^{(\mathcal{M})}(x=a|c) = \begin{cases} \frac{p(x=a|c)}{\sum_{b\in\mathcal{S}_{\mathcal{M}}} p(x=b|c)}, & a\in\mathcal{S}_{\mathcal{M}}(p(\cdot|c)), \\ 0, & \text{otherwise}, \end{cases} \quad (55)$$

where $\mathcal{S}_{\mathcal{M}}(p(\cdot|c))$ represents the set of retained categories (i.e. non-zero probabilities) in the truncated distribution, which then undergoes renormalization over the retained categories.

We explore two complementary approaches: Top-$k$ and Top-$p$ (nucleus) sampling. As in all of our settings, we evaluate the student cross-entropy against the data distribution with all categories, as this is the model property we are most interested in (a model can trivially match the target distribution if all categories except one are removed).

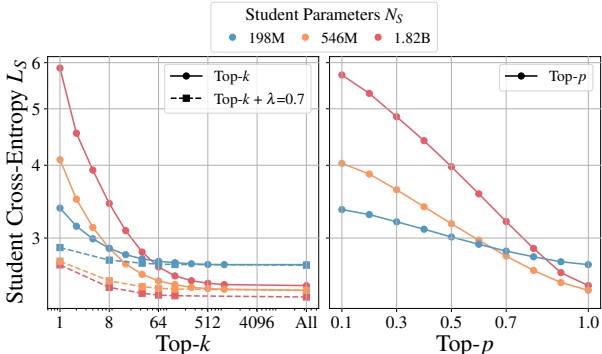

*Figure 54.* **Distribution truncation analysis.** Top-$k$ (left) and Top-$p$ (right) truncation of teacher logits $\boldsymbol{z}_T^{(i)}$ for student-teacher pairs with $N_S$ in $\{198\text{M}, 546\text{M}, 1.82\text{B}\}$ and corresponding $N_T$ in $\{7.75\text{B}, 1.82\text{B}, 546\text{M}\}$. Standard truncation degrades performance: at $k = 128$, validation loss increases by 0.11 nats compared to full distillation ($k = 32768$), while Top-$p$ with $p = 0.9$ degrades by 0.13 nats versus $p = 1.0$. Using $\lambda = 0.7$ with $k = 128$ maintains performance within 0.01 nats while enabling efficient post-hoc training.

For Top-$k$, we *zero-out* all but the largest $k$ probabilities, and Top-$p$, we *zero-out* all but the smallest set of probabilities that sum to at least $p$. The set definitions $\mathcal{S}_{\mathcal{M}}$ for Top-$k$ and Top-$p$ are

$$\mathcal{S}_k(\hat{p}) = \text{Top}(\hat{p},\, k), \qquad\qquad \mathcal{S}_p(\hat{p}) = \{a : \sum_{b\in\text{sort}\downarrow(\hat{p},a)} \hat{p} \le p\}. \quad (56)$$

As the truncation parameters increase ($k \to V$ or $p \to 1$), both methods approach the full teacher distribution, and the student's cross-entropy converges to the baseline using the entire $\hat{p}_T$. Conversely, aggressive truncation (small $k$ or $p$) induces quantization that preserves only high-probability tokens while discarding information in the tail of the distribution.

Our empirical analysis (Figure 54) reveals that both truncation methods directly correlate with reduced evaluation likelihoods. However, this performance degradation can be effectively mitigated through a combination of truncated distributions and ground truth next-token prediction using a mixing coefficient $\lambda \in (0, 1)$ (Equation 7). Specifically, with $k = 128$ and $\lambda = 0.7$, we achieve validation losses statistically indistinguishable from those obtained using the complete teacher distribution. For large-scale distillation scenarios where maintaining multiple models in memory is prohibitive, particularly with large teacher models, storing only the Top-$k$ teacher predictions (with $\lambda > 0$) enables efficient post-hoc distillation.

### G.5. Forward and reverse KL divergence

We investigate both forward (mode spreading) and reverse (mode seeking) Kullback-Leibler divergences for distillation from $N_T = 1.82\text{B}$ to $N_S = 546\text{M}$. The forward KLD $D_{\text{KL}}(\hat{p}_T||\hat{q}_S)$ (Equation 7), minimizes $\mathcal{L}_{\text{forward}} = H(\hat{p}_T, \hat{q}_S) - H(\hat{p}_T)$, where $H(\hat{p}_T)$ is dropped during optimization as it depends on only fixed teacher parameters. In contrast, the reverse KLD $D_{\text{KL}}(\hat{q}_S||\hat{p}_T)$ requires explicitly computing the student's entropy, $\mathcal{L}_{\text{reverse}} = H(\hat{q}_S, \hat{p}_T) - H(\hat{q}_S)$.

The forward KL achieves a lower data cross-entropy compared to the reverse KL (Table 7), with an average improvement of 0.28 nats. This suggests that explicitly regularizing with respect to the student's entropy during training may not provide additional benefits for distillation quality. Given both the improved performance and reduced computational overhead of forward KL (which avoids computing student entropy), we recommend using standard forward KL for distillation.

*Table 7.* Forward vs Reverse KL Divergence for $N_T = 1.82$B to $N_S = 546$M distillation. Reverse KL is slightly more expensive with respect to vocabulary size $V$ due to the entropy calculation.

| Method | Cross-Entropy | Computational Cost |
|---|---|---|
| Forward KL | 2.42 | $\mathcal{O}(V)$ |
| Reverse KL | 2.70 | $\mathcal{O}(2V)$ |

# H. Parameters and Floating Operation Estimation

Here we outline the number of parameters (Appendix H.2) and the number of FLOPs per token (Appendix H.3) for our experimental settings. The symbol notation is provided in Table 8. For our scaling laws, we find, as in Kaplan et al. (2020) using that the number of *non-embedding-parameters* provides the cleanest fit and extrapolation behavior.

Our expressions for approximate compute (FLOPs per token) differ from prior work in that we are interested in *small models that are capable*. This means we are unable to ignore the context-dependent term that arises from the quadratic computational complexity of the attention mechanism. As our architectures are *fixed aspect ratio*, there is a modified approximation we can use. This expression is discussed in Appendix H.1

For ease of reference, we provide a comparison of the expressions we use to commonly used existing expressions (Kaplan et al., 2020; Hoffmann et al., 2022; Narayanan et al., 2021), and provide comments for significant differences.

*Table 8.* The notation we use for parameter and FLOPs estimation.

| Component | Notation |
|---|---|
| Sequence length/context size | $n_{\text{ctx}}$ |
| Vocabulary size | $n_{\text{vocab}}$ |
| Number of blocks/layers | $n_{\text{layers}}$ |
| Number of query heads | $n_{\text{heads}}$ |
| Number of key/value heads | $n_{\text{kv-heads}}$ |
| Model/embedding dimension | $d_{\text{model}}$ |
| Head dimension | $d_{\text{head}}$ |
| Feed-forward dimension | $d_{\text{ffn}}$ |
| Number of feed-forward linears | $n_{\text{ffn}}$ |
| Group size in Group Query Attention (GQA) $n_{\text{heads}}/n_{\text{kv-heads}}$ | $g_{\text{size}}$ |
| Model aspect ratio $d_{\text{model}}/n_{\text{layers}}$ | $\rho_{\text{model}}$ |
| Feed-forward ratio $d_{\text{ffn}}/d_{\text{model}}$ | $\rho_{\text{ffn}}$ |

### H.1. Alternative approximation for FLOPs per token as a function of $N$

From Table 10 and Equation 71 and Table 12 we can read our approximate values for non-embedding parameters and total compute (dropping contributions from normalization layers) as[8]

$$N = n_{\text{layers}}d_{\text{model}}^2 \left( 2 + \frac{2}{g_{\text{size}}} + n_{\text{ffn}}\rho_{\text{ffn}} \right) \tag{57}$$

$$C_{\text{Forward}} = 2n_{\text{layers}}d_{\text{model}}^2 \left( 2 + \frac{2}{g_{\text{size}}} + n_{\text{ffn}}\rho_{\text{ffn}} \right) + 2n_{\text{layers}}n_{\text{ctx}}d_{\text{model}} \tag{58}$$

$$= 2N + 2n_{\text{layers}}n_{\text{ctx}}d_{\text{model}} + 2n_{\text{vocab}}d_{\text{model}}. \tag{59}$$

---

[8]It was shown in Porian et al. (2024) that ignoring the embedding parameters and FLOPs can lead to systematic estimation bias for small models, and is one of the primary drivers between different exponents reported in Kaplan et al. (2020) and Hoffmann et al. (2022). We find that the the *non-embedding parameters* gives a tighter scaling behavior. However, in the *fixed-aspect-ratio* setting, we are able to use both the *non-embedding parameters* in the scaling law and the *approximate* total compute simultaneously, removing estimation bias. Indeed, in the supervised setting, our coefficients $a$ and $b$ are consistent with those from Hoffmann et al. (2022) (see Table 6).

Typically the term $2n_{\text{layers}}n_{\text{ctx}}d_{\text{model}}$ would be dropped, and the embedding parameters included into the total parameters (Hoffmann et al., 2022) or discarded (Kaplan et al., 2020) yielding the expression $C_{\text{Forward}}$ and the familiar expression $C = 6ND$ (Kaplan et al., 2020; Hoffmann et al., 2022). For our investigation we are interested in small, capable models, which may have a large context, and so both of these terms cannot be ignored in general at the peril of making a systematic error in the region of configuration space we are most interested in. Fortunately, we will see that our choice of *fixed aspect ratio* $\rho_{\text{model}} = d_{\text{model}}/n_{\text{layers}}$ architectures allows us a simple to use, more precise estimate. The trick will be to use this fixed aspect ratio to come up with an approximation for $n_{\text{layers}}$ and $d_{\text{model}}$ as a function of $N$ and $\rho_{\text{model}}$. With these approximated, the term $2n_{\text{layers}}n_{\text{ctx}}d_{\text{model}}$ can be represented as a function of $N$. First define[9]

$$\omega \equiv 2 + \frac{2}{g_{\text{size}}} + n_{\text{ffn}}\rho_{\text{ffn}} \tag{61}$$

so that

$$N = n_{\text{layers}}d_{\text{model}}^2\omega. \tag{62}$$

Then we can substitute in $\rho_{\text{model}} \equiv d_{\text{model}}/n_{\text{layers}}$ so that

$$N = n_{\text{layers}}d_{\text{model}}^2\omega = n_{\text{layers}}^3\rho_{\text{model}}^2\omega, \tag{63}$$

and solve for $n_{\text{layers}}$ and $d_{\text{model}}$

$$n_{\text{layers}} = \left(\frac{N}{\rho_{\text{model}}^2\omega}\right)^{1/3}, \qquad\qquad d_{\text{model}} = \left(\frac{N\rho_{\text{model}}}{\omega}\right)^{1/3}, \tag{64}$$

The $C_{\text{Forward}}$ term can then be represented as a function of $N$. The context-dependent term becomes

$$2n_{\text{ctx}}n_{\text{layers}}d_{\text{model}} = 2n_{\text{ctx}}n_{\text{layers}}^2\rho_{\text{model}} = 2\left(\frac{N}{\rho_{\text{model}}^2\omega}\right)^{2/3}\rho_{\text{model}}n_{\text{ctx}} \equiv 2n_{\text{ctx}}\sigma_1 N^{2/3} \tag{65}$$

where

$$\sigma_1 = \left(\frac{1}{\rho_{\text{model}}^2\omega}\right)^{2/3}\rho_{\text{model}} = \left(\frac{1}{\rho_{\text{model}}\omega^2}\right)^{1/3}. \tag{66}$$

The vocabulary projection term becomes

$$2n_{\text{vocab}}d_{\text{model}} = 2n_{\text{vocab}}\left(\frac{N\rho_{\text{model}}}{\omega}\right)^{1/3} = 2n_{\text{vocab}}\left(\frac{\rho_{\text{model}}}{\omega}\right)^{1/3}N^{1/3} \equiv 2n_{\text{vocab}}\sigma_2 N^{1/3}, \tag{67}$$

where

$$\sigma_2 = \left(\frac{\rho_{\text{model}}}{\omega}\right)^{1/3}. \tag{68}$$

In total

$$C_{\text{Forward}} = 2N + 2n_{\text{ctx}}\sigma_1 N^{2/3} + 2n_{\text{vocab}}\sigma_2 N^{1/3} = 2N\left(1 + \sigma_1\frac{n_{\text{ctx}}}{N^{1/3}} + \sigma_2\frac{n_{\text{vocab}}}{N^{2/3}}\right), \tag{69}$$

where $\sigma_1$ and $\sigma_2$ are independent of model and context size. In the large $N$ limit, or the small $n_{\text{ctx}}$ small $n_{\text{vocab}}$ limit this becomes the familiar $C_{\text{Forward}} = 2N$. The backward FLOPS per token is taken as twice the forward FLOPs (Blondel & Roulet, 2024)

$$C_{\text{Backward}} = 2\,C_{\text{Forward}}. \tag{70}$$

Given the simplicity of the compute expression as a function of $N$, the better tightness of fit in the scaling law, the improved intuition that the model size more directly corresponds to *work being done by the model*, and the predictability of hyperparameters at larger scales, we recommend the scaling law community consider adopting fixed aspect ratio models.

---

[9]In our setting (Appendix I) $\omega$ takes values

$$\omega = 2 + \frac{2}{g_{\text{size}}} + n_{\text{ffn}}\rho_{\text{ffn}} = 2 + \frac{2}{1} + 3 \times \frac{8}{3} = 12. \tag{60}$$

## H.2. Model parameters

In Table 9 we present our parameter counting compared to commonly used existing expressions (Kaplan et al., 2020; Hoffmann et al., 2022; Narayanan et al., 2021). We present a convenient substitution in Table 10 which can be easier to work with analytically. Our total expressions match the architecture we are using, which includes only gains for the normalization layers, whereas while (Narayanan et al., 2021) has both weights and biases. We account for potential use of (Ainslie et al., 2023) as well as use of gated linear attention mechanisms which are becoming prevalent in modern architectures (Shazeer, 2020) including the one used in this work (Appendix I).

*Table 9.* Parameter counts for embedding projector, a single transformer layer, final normalization and output layer. *Ours* indicates the expressions we use in the paper for the total number of parameters (note that the quantity $N$ that appears in our scaling laws is the number of *non-embedding parameters*, but still includes parameters associated with normalization layers). *Approx.* indicates taking the within-section total and dropping all terms that are not at least quadratic in one of $d_{\text{model}}, n_{\text{vocab}}$, and will be used for estimating the FLOPs per token from a given model size (Appendix H.1), and does not differ significantly from the number of non-embedding parameters.

| Parameters | (Kaplan et al., 2020) | (Hoffmann et al., 2022) | (Narayanan et al., 2021) | Ours (Total) |
|---|---|---|---|---|
| Embedding | $(n_{\text{vocab}} + n_{\text{ctx}})d_{\text{model}}$ | $(n_{\text{vocab}} + n_{\text{ctx}})d_{\text{model}}$ | $(n_{\text{vocab}} + n_{\text{ctx}})d_{\text{model}}$ | $n_{\text{vocab}}d_{\text{model}}$ |
| *Attention (one transformer layer)* | | | | |
| PreNorm | — | — | $2d_{\text{model}}$ | $d_{\text{model}}$ |
| QKNorm | — | — | — | $2d_{\text{head}}$ |
| QKV | $3n_{\text{heads}}d_{\text{model}}d_{\text{head}}$ | $3n_{\text{heads}}d_{\text{model}}d_{\text{head}}$ | $3n_{\text{heads}}(d_{\text{model}}+1)d_{\text{head}}$ | $(n_{\text{heads}} + 2n_{\text{kv-heads}})d_{\text{model}}d_{\text{head}}$ |
| Project | $n_{\text{heads}}d_{\text{head}}d_{\text{model}}$ | $n_{\text{heads}}d_{\text{head}}d_{\text{model}}$ | $(n_{\text{heads}}d_{\text{head}}+1)d_{\text{model}}$ | $n_{\text{heads}}d_{\text{head}}d_{\text{model}}$ |
| Total | $4n_{\text{heads}}d_{\text{head}}d_{\text{model}}$ | $4n_{\text{heads}}d_{\text{head}}d_{\text{model}}$ | $4n_{\text{heads}}d_{\text{head}}d_{\text{model}} + 3(n_{\text{heads}}d_{\text{head}} + d_{\text{model}})$ | $2(n_{\text{heads}} + n_{\text{kv-heads}})d_{\text{head}}d_{\text{model}} + 2d_{\text{head}} + d_{\text{model}}$ |
| Approx. | $4n_{\text{heads}}d_{\text{head}}d_{\text{model}}$ | $4n_{\text{heads}}d_{\text{head}}d_{\text{model}}$ | $4n_{\text{heads}}d_{\text{head}}d_{\text{model}} + 3(n_{\text{heads}}d_{\text{head}} + d_{\text{model}})$ | $2(n_{\text{heads}} + n_{\text{kv-heads}})d_{\text{head}}d_{\text{model}}$ |
| *Feed-forward (one transformer layer)* | | | | |
| PreNorm | — | — | $2d_{\text{model}}$ | $d_{\text{model}}$ |
| MLP | $2d_{\text{model}}d_{\text{ffn}}$ | $2d_{\text{model}}d_{\text{ffn}}$ | $2d_{\text{model}}d_{\text{ffn}} + d_{\text{ffn}} + d_{\text{model}}$ | $n_{\text{ffn}}d_{\text{model}}d_{\text{ffn}}$ |
| Total | $2d_{\text{model}}d_{\text{ffn}}$ | $2d_{\text{model}}d_{\text{ffn}}$ | $2d_{\text{model}}d_{\text{ffn}} + d_{\text{ffn}} + 3d_{\text{model}}$ | $n_{\text{ffn}}d_{\text{model}}d_{\text{ffn}} + d_{\text{model}}$ |
| Approx. | $2d_{\text{model}}d_{\text{ffn}}$ | $2d_{\text{model}}d_{\text{ffn}}$ | $2d_{\text{model}}d_{\text{ffn}} + d_{\text{ffn}} + 3d_{\text{model}}$ | $n_{\text{ffn}}d_{\text{model}}d_{\text{ffn}}$ |
| OutputNorm | — | — | — | $d_{\text{model}}$ |
| Final logits | — | — | — | — |

*Table 10.* Parameter counts displayed in Table 9 using simplified notation $n_{\text{heads}}d_{\text{head}} = d_{\text{model}}, d_{\text{ffn}} = \rho_{\text{ffn}}d_{\text{model}}$, and $n_{\text{heads}} = g_{\text{size}}n_{\text{kv-heads}}$.

| Parameters | (Kaplan et al., 2020) | (Hoffmann et al., 2022) | (Narayanan et al., 2021) | Ours (Total) |
|---|---|---|---|---|
| Embedding | $(n_{\text{vocab}} + n_{\text{ctx}})d_{\text{model}}$ | $(n_{\text{vocab}} + n_{\text{ctx}})d_{\text{model}}$ | $(n_{\text{vocab}} + n_{\text{ctx}})d_{\text{model}}$ | $n_{\text{vocab}}d_{\text{model}}$ |
| *Attention (one transformer layer)* | | | | |
| PreNorm | — | — | $2d_{\text{model}}$ | $d_{\text{model}}$ |
| QKNorm | — | — | — | $2d_{\text{head}}$ |
| QKV | $3d_{\text{model}}^2$ | $3d_{\text{model}}^2$ | $3(d_{\text{model}}^2 + d_{\text{model}})$ | $(1 + 2/g_{\text{size}})d_{\text{model}}^2$ |
| Project | $d_{\text{model}}^2$ | $d_{\text{model}}^2$ | $d_{\text{model}}^2 + d_{\text{model}}$ | $d_{\text{model}}^2$ |
| Total | $4d_{\text{model}}^2$ | $4d_{\text{model}}^2$ | $4d_{\text{model}}^2 + 6d_{\text{model}}$ | $2(1 + 1/g_{\text{size}})d_{\text{model}}^2 + 2d_{\text{head}} + d_{\text{model}}$ |
| Approx. | $4d_{\text{model}}^2$ | $4d_{\text{model}}^2$ | $4d_{\text{model}}^2 + 6d_{\text{model}}$ | $2(1 + 1/g_{\text{size}})d_{\text{model}}^2$ |
| *Feed-forward (one transformer layer)* | | | | |
| PreNorm | — | — | $2d_{\text{model}}$ | $d_{\text{model}}$ |
| MLP | $2\rho_{\text{ffn}}d_{\text{model}}^2$ | $2\rho_{\text{ffn}}d_{\text{model}}^2$ | $2\rho_{\text{ffn}}d_{\text{model}}^2 + (1 + \rho_{\text{ffn}})d_{\text{model}}$ | $n_{\text{ffn}}\rho_{\text{ffn}}d_{\text{model}}^2$ |
| Total | $2\rho_{\text{ffn}}d_{\text{model}}^2$ | $2\rho_{\text{ffn}}d_{\text{model}}^2$ | $2\rho_{\text{ffn}}d_{\text{model}}^2 + (3 + \rho_{\text{ffn}})d_{\text{model}}$ | $n_{\text{ffn}}\rho_{\text{ffn}}d_{\text{model}}^2 + d_{\text{model}}$ |
| Approx. | $2\rho_{\text{ffn}}d_{\text{model}}^2$ | $2\rho_{\text{ffn}}d_{\text{model}}^2$ | $2\rho_{\text{ffn}}d_{\text{model}}^2 + (3 + \rho_{\text{ffn}})d_{\text{model}}$ | $n_{\text{ffn}}\rho_{\text{ffn}}d_{\text{model}}^2$ |
| OutputNorm | — | — | — | $d_{\text{model}}$ |
| Final logits | — | — | — | — |

This results in an approximation for the number of non-embedding parameters, dropping subleading terms

$$N \approx n_{\text{layers}}d_{\text{model}}^2 \left(2 + \frac{2}{g_{\text{size}}} + n_{\text{ffn}}\rho_{\text{ffn}}\right) \tag{71}$$

which can be used to estimate forward FLOPs per token from the model size (Appendix H.1).

## H.3. FLOPs per token

In Table 11 we present our counting of the total number of FLOPs per token performed per token during a forward pass compared to commonly used existing expressions (Kaplan et al., 2020; Hoffmann et al., 2022; Narayanan et al., 2021). We present a convenient substitution in Table 12 which can be easier to work with analytically.

Beyond the potential accounting for gated linear layers and grouped query attention, the most important discrepancy across methods is how the attention mechanism is handled. As was also noted in Porian et al. (2024), the expression used in Kaplan et al. (2020) is consistent with efficiently computing a *causal* attention mechanism (Dao et al., 2022; Dao, 2024) whereas Hoffmann et al. (2022); Narayanan et al. (2021) are consistent with counting attention FLOPs for a bidirectional (non-causal) attention mechanism, where the masked component of the attention matrix (zero by construction) is still being computed. We adopt the efficient expression of assuming a causal computation as this more closely reflects best practice.

*Table 11.* Forward FLOPs per for token for embedding projector, a single transformer layer, final normalization and output layer. *Ours* indicates the expressions we use in the paper for the total (note that the quantity $C_{\text{Forward}}$ that appears in compute constraints is the number of *non-embedding floating operations*. *Approx.* indicates taking the within-section total and dropping all terms that are not at least quadratic in one of $d_{\text{model}}$, $n_{\text{vocab}}$, and will be used for estimating the FLOPs per token from a given model size (Appendix H.1).

| FLOPs | (Kaplan et al., 2020) | (Hoffmann et al., 2022) | (Narayanan et al., 2021) | Ours (Total) |
|---|---|---|---|---|
| Embedding | $4d_{\text{model}}$ | $2n_{\text{vocab}}d_{\text{model}}$ | — | $2d_{\text{model}}$ |
| *Attention (one transformer layer)* | | | | |
| PreNorm | — | — | — | — |
| QKNorm | — | — | — | — |
| QKV | $3n_{\text{heads}}2d_{\text{model}}d_{\text{head}}$ | $3n_{\text{heads}}2d_{\text{model}}d_{\text{head}}$ | $3n_{\text{heads}}2d_{\text{model}}d_{\text{head}}$ | $(n_{\text{heads}}+2n_{\text{kv-heads}})2d_{\text{model}}d_{\text{head}}$ |
| Logits | $2n_{\text{heads}}n_{\text{ctx}}d_{\text{head}}$ | $2n_{\text{heads}}n_{\text{ctx}}d_{\text{head}}$ | $2n_{\text{heads}}n_{\text{ctx}}d_{\text{head}}$ | $n_{\text{heads}}n_{\text{ctx}}d_{\text{head}}$ |
| Softmax | — | $3n_{\text{heads}}n_{\text{ctx}}$ | — | $2.5n_{\text{heads}}n_{\text{ctx}}$ |
| Values | — | $2n_{\text{heads}}n_{\text{ctx}}d_{\text{head}}$ | $2n_{\text{heads}}n_{\text{ctx}}d_{\text{head}}$ | $n_{\text{heads}}n_{\text{ctx}}d_{\text{head}}$ |
| Project | $n_{\text{heads}}2d_{\text{head}}d_{\text{model}}$ | $n_{\text{heads}}2d_{\text{head}}d_{\text{model}}$ | $n_{\text{heads}}2d_{\text{head}}d_{\text{model}}$ | $n_{\text{heads}}2d_{\text{head}}d_{\text{model}}$ |
| Total | $2n_{\text{heads}}d_{\text{head}}(4d_{\text{model}}+n_{\text{ctx}})$ | $4n_{\text{heads}}d_{\text{head}}(2d_{\text{model}}+n_{\text{ctx}})+3n_{\text{heads}}n_{\text{ctx}}$ | $4n_{\text{heads}}d_{\text{head}}(2d_{\text{model}}+n_{\text{ctx}})$ | $4n_{\text{heads}}d_{\text{head}}(d_{\text{model}}+n_{\text{ctx}}/2)+4n_{\text{kv-heads}}d_{\text{model}}d_{\text{head}}+2.5n_{\text{heads}}n_{\text{ctx}}$ |
| Approx. | $2n_{\text{heads}}d_{\text{head}}(4d_{\text{model}}+n_{\text{ctx}})$ | $4n_{\text{heads}}d_{\text{head}}(2d_{\text{model}}+n_{\text{ctx}})+3n_{\text{heads}}n_{\text{ctx}}$ | $4n_{\text{heads}}d_{\text{head}}(2d_{\text{model}}+n_{\text{ctx}})$ | $4n_{\text{heads}}d_{\text{head}}(d_{\text{model}}+n_{\text{ctx}}/2)+4n_{\text{kv-heads}}d_{\text{model}}d_{\text{head}}$ |
| *Feed-forward (one transformer layer)* | | | | |
| PreNorm | — | — | — | — |
| MLP | $4d_{\text{model}}d_{\text{ffn}}$ | $4d_{\text{model}}d_{\text{ffn}}$ | $4d_{\text{model}}d_{\text{ffn}}$ | $2n_{\text{ffn}}d_{\text{model}}d_{\text{ffn}}$ |
| OutputNorm | — | — | — | — |
| Final logits | $2n_{\text{vocab}}d_{\text{model}}$ | $2n_{\text{vocab}}d_{\text{model}}$ | $2n_{\text{vocab}}d_{\text{model}}$ | $2n_{\text{vocab}}d_{\text{model}}$ |

*Table 12.* Forward FLOPs counts per token from Table 11 simplified using $n_{\text{heads}}d_{\text{head}}=d_{\text{model}}$, $d_{\text{ffn}}=\rho d_{\text{model}}$, and $n_{\text{heads}}=g_{\text{size}}n_{\text{kv-heads}}$.

| FLOPs | (Kaplan et al., 2020) | (Hoffmann et al., 2022) | (Narayanan et al., 2021) | Ours (Total) |
|---|---|---|---|---|
| Embedding | $4d_{\text{model}}$ | $2n_{\text{vocab}}d_{\text{model}}$ | — | $2d_{\text{model}}$ |
| *Attention (one transformer layer)* | | | | |
| PreNorm | — | — | — | — |
| QKNorm | — | — | — | — |
| QKV | $6d_{\text{model}}^2$ | $6d_{\text{model}}^2$ | $6d_{\text{model}}^2$ | $2(1+2/g_{\text{size}})d_{\text{model}}^2$ |
| Logits | $2d_{\text{model}}n_{\text{ctx}}$ | $2d_{\text{model}}n_{\text{ctx}}$ | $2d_{\text{model}}n_{\text{ctx}}$ | $d_{\text{model}}n_{\text{ctx}}$ |
| Softmax | — | $3n_{\text{heads}}n_{\text{ctx}}$ | — | $2.5n_{\text{heads}}n_{\text{ctx}}$ |
| Values | — | $2d_{\text{model}}n_{\text{ctx}}$ | $2d_{\text{model}}n_{\text{ctx}}$ | $d_{\text{model}}n_{\text{ctx}}$ |
| Project | $2d_{\text{model}}^2$ | $2d_{\text{model}}^2$ | $2d_{\text{model}}^2$ | $2d_{\text{model}}^2$ |
| Total | $8d_{\text{model}}^2+2n_{\text{ctx}}d_{\text{model}}$ | $8d_{\text{model}}^2+4n_{\text{ctx}}d_{\text{model}}+3n_{\text{heads}}n_{\text{ctx}}$ | $8d_{\text{model}}^2+4n_{\text{ctx}}d_{\text{model}}$ | $(4+4/g_{\text{size}})d_{\text{model}}^2+2n_{\text{ctx}}d_{\text{model}}+2.5n_{\text{heads}}n_{\text{ctx}}$ |
| Approx. | $8d_{\text{model}}^2+2n_{\text{ctx}}d_{\text{model}}$ | $8d_{\text{model}}^2+4n_{\text{ctx}}d_{\text{model}}+3n_{\text{heads}}n_{\text{ctx}}$ | $8d_{\text{model}}^2+4n_{\text{ctx}}d_{\text{model}}$ | $(4+4/g_{\text{size}})d_{\text{model}}^2+2n_{\text{ctx}}d_{\text{model}}$ |
| *Feed-forward (one transformer layer)* | | | | |
| PreNorm | — | — | — | — |
| MLP | $4\rho_{\text{ffn}}d_{\text{model}}^2$ | $4\rho_{\text{ffn}}d_{\text{model}}^2$ | $4\rho_{\text{ffn}}d_{\text{model}}^2$ | $2n_{\text{ffn}}\rho_{\text{ffn}}d_{\text{model}}^2$ |
| OutputNorm | — | — | — | — |
| Final logits | $2n_{\text{vocab}}d_{\text{model}}$ | $2n_{\text{vocab}}d_{\text{model}}$ | $2n_{\text{vocab}}d_{\text{model}}$ | $2n_{\text{vocab}}d_{\text{model}}$ |

This results in an approximation for the number of non-embedding floating operations per token, dropping subleading terms

$$C_{\text{Forward}} \approx 2n_{\text{layers}}d_{\text{model}}^2\left(2+\frac{2}{g_{\text{size}}}+n_{\text{ffn}}\rho_{\text{ffn}}\right)+2n_{\text{layers}}n_{\text{ctx}}d_{\text{model}}+2n_{\text{vocab}}d_{\text{model}} \tag{72}$$

which can be used to estimate forward FLOPs per token from the model size (Appendix H.1).

# I. Model architecture

All models are based on Gunter et al. (2024) and are trained using AXLearn (Apple, 2023). All models use decoupled weight decay Loshchilov & Hutter (2019) of $10^{-4}$ for regularization, as well as a simplified version of $\mu$P (Yang & Hu, 2021; Yang & Littwin, 2023; Yang et al., 2022; Wortsman et al., 2023; Yang et al., 2023), following what is described as $\mu$P (simple) in (Wortsman et al., 2024). Because of $\mu$P (simple), we fix the learning rate to $1e-2$ across all model sizes. Multi-headed attention (MHA) is used ($g_{size} = 1$), with Pre-Normalization (Nguyen & Salazar, 2019) using RMSNorm (Zhang & Sennrich, 2019). We train all models with a sequence length of $n_{ctx} = 4096$, with RoPE (Su et al., 2024) positional embeddings (base frequency set to 500k). All model architectures in this work are presented in Table 13, have a *fixed aspect ratio* $d_{model} = 128$ and a *fixed ffn ratio* $\rho_{ffn} = 8/3$ coupled with gated linear activation ($n_{ffn} = 3$).

*Table 13.* The models used in this work. The different parameter values and FLOPs per token are shown in billions. $N$ is the number of *non-embedding parameters* and isthe value we use in our scaling laws. $N_{total}$ counts all parameters in the model.$C_{fwd}$ is the total number of forward FLOPs per token given by the fulltotal in Tables 11 and 12.$C_{fwd-approx(2N)}$ is the estimated value of forward FLOPs per tokenbased on the $2N$ approximation, and is accompanied by its relative error.$C_{fwd-approx(2N+\sigma)}$ is the estimated value of forward FLOPs per tokenbased on the approximation given in Equation 69, and is accompanied by its relative error.The $C_{fwd-approx(2N+\sigma)}$ is the one we use in this work.

| Name | $N$ (B) | $N_{total}$ (B) | $n_{layers}$ | $d_{model}$ | $d_{ff}$ | $C_{fwd}$ (B) | $C_{fwd-approx(2N)}$ (B) | $C_{fwd-approx(2N+\sigma)}$ (B) |
|---|---|---|---|---|---|---|---|---|
| 103M | 0.1028 | 0.1363 | 8 | 1024 | 2816 | 0.3411 | 0.2056 (-39.74%) | 0.3398 (-0.39%) |
| 143M | 0.1434 | 0.1811 | 9 | 1152 | 3072 | 0.4487 | 0.2867 (-36.10%) | 0.4471 (-0.34%) |
| 198M | 0.1983 | 0.2402 | 10 | 1280 | 3456 | 0.587 | 0.3965 (-32.44%) | 0.5853 (-0.29%) |
| 266M | 0.2657 | 0.3118 | 11 | 1408 | 3840 | 0.7524 | 0.5314 (-29.38%) | 0.7505 (-0.25%) |
| 340M | 0.3398 | 0.3901 | 12 | 1536 | 4096 | 0.9333 | 0.6796 (-27.19%) | 0.9312 (-0.22%) |
| 435M | 0.4348 | 0.4893 | 13 | 1664 | 4480 | 1.158 | 0.8695 (-24.91%) | 1.156 (-0.19%) |
| 546M | 0.546 | 0.6047 | 14 | 1792 | 4864 | 1.417 | 1.092 (-22.96%) | 1.415 (-0.17%) |
| 664M | 0.6636 | 0.7265 | 15 | 1920 | 5120 | 1.692 | 1.327 (-21.54%) | 1.689 (-0.15%) |
| 810M | 0.8096 | 0.8767 | 16 | 2048 | 5504 | 2.025 | 1.619 (-20.03%) | 2.022 (-0.14%) |
| 975M | 0.9755 | 1.047 | 17 | 2176 | 5888 | 2.4 | 1.951 (-18.69%) | 2.397 (-0.12%) |
| 1.15B | 1.147 | 1.222 | 18 | 2304 | 6144 | 2.787 | 2.293 (-17.72%) | 2.784 (-0.11%) |
| 1.35B | 1.355 | 1.434 | 19 | 2432 | 6528 | 3.25 | 2.709 (-16.65%) | 3.247 (-0.10%) |
| 1.59B | 1.586 | 1.67 | 20 | 2560 | 6912 | 3.763 | 3.172 (-15.70%) | 3.759 (-0.09%) |
| 1.82B | 1.821 | 1.909 | 21 | 2688 | 7168 | 4.284 | 3.642 (-14.99%) | 4.28 (-0.09%) |
| 2.1B | 2.102 | 2.194 | 22 | 2816 | 7552 | 4.899 | 4.203 (-14.21%) | 4.895 (-0.08%) |
| 2.41B | 2.41 | 2.506 | 23 | 2944 | 7936 | 5.571 | 4.819 (-13.49%) | 5.567 (-0.07%) |
| 2.72B | 2.718 | 2.819 | 24 | 3072 | 8192 | 6.246 | 5.436 (-12.96%) | 6.241 (-0.07%) |
| 3.08B | 3.082 | 3.187 | 25 | 3200 | 8576 | 7.034 | 6.165 (-12.36%) | 7.03 (-0.06%) |
| 3.48B | 3.478 | 3.587 | 26 | 3328 | 8960 | 7.887 | 6.956 (-11.81%) | 7.883 (-0.06%) |
| 3.87B | 3.87 | 3.983 | 27 | 3456 | 9216 | 8.736 | 7.74 (-11.40%) | 8.731 (-0.05%) |
| 4.33B | 4.329 | 4.446 | 28 | 3584 | 9600 | 9.72 | 8.658 (-10.93%) | 9.715 (-0.05%) |
| 4.82B | 4.823 | 4.944 | 29 | 3712 | 9984 | 10.78 | 9.646 (-10.49%) | 10.77 (-0.05%) |
| 5.31B | 5.309 | 5.434 | 30 | 3840 | 10240 | 11.82 | 10.62 (-10.16%) | 11.81 (-0.05%) |
| 5.87B | 5.873 | 6.003 | 31 | 3968 | 10624 | 13.02 | 11.75 (-9.78%) | 13.01 (-0.04%) |
| 6.48B | 6.476 | 6.611 | 32 | 4096 | 11008 | 14.3 | 12.95 (-9.43%) | 14.29 (-0.04%) |
| 7.07B | 7.066 | 7.204 | 33 | 4224 | 11264 | 15.56 | 14.13 (-9.16%) | 15.55 (-0.04%) |
| 7.75B | 7.747 | 7.889 | 34 | 4352 | 11648 | 17 | 15.49 (-8.85%) | 16.99 (-0.04%) |
| 8.47B | 8.47 | 8.617 | 35 | 4480 | 12032 | 18.52 | 16.94 (-8.55%) | 18.52 (-0.03%) |
| 9.17B | 9.173 | 9.324 | 36 | 4608 | 12288 | 20.01 | 18.35 (-8.33%) | 20.01 (-0.03%) |
| 10B | 10.05 | 10.2 | 37 | 4736 | 12672 | 21.85 | 20.1 (-8.02%) | 21.84 (-0.03%) |
| 10.8B | 10.84 | 11 | 38 | 4864 | 13056 | 23.51 | 21.67 (-7.83%) | 23.5 (-0.03%) |
| 11.7B | 11.66 | 11.83 | 39 | 4992 | 13312 | 25.26 | 23.33 (-7.64%) | 25.25 (-0.03%) |
| 12.6B | 12.61 | 12.78 | 40 | 5120 | 13696 | 27.24 | 25.22 (-7.42%) | 27.23 (-0.03%) |

We rescale the gradients, such that the maximum of the global norm is 1.0. A cosine learning rate schedule is used with warmup (2000 steps), with a final learning rate of one thousandths of the peak learning rate. A Z-loss (Chowdhery et al.,

2023) of $10^{-4}$ is used for stability, slightly decreasing norm growth at the end of the training.

For all experiments, the English-only subset of the C4 dataset (Raffel et al., 2020) is used. The C4 dataset was chosen because of its wide usage in the research community. While C4 is big enough for larger-scale experiments, it is small enough to allow for reproduction of experiments. For all distillation trainings, the teacher is trained on a different split as the student. The C4 dataset has roughly 180B tokens in total, which results in 90B unique tokens for the teacher training and 90B unique tokens for the student training. Except for the largest models, all Chinchilla-optimal models do not repeat data. Models that overtrain on more than 90B tokens will have data repetition too. Muennighoff et al. (2023b) has shown (on the C4 dataset) that repeating data up to 4 times has negligible impact to loss compared to having unique data.

## J. Contributions

All authors contributed to writing this paper, designing the experiments, discussing results at each stage of the project.

**Writing and framing**  Majority of writing done by Dan Busbridge, Jason Ramapruam, and Amitis Shidani. Research direction led by Dan Busbridge, with research framing, question identification, and prioritization done by all authors.

**Scaling law experiments**  Fixed aspect ratio models (Appendix I) FLOP counting methods (Appendix H.1), and model implementation done by Dan Busbridge, Amitis Shidani, and Floris Weers. Dataset preparation done by Floris Weers. IsoFLOP experimental design (Section 4.1) done by Dan Busbridge. Teacher training and distillations done by Dan Busbridge, Amitis Shidani, and Floris Weers. Longer training duration (512B token) teachers and students trained by Floris Weers.

**Scaling law analysis**  Original scaling law fitting code based on Besiroglu et al. (2024) developed by Amitis Shidani. Generalized, JAX Just In Time (JIT) compilation compatible scaling law fitting code, and numerical minimization approaches for compute optimal analysis (Section 5 and Appendix D) done by Dan Busbridge. Functional form (Equation 8) developed by Dan Busbridge, in collaboration with Jason Ramapuram, Amitis Shidani, Russ Webb, and Floris Weers.

**Scaling law downstream metrics**  Implementations of calibration Appendix E.8, Cumulative Distribution Function (CDF) and top-$k$ metrics done by Amitis Shidani. Downstream model evaluations (Appendix E.1) done by Floris Weers.

**Teacher student capacity gaps**  Kernel regression demonstration of the capacity gap phenomenon (Appendix C.1) done by Etai Littwin. MLP synthetic demonstration of the capacity gap phenomenon (Appendix C.2) done by Russ Webb.

**Distilling language models in practice**  Mixing coefficient sensitivity analysis (Appendix G.1) done by Dan Busbridge and Jason Ramapuram. Temperature (Appendix G.2) and learning rate (Figure 53) sensitivity analyses done by Dan Busbridge. Top-$k$ and top-$p$ distribution truncation (Appendix G.4) implementation and analyses done by Jason Ramapuram. Mixing coefficient combined with truncation analysis (Appendix G.4) done by Jason Ramapuram. Reverse KL divergence Appendix G.5 implementation and analysis done by Jason Ramapuram.

