# OpenReview forum: "Distillation Scaling Laws"
_ICML.cc/2025/Conference — ICML 2025 poster_

### Official Review · Reviewer_wCXj · 2025-03-02

**Overall Recommendation:** 3

**Summary:**

The authors propose a distillation scaling law that estimates the performance of distilled models based on a computed budget and its allocation between the student and teacher models. The study provides insights into when distillation outperforms supervised pretraining and offers compute-optimal distillation recipes. The authors conduct extensive experiments with models ranging from 143M to 12.6B parameters, trained on data ranging from a few billion to 512 billion tokens. The findings suggest that distillation is more efficient than supervised learning under specific conditions, such as when a teacher already exists or when the teacher has uses beyond a single distillation.

**Claims And Evidence:**

Yes, the claims are supported by extensive empirical data and theoretical analysis. The authors provide a distillation scaling law (Equation 8) that accurately predicts student performance based on the teacher's cross-entropy, student size, and the amount of distillation data. The experiments demonstrate that the scaling law holds across a wide range of model sizes and compute budgets.

**Essential References Not Discussed:**

The paper could benefit from discussing more recent literature on distillation and scaling laws, particularly from 2022 to 2024.

**Experimental Designs Or Analyses:**

The experimental design is robust, with a large-scale study involving models of varying sizes and compute budgets.

**Methods And Evaluation Criteria:**

The methods are well-designed and solid. The authors use a combination of fixed model size, varied data, and IsoFLOP profiles to fit the scaling law coefficients. They also validate the scaling law through extrapolation and comparison with supervised learning. The evaluation criteria are clear, focusing on cross-entropy loss and downstream task performance.

**Other Comments Or Suggestions:**

The author provides detailed theoretical support, but can this theory be applied in practical scenarios, such as selecting teacher models for the LLaMA model and summarizing the actual effects?

**Other Strengths And Weaknesses:**

**Strengths**
1. The paper provides a comprehensive empirical study of distillation, with systematic ablations and a clear scaling law. Besides, the theoretical analysis and empirical results are well-aligned, providing strong evidence for the proposed scaling law.
2. The distillation scaling law offers practical guidance for producing smaller, more powerful models with lower inference costs.
3. The visualization is good for understanding the distillation law.

 **Weaknesses**
1. Is the expression and coefficients of Formula 8 a conclusion derived only from a few hundred samples? I think it is not a universal conclusion.
2. The paper could benefit from more detailed ablation studies, such as the impact of different distillation temperatures or mixing coefficients.

**Questions For Authors:**

I have doubts about the statement "When teacher training is included in compute, the best student cross-entropy is always higher than in the supervised setting" on line 385. When we need to deploy a pocket-size model, if we want to improve the model's accuracy, we can only achieve this by distilling a larger teacher model through training, rather than relying solely on student cross-entropy.

**Relation To Broader Scientific Literature:**

The paper builds on previous work on scaling laws and knowledge distillation, providing a comprehensive framework for understanding and optimizing distillation in large language models. It addresses the growing concern of inference costs and offers practical guidelines for compute-optimal distillation.

**Theoretical Claims:**

Yes,  I have checked the correctness of proofs for theoretical claims.

---

> ### Author Rebuttal · Authors · 2025-03-31
>
> Thank you wCXj for your thoughtful feedback. We’re encouraged you found our study comprehensive and useful, particularly the practical guidance provided by our distillation scaling law.
>
> ## There are missing distillation and scaling law references from 2022-2024
>
> We extensively cover distillation (App B.1) and scaling laws (App B.2). If specific key papers or topics are missing, please clarify, and we'll be happy to include them.
>
> ## How is  [formula 8] the Distillation Scaling Law derived?
>
> The distillation scaling law was the simplest extrapolating function we found that:
>
> 1. Satisfied 3 properties (L243-250 right)
> 2. Extrapolated to unseen student-teacher combinations (orange scatter in grey region Fig 5).
> 3. Was a power law in student size $N_S$ and tokens $D_S$ (L187-189 right).
>
> ## Is the Distillation Scaling Law universal?
>
> Other expressions could work.
>
> Even standard supervised scaling laws differ (footnote 9, L106-109), as well as which terms belong in the scaling law (Appendix H.1, footnote 7, L2965-2968).
>
> We note this in our limitations, highlighting all scaling laws share this problem unless explicitly derived [1].
>
> Practically what matters is correct limits and reliable extrapolation, both of which our law satisfies.
>
> ## Are the coefficients universal?
>
> The coefficients (Appendix F.3, Table 5, L2606-2619), are not universal, they’re dataset-dependent.
>
> This limitation applies to all scaling law studies; data manifolds of different intrinsic dimensionality produce different scaling coefficients [2].
>
> Broader conclusions from previous studies [3,4] have successfully generalized beyond their datasets. We expect our main conclusions and recommendations (Sec 1) to similarly generalize.
>
> ##  The paper needs more detailed ablation studies (e.g. distillation temperature, mixing coefficients)
>
> We provide detailed studies in the appendix. The two you mention are:
>
> 1. Distillation temperature: App G.3
> 2. Mixing coefficients: App G.4
>
> Additionally, we study learning rates (G.5), divergence measures (G.6), and truncation strategies (G.1-2). We believe these detailed analyses significantly expand the empirical understanding of distillation.
>
> ## Can the findings be used in practice?
>
> Yes. Our scaling law facilitates optimal experimental design (Sec 5, App D.2-4), assuming the scaling law is computed for your scenario (Sec 4). We'll clarify these practical application steps further in our revision.
>
> ## Can the findings be used to select the teacher model for LLaMA?
>
> Yes.
>
> Given a computed scaling law (Sec 4) and candidate LLaMA teachers $\{\text{LLaMA}_i\}_{i=1}^n$:
>
> 1. Compute each teacher’s cross-entropy $L_T^{(i)}$ on target data
> 2. Given target student size ($N_S$) and token or compute budget, select the teacher yielding lowest student cross-entropy ($L_S$).
>
> We'll clarify this selection method in the manuscript.
>
> ## Is the statement “When teacher training is included in compute, the best student cross-entropy is always higher than in the supervised setting" on L385 correct? (if we want to improve the model's accuracy, we can only achieve this by distilling a larger teacher model through training, rather than relying solely on student cross-entropy)
>
> The statement is about cross-entropy, not accuracy, and is correct. See Figure 8. Strategies involving training a teacher (red and green) produce student cross-entropies (y-axis) that are strictly above the cross-entropy of a non-distilled model (black line) for all compute budgets. This is expected, as to find the reverse would imply the existence of an algorithm more efficient than direct maximum likelihood optimization.
>
> Predicting accuracy is an open problem in scaling studies. We observe some correspondence between downstream accuracy and cross-entropy (Appendix E.1), however, our primary claims throughout the work only relate to cross-entropy, as is standard practice in scaling law studies.
>
> We thank you again for your valuable feedback which will help improve our work, and hope our response clarifies any misunderstandings regarding some points of the work that will be clarified in an updated version.
>
> [1] 4+3 Phases of Compute-Optimal Neural Scaling Laws https://arxiv.org/abs/2405.15074
>
> [2] A Neural Scaling Law from the Dimension of the Data Manifold https://arxiv.org/abs/2004.10802
>
> [3] Scaling Laws for Neural Language Models https://arxiv.org/abs/2001.08361
>
> [4] Training Compute-Optimal Large Language Models https://arxiv.org/abs/2203.15556

---

### Official Review · Reviewer_hWGR · 2025-03-13

**Overall Recommendation:** 4

**Summary:**

In this paper, authors establish a distillation scaling law to predict the performance of a student model based on compute budget and its allocation between teacher and student models. Using this law, authors can infer optimal distillation strategies for different scenarios, including cases where a teacher already exists or where both the teacher and student need to be trained.

**Claims And Evidence:**

Claim 1: distillation performance scaling law. The claim is supported by the parametric fits shown in Figure 1.

Claim 2: If a teacher is too strong compared to the student, distillation becomes less effective because of the capacity gap. The claim is supported by Figure 2, 3, and 4 clearly showing that beyond a certain teacher-to-student ratio, further improving the teacher does not improve (or even worsens) student performance.

Claim 3: Distillation outperforms supervised learning only if a teacher already exists or can be reused across multiple distillations. In Figure 8, my understanding is that with a limit FLOPs, there is some advantages only training the student model without first training the teacher then distilling. However, as we increase FLOPS, aiming for lower cross entropy loss, I no longer see a meaningful difference. I do not think the result in Figure 8 supports the claim.

**Essential References Not Discussed:**

N/A

**Experimental Designs Or Analyses:**

The study conducts an extensive controlled distillation study with models ranging from 143M to 12.6B parameters. The experiments are thorough and extensive.

**Methods And Evaluation Criteria:**

To establish the distillation scaling law, authors examine teacher model size, number of training tokens, student model size and number of distillation tokens. All of above are key factors impacting the distillation quality.

**Other Comments Or Suggestions:**

While distillation scaling law is interesting, my understanding is that a more capable model is often used to generate synthetic data, which in turn is used to train the student model. Is there any known work or examples where we know teacher distillation is actually used to pretrain LLMs?

**Other Strengths And Weaknesses:**

Strength:

1. This paper is a large scale empirical study of distillation scaling laws, with extensive amount of experiment examining all relevant axis of distillation quality.

2. The results and analysis provided by the paper leads to actionable guidance for LLM pretraining.

Weakness:

1. This paper assumes a very naive distillation paradigm. However, I could imagine common known distillation tricks such as adaptive teacher training (training on better teachers as the student is learning), distillation temperature tuning (higher temperature might improve the distillation effectiveness). While the goal is not to examine every trick but the most-known best ones might worth considering.

**Questions For Authors:**

See comments.

**Relation To Broader Scientific Literature:**

This paper contributes to the field of scaling laws. As distillation becomes more common with the prevalence of pretrained models, such an analysis is important for the scientific community.

**Theoretical Claims:**

N/A

---

> ### Author Rebuttal · Authors · 2025-03-31
>
> Dear hWGR. Thank you for your taking the time to review our paper and for your detailed feedback. We are encouraged that you found our extensive study useful for showcasing different aspects of distillation, and that you find the overall guidance provided by the Distillation Scaling Law to have utility for practical training scenarios. Finally, we are happy to see your overall enthusiasm for the work, and understanding of its place as a step towards more reliable distillation training, and extending the scaling laws field.
>
> ## Figure 8 doesn't support the claim “Distillation outperforms supervised learning only if a teacher already exists or can be reused across multiple distillations [...] as we increase FLOPS there is no longer a meaningful difference”
>
> Your understanding is correct, and is also the claim we make in the paper:
>
> * (L020-022) If [...] a teacher already exists, distillation outperforms supervised pretraining until a compute level which grows predictably with student size.
> * (L078-080, right) [...] distillation is only more efficient than supervised learning only if both of the following are true: i) the total compute or tokens used for the student is not larger than student size-dependent threshold given by our scaling law, and ii) a teacher already exists [...]
> * (L427, right) distillation is only more efficient than supervised learning if: i) the total compute or tokens used for distillation is not larger than a student size-dependent threshold, and ii) a teacher already exists [...]
>
> i.e. the right hand side of Figure 8 you point out is precisely the second part of our condition “compute or tokens is not too great, when supervised learning and distillation produce the same answer”. We provide further analysis in Appendix E.6 on the vanishing of the difference between methods.
>
> ## The study focuses on logit-level distillation
>
> This is true, we only make statements about logit-level distillation.
>
> This is a popular distillation technique for training language models, as it is used by Gemini and Gemma [1], Apple Foundation Models [2], and Minitron [3].
>
> The other primary distillation technique for language model training is SeqKD [4] employed by e.g. DeepSeek-R1 [5], and would be a good choice for an extension of our work, (discussed on L1117-1126).
>
> While we do agree that other distillation techniques are also important to explore, in order to keep the scope of this paper (and compute) constrained we choose to focus on a commonly used form of distillation. We are happy we were able to make progress on one of the most popular techniques, but agree there is a lot more that can be done to bring remaining techniques to the same level of actionable understanding.
>
> [1] Gemma 2: Improving Open Language Models at a Practical Size https://arxiv.org/abs/2408.00118
>
> [2] Apple Intelligence Foundation Language Models https://arxiv.org/abs/2407.21075
>
> [3] LLM Pruning and Distillation in Practice: The Minitron Approach https://arxiv.org/abs/2408.11796
>
> [4] Sequence-Level Knowledge Distillation https://arxiv.org/abs/1606.07947
>
> [5] DeepSeek-R1: Incentivizing Reasoning Capability in LLMs via Reinforcement Learning https://arxiv.org/abs/2501.12948
>
> ## What happens in the case where a more capable model is used for synthetic data generation (instead of logit distillation)
>
> This is the case of SeqKD (see also answer above on logic-level distillation). In this case, the student would be trained on the modes of the distributions represented by the teacher, which should correspond to the modes of language. There is a qualitative and quantitative difference between SeqKD and token level logit distillation (see [4]). Studying the scaling properties of this would be a great next step.
>
> [4] Sequence-Level Knowledge Distillation https://arxiv.org/abs/1606.07947
>
> ## Are there known examples where teacher distillation is used in pretraining?
>
> Yes (see also answer to logic-level query, as well as references at L073-080).
>
> Gemini and Gemma [1], Apple Foundation Models [2], and Minitron [3] use token-level logit distillation, the subject of our investigation. DeepSeek-R1 [5], and the LLaMA family [6] use language model synthetic generation, i.e. a form of SeqKD [4].
>
> [1] Gemma 2: Improving Open Language Models at a Practical Size https://arxiv.org/abs/2408.00118
>
> [2] Apple Intelligence Foundation Language Models https://arxiv.org/abs/2407.21075
>
> [3] LLM Pruning and Distillation in Practice: The Minitron Approach https://arxiv.org/abs/2408.11796
>
> [4] Sequence-Level Knowledge Distillation https://arxiv.org/abs/1606.07947
>
> [5] DeepSeek-R1: Incentivizing Reasoning Capability in LLMs via Reinforcement Learning https://arxiv.org/abs/2501.12948
>
> [6] The Llama 3 Herd of Models https://arxiv.org/abs/2407.21783
>
> Again, thank you for reading our work and provide valuable feedback which will be incorporated in an improved version.

---

### Official Review · Reviewer_1KoJ · 2025-03-14

**Overall Recommendation:** 4

**Summary:**

The paper discusses a very interesting topic related to scaling laws for distillation. Following the strategies of recent works on scaling laws given a fixed compute budget, this paper provides a framework to estimate the performance of distilled student models based on different cost settings related to the teacher and the student models. The authors investigated the optimum distillation strategies for different teacher scenarios (e.g. teacher model exists already, teacher model needs to be pre-trained first, etc..).
From their findings one main recipe stood out is that distillation is powerful to some extent. Distillation is efficient if the teacher training cost was not taken into account (already exists) but if it was in a reverse scenario, then supervised training would be more beneficial than distillation given the fixed data setting.

**Claims And Evidence:**

The authors effectively supported their claims through extensive empirical analysis. They conducted a comprehensive study on distillation providing various insights about the synergy of distillation between teacher and student models. They thoroughly crafted their experimental design and demonstrated a strong foundation.

However, the study’s generalizability is a potential concern. While the authors explored distillation across various model capacities, the architectural diversity of these models introduces a significant variable. The authors could have shown some distinction between families like LLaMA, Gemini, Mistral, Deepseek, etc; and how the architectural choice also might affect the generalizability of their framework. The current analysis does not fully address how architectural discrepancies impact distillation efficacy. As a user heavily reliant on distillation techniques, I am particularly interested in seeing a more focused investigation into family-based distillation. Specifically, exploring whether established scaling laws hold true when distilling within the same architectural family versus across different architectures would provide invaluable insights. This targeted examination would significantly strengthen the study's applicability and broaden our understanding of distillation's nuanced dynamics.

**Essential References Not Discussed:**

There are some papers in the literature that could strengthen the calibration and capacity problem studies in their related work:
[1] Lee, Dongkyu, et al. "Hard Gate Knowledge Distillation--Leverage Calibration for Robust and Reliable Language Model." arXiv preprint arXiv:2210.12427 (2022).
[2] Amara, Ibtihel, et al. "Bd-kd: balancing the divergences for online knowledge distillation." arXiv preprint arXiv:2212.12965 (2022).
[3] Fan, Wen-Shu, et al. "Revisit the essence of distilling knowledge through calibration." Forty-first International Conference on Machine Learning. 2024.

**Experimental Designs Or Analyses:**

Yes, I have carefully examined the soundness and validity of the experimental designs and analyses provided in the paper. The choice of analytical methods aligns well with their research questions, ensuring a coherent and rigorous evaluation.
However, it would have been beneficial if the authors had included additional analyses addressing the capacity gap problem. This inherent challenge in knowledge distillation can be mitigated through various distillation strategies, a topic that has been extensively explored in prior research. Many studies have focused on improving distillation techniques to better calibrate student models, which highlights the importance of further investigation in this area.
[1] Lee, Dongkyu, et al. "Hard Gate Knowledge Distillation--Leverage Calibration for Robust and Reliable Language Model." arXiv preprint arXiv:2210.12427 (2022).
[2] Amara, Ibtihel, et al. "Bd-kd: balancing the divergences for online knowledge distillation." arXiv preprint arXiv:2212.12965 (2022).

A key limitation of the current work is its exclusive focus on logit based distillation. While this is a fundamental approach, incorporating analyses of alternative distillation techniques, such as feature-based, layer-based, or other advanced methods, could provide deeper insights into both the capacity gap issue and the generalizability of the proposed scaling laws.

**Methods And Evaluation Criteria:**

The paper does not propose any benchmark datasets. They evaluate on the C4 dataset for
their framework. While understandable at a high level, this could present a notable limitation.
The absence of diverse dataset raises concerns about the generalizability of the proposed scaling law framework. As previously mentioned, this singular evaluation environment may not accurately reflect performance in varied research settings. This should be better emphasized in the paper so that researchers would easily grasp this before adapting their findings to their different research settings.

**Other Comments Or Suggestions:**

There are a few typos, and I recommend a thorough proofread. Additionally, some references are missing, including a TODO placeholder in the Supplemental Material (L-1888).

As for other comments, please refer to all the comments stated above.

**Other Strengths And Weaknesses:**

I would like to summarize most of the points I mentioned above:
Strengths:
- The authors provided a comprehensive analysis: They have conducted a large scale study offering a valuable insight into understanding distillation synergy between large teachers and smaller student models.
- The scaling law framework for distillation: The authors provided compute-optimal distillation recipes that are useful as a reference for users aiming to maximize the performance within a particular computational constraints.

Weaknesses:
Generalizability: There are many dimensions I would like to address with respect to this point (explained in the first three bullet points).
- Architecture of the models (as explained above): There should be more analyses and experiments as how the architecture of the model can disrupt the generalizability of the scaling law. For example, distilling LLaMa teacher to a LLaMa student model or distilling a Gemini teacher to a LLaMa student. It would have been nice to see how this could affect the distillation laws.
- Distillation Schemes:  There are many ways to perform distillation. The study only focuses on logit-based distillation, which again might be limiting the generalization aspect of the findings. What if we want to distill layer-wise ? Would the findings still hold?
- Training assumptions: The assumption in this study that both teacher and student models are pretrained using C4 dataset. What if the teacher was pretrained on a different dataset? Would the framework still hold?
- Readability: The Results section was difficult to follow due to frequent references to supplementary materials for key analyses. I suggest adding more details to figure captions and integrating essential explanations directly into the main text. To make space, the authors could streamline the experimental design section and the table of definitions. This would improve clarity and ensure the findings are more accessible without excessive cross-referencing.

**Questions For Authors:**

No other questions. Please refer to the Limitations above.

**Relation To Broader Scientific Literature:**

The study introduces an important research aspect related to distillation. They introduced a novel framework that can estimate the distilled model performance based on compute budget and their allocation between both teacher and student. In addition to that, their findings suggest that distillation surpasses supervised learning under certain conditions. They contributed by giving the optimal recipe given the existing costs tied to the teacher model: If a teacher model already exists distillation is the best route for student model training and if the teacher model has to be pretrained first, then it is safer to take the route of supervised learning.
These contributions build upon prior work by offering a more comprehensive understanding of the choices.

**Theoretical Claims:**

Yes. I checked. The theoretical claims seems correct.

---

> ### Author Rebuttal · Authors · 2025-03-31
>
> Thank you, 1KoJ for your valuable feedback. We’re happy you found our analysis useful, the compute-optimal recipes useful for practitioners, and appreciated our work in the context of the scaling literature.
>
> ## Architectural diversity, a significant variable, was not investigated, limiting generalizability
>
> Compared to scale, architectural diversity is not significant.
>
> Although architectural modifications do influence performance, the effect of scale (model size, or data) is more significant. E.g. in [1], Fig 5, varying:
>
> * depth/width by 40 alters cross-entropy by <3%
> * number of heads by 100 alters cross-entropy by ~1%
>
> In contrast, varying model size can alter cross-entropy by 10-50%.
>
> ## How does the Distillation Scaling Law change when distilling across model families?
>
> Architectural discrepancies should have very little impact distillation efficacy.
>
> Student cross-entropy is influenced by teacher size only through teacher cross-entropy (L104-107). What matters is how well the teacher understands the data distribution, not how the teacher is parameterized. Combining these:
>
> 1. Properties of the teachers are summarized in its cross-entropy
> 2. Variations of the student impact its performance significantly less than the scale of the student (see answer to “architectural diversity”)
>
> ## Evaluations are only on C4, limiting generalizability
>
> The scientific conclusions (L99-81 right) should generalize, even though the coefficients are tied to C4.
>
> We train on C4 (L193) and discuss limitations (L1107-1115).
>
> Recent work [2] and prior studies [1,4] trained on similar datasets with generalizable conclusions.
>
> Coefficients are tied to experiments, a limitation shared by all scaling studies; data manifolds of different intrinsic dimensionality produce different scaling coefficients [3].
>
> Beyond C4, we evaluate on many downstream tasks (App E.1).
>
> ## Techniques addressing the capacity gap problem could have been investigated
>
> Early-stopping mitigation [5] is predicted by our Distillation Scaling Law, interchangeable with teacher size. We thoroughly analyze the capacity gap through scaling laws and kernel methods (App B.3, C.1-2) and agree exploring additional strategies is valuable.
>
> ## Many studies have focused on improving distillation techniques to better calibrate student model
>
> Appendix E.8 is dedicated to distillation calibration. Teachers and students are well-calibrated in the logit distillation setting, which follows from proper scoring rules.
>
> ## The study is limited by its focus on logit-level distillation
>
> Logit-level distillation is popular for training language models, e.g. Gemini, Gemma [6], Apple Foundation Models [7].
>
> The other main technique is SeqKD [9] e.g. DeepSeek-R1 [10], and would be a good choice for extending our work (L1117-1126). We agree the other distillation techniques you mention are interesting.
>
> A challenge in distillation is the many techniques, none of which are sufficiently well-understood for reliable language model training. Our work shows how one of the most popular methods behaves in scenarios of interest, and represents a step in bringing distillation practice at scale towards a science.
>
> ## What happens if the teacher is trained on a different dataset to the student?
>
> E.g. what happens when using LLaMA (trained on $p_{\text{LLaMA}}(x)$) for distilling a student into $p_{\text{new}}(x)$?
>
> Two possibilities:
>
> 1. Properties of the teachers are summarized in its cross-entropy (see architectural diversity above)
> 2. It depends on how different $p_{\text{LLaMA}}(x)$ and $p_{\text{new}}(x)$ are.
>
> Sketching 2. Assume LLaMA is well-trained its next-token distribution,
> $p_T(y∣x)$ reflects $p_{\text{LLaMA}}(y|x)$ not the new one. A student trained on these outputs will also approximate $p(y|x)_{\text{LLaMA}}$. However, in the case of disjoint support, the teacher is evaluated out-of-distribution, failing to provide a meaningful learning signal. 1. applies if the two distributions are sufficiently close; quantifying this closeness would be valuable.
>
> Our setup intentionally uses the same data for distribution teacher and student, letting us isolate algorithmic effects. Exploring settings where the teacher is trained on unseen data would be valuable, though more complex, as it becomes a question of both data and algorithm. We also note that for LLM training in practice, it is quite common for teacher and to use the same data distribution [7, 8], or have the student distilled on a high quality filtered subset.
>
> Thank you again for your insightful comments — they will contribute significantly an improved version of the paper.
>
> [1] https://arxiv.org/abs/2001.08361
>
> [2] https://arxiv.org/abs/2305.16264
>
> [3] https://arxiv.org/abs/2004.10802
>
> [4] https://arxiv.org/abs/2203.15556
>
> [5] https://arxiv.org/abs/1910.01348
>
> [6] https://arxiv.org/abs/2408.00118
>
> [7] https://arxiv.org/abs/2407.21075
>
> [9] https://arxiv.org/abs/1606.07947
>
> [10] https://arxiv.org/abs/2501.12948

---

> > ### Comment · Reviewer_1KoJ · 2025-04-04
> >
> > # 1. Distilling across model families
> > Thank you for raising the point about the teacher's understanding of the data distribution.
> >
> > While that is certainly crucial, I would like to respectfully suggest that the teacher's architecture also plays a significant role, potentially more than implied by the statement "What matters is how well the teacher understands the data distribution, not how the teacher is parameterized."
> > Architecture fundamentally shapes how a model generalizes, which reflects in its cross-entropy. Yet, cross-entropy alone may not be the sole determinant of a good teacher for distillation. As some studies show [1] (as an example only), successful distillation doesn't always require a top-performing teacher model.
> >
> >
> > [1] *Furlanello, Tommaso, et al. "Born again neural networks." International conference on machine learning. PMLR, 2018.*
> > (The core idea is to train a student model using knowledge distillation from a not “perfect” or “fully-optimized” teacher, yet serve as an effective guide for a better student.)
> >
> > It seems plausible that differences in architectural choices like attention variations or training objectives between the teacher and student could influence the distillation process significantly. For instance, such differences might affect model confidence or knowledge transfer in ways not fully captured by cross-entropy metrics alone. We currently lack a clear confirmation that lower teacher cross-entropy reliably translates to superior student performance post-distillation.
> > Given this, a focused study on how architectural disparities impact LLM distillation could offer valuable insights.
> >
> > Also, it might be helpful for readers if the authors could clarify the architectural scope in the paper. Specifying that the study utilizes models with similar architectures would proactively set expectations and frame the contributions within that specific context, highlighting an avenue for future work. I hope this suggestion is helpful.
> >
> > # 2. Calibration
> >
> > The observation that the large teacher models exhibit near-perfect calibration, as presented in the study, warrants further discussion. It is generally understood that larger models, particularly in deep learning, tend to suffer from poor calibration, often displaying overconfidence in their predictions. This discrepancy between established trends and the reported results raises a significant question regarding the methodology and the specific characteristics of the teacher models employed.
> > Could the authors provide a more explanation of why these large teacher models consistently achieve such perfect calibration? Are there specific training procedures, or regularization techniques that contribute to this phenomenon?  It would be beneficial to compare these findings to existing literature on calibration in large models.
> >
> > # 3. Readability
> >
> > Building upon my previous feedback regarding readability, I want to reiterate the importance of making the core findings more accessible within the main body of the manuscript. I quote from my previous review: “I suggest adding more details to figure captions and integrating essential explanations directly into the main text. To make space, the authors could streamline the experimental design section and the table of definitions. This would improve clarity and ensure the findings are more accessible without excessive cross-referencing.”
> >
> > I appreciate the authors’ efforts and hard work. If they could provide more clarifications on the points mentioned above, I will be willing to update my scores.
> >
> > POST - RESPONSE BELOW:
> > I appreciate the author's comments and clarifications. I am updating the score from weak accept to accept.

---

> > > ### Author Response · Authors · 2025-04-04
> > >
> > > ## 1. Distillation across model families
> > >
> > > > “Successful distillation doesn't always require a top-performing teacher model”
> > >
> > > We agree. Even within a single model family, there appears to be an optimal teacher cross-entropy, beyond which the capacity gap emerges.
> > >
> > > > “Born Again Networks show utility of both variations of architectures and imperfect teachers”
> > >
> > > We agree, and note a key difference between BAN and our study: data repetition.
> > >
> > > In BAN, the (imperfect) teacher plays two roles: a learning signal and a function space regularization [1,2], since the model is either trained to convergence in the vision case, or for 40 epochs in the language case.
> > >
> > > In our study, there is no data repetition and the student is underparameterized. This simplifies our setup: the teacher only serves as a learning signal. This distinction enables consistency between our findings and those of BANs.
> > >
> > > > “We currently lack a clear confirmation that lower teacher cross-entropy reliably translates to superior student performance post-distillation. Given this, a focused study on how architectural disparities impact LLM distillation could offer valuable insights.”
> > >
> > > Upon further reflection, we agree.
> > >
> > > Within our chosen transformer → transformer architecture class, our results demonstrate a dependence through the teacher’s cross-entropy alone. Here is a newer plot demonstrating this more clearly (https://postimg.cc/pm4kQv7n) and will be included in an updated main text.
> > >
> > > Across architecture classes (different types of transformers or beyond), we agree it is possible that teacher-student differences could yield different scaling properties that depend on those differences, as indicated by BANs (albeit in a different training regime). This could stem from non-perfectly nested hypothesis classes at given relative sizes, different solution types that yield the same cross-entropy, or any of the reasons you mention.
> > >
> > > Definitely answering this question would require significant further investigation and represents an important direction for future work.
> > >
> > > We will expand the discussion around this point and clarify the architecture to which our results directly apply in main text.
> > >
> > > [1] Self-Distillation Amplifies Regularization in Hilbert Space https://arxiv.org/abs/2002.05715
> > >
> > > [2] Understanding the Gains from Repeated Self-Distillation https://arxiv.org/abs/2407.04600
> > >
> > > ## 2. Calibration
> > >
> > > > “Larger models tend to suffer from poor calibration, often displaying overconfidence”
> > >
> > > We agree that large models *can* be overconfident. Sec. 3 from [3] indicates the overconfidence observed in [4] arises from overfitting, regardless of the training set correctness.
> > >
> > > > “The observation that large teacher models exhibit near-perfect calibration, as presented in the study, warrants further discussion”
> > >
> > > The primary distinction in our setup, compared to works showing overconfidence, is that:
> > >
> > > 1. The model is underparameterized ($N<D$), and
> > > 2. The data is not repeated.
> > >
> > > This means overfitting to the training set does not occur [5], so model overconfidence does not arise to the same extent as in many prior calibration studies.
> > >
> > > Instead, in our setting, increasing model size or training tokens, improves the approximation of the seen distribution $p(x)$ with minimal generalization gap, and thus yields better calibration [6, 7].
> > >
> > > Our observation of good calibration in large models aligns with other findings in language model calibration, e.g.:
> > >
> > > * [8] “In pre-training, we find that model calibration improves as parameter scales and training dynamics increases“
> > > * [9] ”language models can produce well-calibrated predictions for token probabilities on-distribution“, and
> > > * [10] See Fig 8.
> > >
> > > We will gladly add a discussion on this topic in an updated version, thank you for the suggestion.
> > >
> > > [3] Calibrating Deep Neural Networks using Focal Loss https://arxiv.org/abs/2002.09437
> > >
> > > [4] Revisiting the Calibration of Modern Neural Networks https://arxiv.org/abs/2106.07998
> > >
> > > [5] Why you don't overfit, and don't need Bayes if you only train for one epoch https://arxiv.org/abs/2411.14478
> > >
> > > [6] The Calibration Generalization Gap https://arxiv.org/abs/2210.01964
> > >
> > > [7] When Does Optimizing a Proper Loss Yield Calibration? https://arxiv.org/abs/2305.18764
> > >
> > > [8] On the Calibration of Large Language Models and Alignment https://arxiv.org/abs/2311.13240
> > >
> > > [9] Language Models (Mostly) Know What They Know https://arxiv.org/abs/2207.05221
> > >
> > > [10] GPT 4 Technical Report https://cdn.openai.com/papers/gpt-4.pdf
> > >
> > > ## 3. Readability
> > >
> > > Absolutely, thank you. We'll work to improve the readability as suggested. We weren’t able to address this specific point earlier due to character limits.
> > >
> > > We sincerely thank Reviewer 1KoJ for their continued engagement throughout the review process and for working with us to improve our work and make it more useful to the research community.

---

### Decision · Program_Chairs · 2025-05-01

**Decision:**

Accept (poster)

**Comment:**

This paper proposes a Distillation Scaling Law that predicts the performance of distilled models based on compute budget and its allocation between student and teacher. The authors present extensive experiments across model and data scales, and offer practical guidance on when distillation outperforms supervised training. Reviewers raised concerns about the generality of the proposed scaling law, noting it may be dataset-dependent and thus not universally applicable. The authors responded that this limitation is shared across all scaling law studies and doesn't undermine its utility. The requested ablations were included in the authors' response.